# The unexpected smoke layer in the High-Arctic winter stratosphere during MOSAiC 2019-2020

Kevin Ohneiser[1], Albert Ansmann[1], Alexandra Chudnovsky[2], Ronny Engelmann[1], Christoph Ritter[3], Igor Veselovskii[4], Holger Baars[1], Henriette Gebauer[1], Hannes Griesche[1], Martin Radenz[1], Julian Hofer[1], Dietrich Althausen[1], Sandro Dahlke[3], and Marion Maturilli[3]

[1]Leibniz Institute for Tropospheric Research, Leipzig, Germany
[2]Tel Aviv University, Porter School of Earth Sciences and Environment, Tel Aviv, Israel
[3]Alfred Wegener Institute, Helmholtz Centre for Polar and Marine Research, Potsdam, Germany
[4]Prokhorov General Physics Institute of the Russian Academy of Sciences, Moscow, Russia

**Correspondence:** K. Ohneiser
(ohneiser@tropos.de)

**Abstract.**

During the one-year MOSAiC (Multidisciplinary drifting Observatory for the Study of Arctic Climate) expedition, the German icebreaker Polarstern drifted through the Arctic Ocean ice from October 2019 to May 2020, mainly at latitudes between 85°N and 88.5°N. A multiwavelength polarization Raman lidar was operated aboard the research vessel and continuously monitored aerosol and cloud layers up to 30 km height. During our mission, we expected to observe a thin residual volcanic aerosol layer in the stratosphere, originating from the Raikoke volcanic eruption in June 2019, with an aerosol optical thickness (AOT) of 0.005-0.01 at 500 nm over the North Pole area during the winter season. However, the highlight of our measurements was the detection of a persistent, 10 km deep aerosol layer in the upper troposphere and lower stratosphere (UTLS) from about 7-8 km to 17-18 km height with clear and unambiguous wildfire smoke signatures **up to 12 km** and an order of magnitude higher AOT of around 0.1 in the autumn of 2019. Case studies are presented to explain the specific optical fingerprints of aged wildfire smoke in detail. The pronounced aerosol layer was present throughout the winter half year until the strong polar vortex began to collapse in late April 2020. We hypothesize that the detected smoke originated from extraordinarily intense and long-lasting wildfires in central and eastern Siberia in July and August 2019 that reached **the tropopause layer** by the self-lifting process. In this article, we summarize the main findings of our seven-month smoke observations and characterize the aerosol in terms of geometrical, optical, and microphysical properties. The UTLS aerosol optical thickness (AOT) at 532 nm ranged from 0.05-0.12 in October-November 2019 and 0.03-0.06 during the main winter season. The Raikoke aerosol fraction was estimated to be always lower than 15%. **We assume that the volcanic aerosol was above the smoke layer (above 13 km height)**. As an unambiguous sign of the dominance of smoke **in the main aerosol layer from 7-13 km height**, the particle extinction-to-backscatter ratio (lidar ratio) at 355 nm was found to be much lower than at 532 nm with mean values of 55 sr and 85 sr, respectively. The 355-532 nm Ångström exponent of around 0.65 also clearly indicated the presence of smoke aerosol. For the first time, we show a distinct view on the aerosol layering features in the High Arctic from the surface up to 30 km height during the winter half year. Finally, we provide a vertically resolved view on the late winter and early spring conditions

regarding ozone reduction, smoke occurrence, and polar stratospheric cloud formation. The latter will largely stimulate research on a potential impact of the unexpected stratospheric aerosol perturbation on the record-breaking ozone depletion in the Arctic in spring 2020.

## 1 Introduction

As part of the one-year MOSAiC (Multidisciplinary drifting Observatory for the Study of Arctic Climate) expedition (MO-SAiC, 2021) (September 2019 to October 2020), an advanced multiwavelength polarization Raman lidar was operated aboard the German icebreaker Polarstern (Knust, 2017) which served as the main MOSAiC platform for advanced remote sensing studies of the atmosphere. The ice breaker was trapped in the ice from October 2019 to May 2020 and drifted through the Arctic Ocean at latitudes mainly between 85°N and 88.5°N for seven and a half months, continuously monitoring aerosol and cloud layers in the Central Arctic up to 30 km height (Engelmann et al., 2020). MOSAiC was the largest Arctic field campaign ever conducted. The expedition was motivated by the rapid sea ice loss, the unusual Arctic warming, and our incomplete knowledge about the complex processes controlling the Arctic climate (Wendisch et al., 2017, 2019).

We expected to detect a residual stratospheric sulfate aerosol layer, originating from the Raikoke volcanic eruption, with an aerosol optical thickness (AOT) of the order of 0.005-0.01 at 500 nm wavelength over the High Arctic during the main MOSAiC winter months (December - February) as will be outlined in detail in Sect. 3. The volcano in the Kuril Islands (48.3°N, 153.3°E) erupted on 22 June 2019 and influenced the aerosol conditions in the lower stratosphere at all latitudes north of about 20°N during the summer and autumn of 2019 (Kloss et al., 2021; Vaughan et al., 2021; Cameron et al., 2021; Gorkavyi et al., 2021). However, instead of an optically thin volcanic aerosol layer well above the tropopause, we observed a 10 km thick aerosol layer in the upper troposphere and lower stratosphere (UTLS) with an order of magnitude higher AOT of around 0.1, when we started our observation end of September 2019. Wildfire smoke was the dominating aerosol type in the main part of the prominent layer with backscatter center just above the local tropopause. This smoke-containing layer was present over the North Pole range throughout the winter half year until the polar vortex, the strongest of the last 40 years (Lawrence et al., 2020), began to collapse in late April 2020. The occurrence of a persistent aerosol layer with clear wildfire smoke signatures over the entire winter half year 2019-2020 is one of the outstanding events observed during the MOSAiC expedition (Engelmann et al., 2020). The question arose: What was the source for this strong **UTLS** perturbation?

We hypothesize that extreme and long-lasting wildfires in central and eastern Siberia in the summer of 2019 (Johnson et al., 2021) were responsible for the UTLS smoke layer. The main burning phase lasted from 19 July to 14 August 2019 (Johnson et al., 2021). Rather intense fires between 55° and 70°N injected enormous amounts of wildfire smoke into the free troposphere, **in close proximity** to the Arctic region. At the end of July 2019, stagnant air conditions favored the accumulation of smoke over the fire areas north of Lake Baikal and led to a strong increase of the AOT above 2 over several days. In the absence of any pyrocumulonimbus (pyroCb) convection (Fromm et al., 2010) and thus of a very efficient process to transport smoke into the lower stratosphere, we hypothesize that the smoke was lifted into the UTLS height range by the so-called self-lifting process (Boers et al., 2010). Details are given in Sect. 3. In this exceptional wildfire year 2019 (CAMS, 2021), there were numerous

other fires within the Artic Circle (e.g., in Alaska, Greenland, northern Canada). However, it seems so far that none of them was strong enough to contribute to a noticeable enhancement of the AOT in the UTLS height range (Kloss et al., 2021).

Once in the UTLS height range, smoke particles become quickly distributed over the entire northern part of the Northern Hemisphere within a few weeks, as observed and documented for the first time in 2001 (Fromm et al., 2008) and recently confirmed after the record-breaking Canadian fires in the summer of 2017 (Khaykin et al., 2018; Baars et al., 2019; Kloss et al., 2020). The decay of the stratospheric perturbation by the Canadian smoke took more than a half year after injection in August 2017 (Baars et al., 2019).

Importantly, in addition to the strong perturbation **of the UTLS aerosol conditions**, a record-breaking ozone depletion was observed over the Arctic in the spring of 2020 (DeLand et al., 2020; Manney et al., 2020; Wohltmann et al., 2020; Inness et al., 2020; Wilka et al., 2021; Dameris et al., 2021; Smyshlyaev et al., 2021; Bognar et al., 2021). To our best knowledge, a potential impact of stratospheric aerosol on the complex chemical and meteorological processes leading to this strong ozone reduction was however not discussed in any of the published studies. As the main reason for the extraordinarily large ozone depletion, the long-lasting, cold polar vortex was identified. The vortex triggered the development of polar stratospheric clouds over a comparably long time period from January to April 2020, strong chlorine activation, and ozone destruction. However, to what extent the polar smoke and sulfate layers influenced ozone depletion in the spring of 2020 remains an open question that needs to be clarified in the course of the MOSAiC data analysis and future studies on the interplay between smoke, polar stratospheric clouds (PSCs), and ozone depletion.

In this article, we present the main results of the MOSAiC smoke observations. In Sect. 2, a brief description of the Polarstern lidar and the data analysis methods are given. Next, in Sect. 3, we illuminate and discuss the possibility that Siberian wildfire smoke was able to reach the UTLS height range. MODIS (Moderate Resolution Imaging Spectroradiometer) and CALIPSO (Cloud-Aerosol Lidar and Infrared Pathfinder Satellite Observation) lidar measurements provide the observational basis in our discussion. In this context, we were also forced to explain why the CALIPSO aerosol-typing scheme misclassified the wildfire smoke as volcanic sulfate aerosol (Ansmann et al., 2021b). This is discussed in Sect. 3 as well. In Sect. 4, we present our MOSAiC smoke observations. We begin with October and November 2019 case studies and continue with an overview of all lidar observations from October 2019 to May 2020. Finally, in Sect. 5, we provide insight into the ozone, smoke, and polar-stratospheric-cloud conditions during the winter and spring months 2019-2020 to stimulate research on a potential impact of wildfire smoke on the record-breaking ozone depletion. A summary and concluding remarks are given in Sect. 6. An introduction to our entire MOSAiC measurement programm and our research goals is provided by Engelmann et al. (2020).

## 2 MOSAiC lidar data analysis

During the one-year MOSAiC expedition the multiwavelength polarization Raman lidar Polly (POrtabLe Lidar sYstem) (Engelmann et al., 2016) was continuously operated aboard Polarstern. An overview of the Polarstern lidar instrument and all retrievable aerosol products is given by Engelmann et al. (2020). Continuous, automated measurements of aerosol and cloud profiles up to stratospheric heights were collected from 26 September 2019 to 2 October 2020. From the beginning of October

2019 to the beginning of April 2020, the research vessel was north of 85°N and reached the maximum northern latitude of 88.6°N on 20 February 2020. A photograph of the Polarstern in the ice and snow-covered Arctic Ocean together with a photograph of the main ship-based MOSAiC atmospheric measurement platforms aboard Polarstern is shown in Fig. 2 in Engelmann et al. (2020). Six containers for in situ aerosol monitoring and for active remote sensing of aerosols and clouds with lidars and

5 radars were deployed on the front deck of the research vessel including the ARM (Atmospheric Radiation Measurement) mobile facility AMF-1 (Lindenmaier et al., 2019, updated hourly).

The Polly instrument is mounted inside the OCEANET-Atmosphere container of the Leibniz Institute for Tropospheric Research (TROPOS). This container is designed for routine operation aboard Polarstern between Bremerhaven, Germany, and Cape Town, South Africa and Punta Arenas, Chile (Kanitz et al., 2011, 2013), and was operated for the first time in the Arctic

during a two-month campaign in June and July 2017 (Griesche et al., 2020). The OCEANET Polly instrument belongs to the lidar network PollyNET (Baars et al., 2016) which is part of the European Aerosol Research Lidar Network (EARLINET) (Pappalardo et al., 2014) organized within the Aerosols, Clouds and Trace gases Research InfraStructure (ACTRIS) project (ACTRIS, 2021).

The setup and basic technical details of the Polly instrument are given in Engelmann et al. (2016). Linearly polarized

laser pulses at 355, 532, and 1064 nm are transmitted into the atmosphere at an elevation angle of 85°. The Polly instrument has 13 measurement channels (polarization sensitive channels, elastic-backscatter, water vapor and nitrogen Raman channels, for near-range and far-range profiling). Height profiles of the particle backscatter coefficient at the laser wavelengths, of the particle extinction coefficient at 355 and 532 nm, respective extinction-to-backscatter ratio (lidar ratio) at 355 nm and 532 nm, the particle linear depolarization ratio (PLDR) at 355 nm and 532 nm (Baars et al., 2016, 2019; Hofer et al., 2017; Haarig et al.,

2018; Ohneiser et al., 2020), and the mixing ratio of water vapor to dry air by using the Raman lidar return signals of water vapor and nitrogen (Dai et al., 2018) can be determined.

The Raman lidar method was exclusively used to determine particle backscatter and extinction profiles in the aerosol layer (<20 km height). The particle backscatter coefficient is obtained from the measured ratio of the elastic backscatter signal (355 nm, 532 nm) to the respective Raman signal (387 nm, 607 nm). The 1064 nm backscatter coefficient is calculated from

25 the ratio of the 1064 nm elastic backscatter signal to the 607 nm nitrogen Raman signal. In the retrieval of the extinction coefficient, a least-squares linear regression (Pappalardo et al., 2004; Russo et al., 2006) is applied to the respective Raman signal profiles. At heights above 20 km, the backscatter and extinction properties of the stratospheric background aerosol are determined from the elastic backscatter signal profiles (alone) by assuming a particle extinction-to-backscatter ratio (lidar ratio) of 50 sr (Fernald, 1984). Calibration heights were generally set into the clean stratosphere (above 20 km height) and in

the retrieval of background aerosol at about 35 km height. The backscatter signal intensities were always sufficient above the detection limits for heights of 25-35 km. This means that the aerosol layers could be always fully resolved from base to top in terms of backscatter profiles.

The particle linear depolarization ratio (PLDR) is obtained from the cross-to-co-polarized signal ratio after correction of Rayleigh contributions to light depolarization. "Co" and "cross" denote the planes of polarization parallel and orthogonal to

35 the plane of linear polarization of the transmitted laser pulses, respectively. The depolarization ratio is observed at 355 and

532 nm and can be used to discriminate spherical particles (producing depolarization ratios close to zero) from non-spherical particles such as dust particles (causing depolarization ratios around 0.3 at 532 nm) (Groß et al., 2015). In the case of Polly, the cross-polarized and total (cross-polarized + co-polarized) backscatter signals are measured. The specific approach to obtain the volume depolarization ratio (VDR) from the Polly observations is described by Engelmann et al. (2016).

Different expressions for the Ångström exponent, a well-established parameter to characterize the spectral dependence of aerosol optical properties, are computed. The Ångström exponent $Å_{x,\lambda_i,\lambda_j} = \ln(x_i/x_j)/\ln(\lambda_j/\lambda_i)$ describes the wavelength dependence of an optical parameter $x$ (backscatter $\beta$ or extinction $\sigma$) in the spectral range from wavelength $\lambda_i$ to $\lambda_j$.

Although PollyNET delivers automatically calculated profiles, the lidar observations were manually analyzed for the smoke and aerosol layers. In order to accurately determine the optical properties with high signal-to-noise ratio, temporal averaging
over comparably long time periods of 2-6 hours were usually necessary to obtain extinction profiles up to 17-20 km height. The basic elastic-backscatter and Raman signal profiles were vertically smoothed with a window length of 457.5 m (61 height bins, 7.5 m vertical resolution) in the case of the backscatter and depolarization-ratio computations. Particle extinction and extinction-to-backscatter ratio (lidar ratio) profiling is based on a least-squares regression analysis (Pappalardo et al., 2004; Russo et al., 2006). Here, we used regression window lengths of 2002.5 m (267 height bins) in the computations. Next,
we further smoothed the profiles of the lidar products (backscatter, extinction, depolarization, and lidar ratios) linearly with window lengths increasing from 8 bins (at smoke layer base) to 11 bins (at layer top) in the case of the particle backscatter and depolarization ratio profiles, and increasing from 15 bins (base) to 20 bins (top) and from 45 bins (base) to 53 bins (top) in the case of the particle extinction and lidar ratio profiles, respectively.

Polar stratospheric clouds (PSCs), which frequently developed during the winter months (December to February), disturbed
the determination of the aerosol backscatter and extinction profiles. Fortunately, most PSCs formed at heights above 17 km and the main part of the aerosol layer was below 17 km height so that the removal of the PSC-affected aerosol profile segments had no large influence on the smoke and aerosol products. We used the 1064 nm signal profile to identify the PSC-affected parts (by visual inspection) and cut out the respective backscatter and extinction contributions from the lidar data set. We estimate that a residual PSC-related uncertainty is $\leq 5\%$ in terms of the 532 nm AOT. Backscatter and extinction contributions from
optically thin PSCs occurring within the aerosol layers were not removed. We will discuss the impact of PSC occurrence on our smoke observations in Sect. 4.

To obtain estimates of microphysical properties of the smoke particles such as mass, volume, and surface area concentration, a conversion method is available (Ansmann et al., 2021a). The 532 nm particle backscatter coefficients are input in the retrieval process. In the case of the MOSAiC data analysis, the smoke backscatter coefficients are converted to smoke extinction profiles,
the first step, by assuming a smoke lidar ratio of 85 sr at 532 nm (mean value of all observations from October 2019 to March 2020). In the second step, the extinction coefficients are converted to microphysical properties by respective smoke conversion factors. To obtain smoke mass concentrations, we assume a smoke particle density of 1.15 g cm$^{-3}$ (Ansmann et al., 2021a). Li et al. (2016) investigated different smoke aerosols in the laboratory and found smoke particle densities of 1.1 to 1.4 g cm$^{-3}$ with values of about 1.05±0.15 g cm$^{-3}$ for organic carbon and 1.8 g cm$^{-3}$ for elemental carbon. Chen et al. (2017) reviewed

many smoke studies and concluded that the smoke particle density is 1.0-1.9 g cm$^{-3}$. Thus in cases with 2-10% of black carbon (BC) the overall smoke particle density should be in the range of 1.0–1.3 g cm$^{-3}$.

As an alternative approach, the retrieval of microphysical properties from backscatter and extinction lidar data by inversion with regularization is available (Müller et al., 1999, 2004; Veselovskii et al., 2002; Wandinger et al., 2002). In the MOSAiC data analysis, we used the method of Veselovskii et al. (2002). The determination of the particle volume size distribution, volume and surface area concentrations, and the complex refractive index (assumed to be wavelength independent in the retrieval) from a small number of measured backscatter (at 355, 532, and 1064 nm) and extinction coefficients (at 355 and 532 nm) is a nonlinear, inverse, ill-posed problem, i.e., solutions are non-unique and highly oscillating without the introduction of appropriate mathematical tools such as regularization. The single scattering albedo SSA, defined as the ratio of scattering-to-extinction coefficient, is finally calculated from the retrieved particle size distribution and complex refractive index characteristics with an uncertainty of $\pm 0.05$.

Table 1 provides an overview of the main lidar products, the selected vertical smoothing and regression window lengths, signal averaging time intervals, and uncertainties in the directly measured and derived microphysical aerosol properties. The uncertainties in the lidar products are obtained from simulation studies, error propagation analysis, and comparison with ground-based sunphotometer and in situ measurements or airborne in situ measurements (e.g., Wandinger et al., 2002; Müller et al., 2004). The volume size distribution, not listed in Table 1 as product, can be derived with high accuracy, especially in cases of pronounced accumulation modes. In situ observations clearly show that size distributions of aged smoke particles are monomodal (Dahlkötter et al., 2014).

Auxiliary data are required in the lidar data analysis in form of temperature and pressure profiles in order to calculate and correct for Rayleigh backscatter and extinction influences on the measured lidar return signal profiles. As an important contribution to MOSAiC, radiosondes were routinely launched every 6 hours throughout the entire MOSAiC period (Maturilli et al., 2021).

Additionally, we compare the Polly observations with CALIOP data (CALIOP, 2021) and also with measurements with the Spitsbergen lidar KARL (Koldewey Aerosol Raman Lidar) (KARL, 2021; Hoffmann et al., 2009; Ritter et al., 2016). The lidar is located in Ny-Ålesund (Svalbard, Norway, 78.9°N, 11.9°E). In the discussion of a potential impact of the wildfire smoke on the record-breaking ozone depletion in the spring of 2020, we use the MOSAiC ozone profiles measured with ozone sondes launched at Polarstern on a regular schedule from October 2019 to May 2020 (von der Gathen and Maturilli, 2020; Wohltmann et al., 2020).

For our studies of the smoke in the UTLS height range, a good knowledge of the tropopause height is important. The tropopause was computed from the MOSAiC radiosonde temperature and pressure profiles (Maturilli et al., 2021) by using the approach of the Global Modeling and Assimilation Office (GMAO), Goddard Space Flight Center, Greenbelt, Maryland, USA (GMAO, 2021). In this approach, the tropopause height $z_{\text{TP}}$ is found from the height profile of the difference $\alpha T(z) - \log_{10} p(z)$ with $\alpha = 0.03$, temperature $T$ (in Kelvin), pressure $p$ (hPa), and height $z$ (m). The tropopause pressure $p(z_{\text{TP}})$ is defined as the pressure where the defined difference reaches its first minimum above the surface. If no clear minimum was found up to $z = 13000$ m over Polarstern, a tropopause height $z_{\text{TP}}$ was not assigned. The obtained tropopause heights agree well with

the ones we obtain by applying the definition of the World Meteorological Organization (WMO, 1992) to the radiosonde temperature profiles and considering refinements in the determination described by Klehr (2012). In most cases, the GMAO approach delivers 20-80 m lower tropopause levels and produces less outliers.

## 3  Siberian wildfires in the summer of 2019: A source of stratospheric smoke? Possible explanation

From the first day of the MOSAiC observations on, in late September 2019, we observed a prominent and well-aged strato-spheric aerosol layer with a broad maximum (in terms of backscattering) around 10 km height, just above the local tropopause. The layer showed smooth internal structures and clear wildfire smoke signatures up to about 12-13 km height according to our Raman lidar observations. Further lidar observations at Leipzig (51.3°N, 12.4°E), Germany (Ansmann et al., 2021b), and Ny-Ålesund, Svalbard (78.9°N, 11.9°E), Norway, indicated a strong increase of the **UTLS AOT** in August 2019 which cannot be explained by the conversion of $SO_2$ plumes, originating from the Raikoke volcanic eruption, into sulfate aerosol. The emitted volcanic $SO_2$ plumes were converted to sulfuric-acid-containing water droplets (about 75% $H_2SO_4$ and 25% $H_2O$) within a few weeks after the eruption in June 2019. The maximum stratospheric sulfate aerosol load occurred in the first half of August 2019 followed by a decrease of the aerosol pollution from September to December 2019 by about a factor of 2 (Kloss et al., 2021; Cameron et al., 2021).

Based on a detailed study of the stratospheric perturbation after the eruption of the Sarychev volcano (48.1°N, 153.2°E, a neighbor volcano of Raikoke) in June 2009 by Haywood et al. (2010) and the fact that the Raikoke $SO_2$ emission was 20% higher than the $SO_2$ amount reaching the stratosphere after the Sarychev eruption, we expected a sulfate AOT at 550 nm of about 0.01 over the High Arctic in November 2019 and around 0.005 during the main winter months. The maximum Sarychev-related 550 nm AOT was 0.02 and the e-folding decay time of the stratospheric perturbation about 80±10 days. Observations of the maximum Raikoke sulfate AOT of 0.025 (mean value for the latitudinal belt from 40°–55°N in August 2019 at a minimum influence of smoke contributions) and modeling studies of the decrease of the Raikoke sulfate AOT by a factor of 2 from August to October 2019 presented by Kloss et al. (2021) corroborated our assumption and are in good agreement with respective decay studies of the Sarychev-related stratospheric perturbation presented by Haywood et al. (2010).

Most of the volcanic sulfat aerosol formed at heights above 13-15 km according to ground-based lidar observations over Leipzig and the United Kingdom (Vaughan et al., 2021) and the spaceborne CALIPSO lidar observations over the northern hemisphere. On the other hand, the main smoke layer was found between 6 and 12 km height over the Polarstern with maximum backscatter values around 10 km height. It is thus likely that the aerosol traces above 13-15 km height visible in the backscatter profiles up to 17–19 km height during the MOSAiC campaign were caused by volcanic aerosol.

The strong changes in the UTLS aerosol load, as observed over Leipzig and Ny-Ålesund in August 2019 pointed to the extraordinarily strong and long lasting forest fires in central and eastern Siberian in July and August 2019 (Johnson et al., 2021). Figure 1 visualizes the tremendous environmental disaster over the central and eastern parts of Siberia at the end of July 2019. Several large fire centers are visible north, northwest and northeast of Lake Baikal (53.5°N, 108°E) within an area of roughly 1000 km× 2000 km (95°–125°E). The time series of monthly mean AOT (550 nm) in Fig. 2, measured with MODIS

(MODIS, 2021) aboard Aqua and Terra over the last 20 years, indicates that the fire season in 2019 belongs to the strongest during the last two decades.

An overview of the smoke situation in terms of mean AOT (for the time period from 20 July to 20 August 2019) is given in Fig. 3. We performed a very careful MODIS data analysis taking all sensitive error sources (especially cloud interference) into consideration (Chudnovsky et al., 2013b, a). The heaviest fires occurred in the region indicated by box 1 (about 650 km x 750 km), located directly north of Lake Baikal (53.5°N, 108°E). Many of the mean AOT values (average of the 27-day long main fire period) exceeded 1.5, and over some regions even 2.0. The daily time series of the area-mean 550 nm AOT for the main fire centers in Fig. 3 shows that the AOT (in box 1) ranged from 1-2.5 over a seven-day period (in July 2019) and was continuously >1 for more than 8 days in August 2019. HYSPLIT ten-day backward trajectories (not shown) indicated stagnant atmospheric conditions (low wind speeds throughout the troposphere) within box 1, especially from 24 to 27 July 2019 (HYSPLIT, 2021). As indicated in Fig. 3, the smoke obviously accumulated during these low-wind conditions in box 1 so that the daily mean 550 nm AOT steadily increased from 0.4 (21 July) to 2.7 (27 July 2019).

We theorize that self-lifting of the smoke (Boers et al., 2010; de Laat et al., 2012; Torres et al., 2020) was initiated at these favorable stagnant conditions and led to the ascent of optically dense aerosol layers. Strong absorption of solar radiation by the black-carbon-containing smoke heated the surrounding air in lofted smoke plumes which were injected into heights of 3-5 km before. The created buoyancy then forced the plumes to further ascend up to the upper troposphere (up to the tropopause at around 10 km). This is the only plausible explanation for an efficient vertical transport over several kilometers in the absence of pyroCb convection. It is well accepted that pyroCb convection can lift large amounts of wildfire smoke into the lower stratosphere (Fromm et al., 2010; Peterson et al., 2018; Ohneiser et al., 2020). However, from mid-July to mid-August 2019, vigorous thunderstorms over the burning areas in Siberia did not develop.

Peterson et al. (2021) argue that prolific pyroCb activity caused the lifting of Siberian smoke into the stratosphere in July and August 2019. We cannot confirm this statement. We checked the available satellite observations over central and eastern Siberia and found no evidence for any noticeable impact of strong cumulus convection and thunderstorm activity over the Siberian fire places on vertical smoke transport. For this reason, we introduced the self-lifting process as the missing link between the Siberian fires in the summer 2019 and the observed stratospheric wildfire smoke layer in late summer, autumn, and winter.

The CALIPSO lidar observations in Fig. 4 corroborate our hypothesis. Smoke layers were observed with the spaceborne lidar in the direct neighborhood of the intense fire areas (box 2 in Fig. 3) up to the tropopause at 10 km on 26 July 2019, 21 UTC. All layers above the tropopause very likely indicate Raikoke-related volcanic aerosol. The backward trajectories in Fig. 5 provide an impression of the stagnant smoke conditions below the CALIPSO flight track. Trajectories arriving at 3 and 7 km height show a direct smoke transport from box 2 to the flight track of the CALIPSO lidar. The trajectory for the arrival height of 9 km (and all trajectories arriving higher up) was not in direct contact with the smoke sources, i.e, a direct smoke uptake during the travel over the fire region was not possible. Thus, the smoke detected by the CALIPSO lidar around 9–10 km height must have been lifted by several kilometers before.

Self-lifting processes were reported several times after major wildfire events (de Laat et al., 2012; Khaykin et al., 2018, 2020; Torres et al., 2020; Ohneiser et al., 2020; Kablick et al., 2020), however, for stratospheric smoke layers only. To our best

knowledge, there is no report in the literature on observed self-lifting processes in the troposphere up to the tropopause. The unique stagnant weather conditions, i.e. the absence of strong winds and wind shear, may be one of the key prerequisites to start well-organized self-lifting processes within the troposphere. **When these lifted smoke plumes are afterwards advected away from the source regions, meteorological processes around the jet stream will result in mixing of the smoke into the lowermost stratosphere.**

This new aspect of tropospheric self-lifting will be illuminated in detail in a follow-up article based on extensive simulation studies. Here, we present a first result (Fig. 6) to provide an impression how much time it takes to lift wildfire smoke at cloudfree or cloudy conditions from the smoke injection heights (e.g., 3-5 km) to UTLS heights of 9-10 km (tropopause level at 10 km in the simulation). We used the radiative transfer model ecRAD (Hogan and Bozzo, 2018) to compute heating rates and subsequently we computed ascent rates as a function of these heating rates for given potential temperature gradients (Boers et al., 2010). In Fig. 6, a smoke layer (with Gaussian shape in terms of light extinction profile) intially centered at 4 km height was simulated. The daily course of the sun for summertime conditions at 65°N was considered. According to Fig. 3, the 500 nm AOT increased from 0.5 to 2.5 from 22 to 26 July 2021 in box 1. This may indicate a day-by-day production of smoke layers with mean AOT of the order of 0.5 at stagnant conditions. The smoke started to ascend and did not leave the tropospheric column for several days so that the AOT increased steadily over days. The lifting of such smoke layers with 500 nm AOT of 0.5, as well as of layers with AOT of 1.0, is simulated in Fig. 6. The BC fraction was set to 15%, 7.5% and 5% during the first 12 h, the next 12 h, and for the rest of the simulation period, respectively, to roughly consider particle aging during the first 24-36 h after injection. For simplicity, we used the layer mean heating rate together with the potential temperature gradient (radiosonde, Olenek, Siberia, July-August mean temperature profile) to compute the lifting rates (Boers et al., 2010). However, it should be mentioned that the vertical profile of the heating rate at cloudfree conditions shows an inhomogeneous behavior (not presented here) with stronger heating at the top and lowest heating at the base of the smoke layer so that a coherent ascent of a given smoke layer over several kilometers is not possible in reality. Instead diffuse aerosol structures are more likely to develop and ascend with different speed.

As can be seen in Fig. 6, smoke can be lifted in stagnant conditions within about 2-6 days up to UTLS heights in the case of moderate smoke layer AOTs. In the case of the simulation with a closed cloud deck below the ascending smoke layer, the solar radiation available for absorption by the smoke layer (with a relatively low AOT) is a factor of 1.5-1.9 higher compared to cloudfree conditions because of strong reflection of solar radiation by liquid-water cloud layers (with albedo close to 1.0). Note that MODIS indicated rather inhomogenous smoke-related AOT fields over central and eastern Siberia and north of the burning areas. The AOT partly exceeded 3.0 over large areas. Figure 6 thus provides only insight into typical self-lifting time scales for moderately enhanced AOT conditions. As mentioned, an interesting aspect is also that heating of smoke layers is vertically inhomogeneous so that a coherent ascent of layers over days, even in rather stagnant conditions (no change of wind direction and speed with height), seems to be almost impossible in the troposphere. The resulting incoherent aerosol structures are visible in the CALIPSO observation in Fig. 4.

The comparably slow ascent (compared to fast lifting within 30-90 minutes in the case of pyroCb convection) and thus comparably long residence times in the humid troposphere (with high levels of condensable gases) has consequences for the

chemical, microphysical and optical properties of the smoke particles. At tropospheric conditions fast particle aging takes place and can be completed within 3-4 days. At the end of the particle aging process, most of the smoke particles have developed a perfect spherical core-shell structure, as clearly indicated by the low particle depolarization ratio measured with lidar. This, in turn, has consequences for a lidar-based aerosol typing strategy and the discrimination of smoke from sulfate particles as will be discussed in Sect. 3.1. The context of smoke aging, changing morphological properties of smoke particles and resulting optical properties is presented in Ansmann et al. (2021a).

Below, we summarize our key findings and critical points to be considered:

(1) The self-lifting hypothesis is of key importance in our attempt to relate the MOSAiC UTLS smoke observations from October 2019 to May 2020 with the strong fires in Siberia in July-August 2019.

(2) Our Raman lidar observations at Leipzig corroborate the hypothesis that Siberian fires reached the lowermost stratosphere and that the smoke particles were well aged and thus spherical in shape. During a clear night, on 13-14 August 2019, we observed a 4-km thick stratospheric layer from the tropopause to 14 km with layer-mean particle extinction coefficients of 30 and 25 $Mm^{-1}$ at 355 and 532 nm, respectively, Ångström exponent of about 0.5 (355-532 nm spectral range), and lidar ratios around 70 sr (355 nm) and 100 sr (532 nm) (Ansmann et al., 2021b). These optical fingerprints are indicative of wildfire smoke, but not for Raikoke-related volcanic sulfate particles. The measured particle depolarization ratio was close to zero as expected for spherical smoke particles. Similar smoke properties were then obtained over Leipzig on 23 August 2019 as well. HYSPLIT 13-day backward trajectories (for 14 August, 01:00 UTC) indicated travel times of more than 14 days from central and eastern Siberia, over Alaska, northern Canada, Greenland to central Europe for the trajectories arriving at 12 and 14 km height over Leipzig.

(3) Importantly, also the CALIPSO lidar measured particle depolarization ratios of close to zero throughout the troposphere in the case of the measurement shown in Fig. 4.

(4) In July 2019, the stratospheric aerosol layers over Leipzig were much thinner (1-2 km in depth). Unfortunately, only the 355 nm AOT, layer-mean extinction coefficient, and lidar ratio could be determined. At 532 nm, a trustworthy data analysis was not possible because of the too low laser-light attenuated by the stratospheric particles. The UV AOT of 0.015 to 0.02, layer-mean extinction values of 5-10 $Mm^{-1}$, and a lidar ratio of 45±15 sr point to the presence of volcanic aerosol in a measurement performed on 23 July 2019.

(5) It should be emphasized again that all our explanations here are based on theoretical arguments. It still remains an open question whether or not the fires in central and eastern Siberia were the source for the UTLS smoke observed over the North Pole region during the MOSAiC expedition. However, the aerosol we observed during the MOSAiC campaign was definitely a smoke-dominated pollution. There may have been Raikoke sulfate aerosol as well, especially above 12-13 km height. Our MOSAiC lidar did not allow the determination of trustworthy lidar ratios for heights above the main aerosol layer (above 12-13 km height) so that it remains an open question what aerosol type dominated in the upper part of the detected aerosol layer over the North Pole region up to 17-19 km height. This point will be further discussed in Sect. 4.

### 3.1 CALIPSO aerosol typing challenge: Siberian smoke misclassified as volcanic sulfate aerosol

Next, we ask the following questions: "Why did CALIPSO not identify Siberian fires (or sources for aged smoke) in the Arctic?" and "Does the CALIPSO aerosol typing scheme consider self-lifting events and corresponding occurrence of spherical smoke particles?" These questions are important because the CALIPSO lidar is the world's unique spaceborne lidar for monitoring the global aerosol distribution including the long-range transport of smoke emissions by strong fires. To the best of our knowledge, there is no article so far that considers the Siberian smoke in the interpretation of measured stratospheric aerosols in 2019 (see, e.g., Kloss et al., 2021).

To answer these question, we need to briefly explain the latest version (V4) of the CALIPSO aerosol typing scheme (Kim et al., 2018). As also discussed by Ansmann et al. (2021b), the stratospheric aerosol classification is primarily based on the particle depolarization measurements. If the depolarization ratio is close to zero (and indicates spherical particles), a stratospheric aerosol layer is usually categorized as volcanic sulfate layer, as long as the atmospheric conditions do not allow for PSC formation. Only if the depolarization ratio is enhanced (indicating nonspherical particles) and is in the range of 0.075 to 0.15, would this layer be classified as smoke layer. There is another option to identify smoke which is based on a threshold value for the spectral dependence of the backscatter coefficient, but this option is of minor importance according to our long lidar record on smoke observations, and thus will be ignored here.

The enhanced smoke depolarization ratio was observed only when pyroCb events were responsible for the stratospheric smoke (Haarig et al., 2018; Hu et al., 2019; Ohneiser et al., 2020). The fast lifting of freshly emitted, non-spherical particles within cumulus towers prohibit particle aging and the development of a spherical shape before the particles reach the stratosphere. In the dry stratosphere (in an environment with low amounts of condensable gases), aging processes are very slow. It takes 3-6 months before the smoke particles become spherical in shape so that the particle depolarization ratio reaches values close to zero (Baars et al., 2019). These non-spherical pyroCb-related smoke particles can be easily discriminated from volcanic sulfate layers by using the depolarization ratio information (Kim et al., 2018; Christian et al., 2020; Ansmann et al., 2021b).

The case of smoke self-lifting events and corresponding occurrence of spherical smoke particles immediately after entering the stratosphere is not considered in the CALIPSO aerosol typing scheme. These particles are automatically assigned as sulfate particles because of the observed low depolarization ratio. A good example for the misclassification is our lidar observation at Leipzig on 14 August 2019, mentioned above (Ansmann et al., 2021b). This 4-km thick layer is stored in the CALIPSO data base as sulfate layer. Note, that also pyroCb-related smoke layers will be misclassified as sulfate layers after 3-6 months of residence in the stratospheric, i.e., when the depolarization ratio approaches values below 0.05.

### 4 MOSAiC observations of UTLS smoke

Figure 7 shows the track of the RV Polarstern from October 2019 to May 2020. The icebreaker was slowly drifting through the ice at latitudes ≥85°N for more than seven months. The highest northern latitude with 88.6°N was reached around 20 February

2020. Favorable weather conditions for lidar measurements up to the middle stratosphere were given most of the time. Note that the spaceborne CALIPSO lidar is blind for the region >81.8°N.

Figure 8 shows that a strong polar vortex built up during the central winter months and controlled the weather and aerosol conditions from mid-December to May. According to Lawrence et al. (2020), the wintertime westerly winds in the polar stratosphere (from about 15–50 km) were extraordinarily strong during the Northern Hemisphere winter of 2019-2020. As a consequence, very stable weather and air flow pattern prevailed below the vortex, and as a result, there were almost constant aerosol conditions in the UTLS height range throughout the winter.

In Figure 8a, the time series of the scaled potential vorticity $sPV$ (Millán et al., 2021) along the route of the Polarstern drift is presented. $sPV$ is a useful quantity to detect the polar vortex edge (Bognar et al., 2021)). ERA5 (ECMWF reanalysis of the global climate, fifth generation, ECMWF: European Centre for Medium-Range Weather Forecasts) data (ERA-5, 2021) of the potential vorticity and temperature on the pressure fields of 30, 50, 70, 100, and 125 hPa were used in the computation of $sPV$ by applying Eq. (2) in Millán et al. (2021). A $sPV$ value of $> 160 \times 10^{-6}$ s$^{-1}$ (Bognar et al., 2021) indicates that the Polarstern was below the polar vortex from January until 20 April 2020 when the vortex began to collapse.

In Figure 8b, the temperatures for six height levels are shown. The temperatures continuously decreased with time. Favorable conditions for PSC formation at heights above 17 km were given in January and February 2020, when the temperatures dropped below $-78°$C. The low variability in the temperature time series for heights below 18 km from mid December 2019 to mid April 2020 corroborates that the weather pattern were rather stable and meridional air-mass exchange was widely suppressed.

## 4.1 Case studies confirming the smoke dominance in the UTLS aerosol layer compared to the Raikoko volcanic impact

A first brief overview of our UTLS smoke observations from October 2019 to May 2020 was given in Engelmann et al. (2020) based on 532 nm AOT and layer-mean extinction coefficient. In this section, we present the full set of observed optical and microphysical properties of the smoke layer. We begin with two case studies to demonstrate that smoke was undoubtedly the dominating aerosol type in the UTLS aerosol layer over the North Pole region. Figure 9 shows the aerosol situation on 25 October 2019. The Polarstern was at 85.4°N and 127.9°E at 10:00 UTC. A haze layer with stratiform structures extended from the surface up to 4.5 km height and the UTLS aerosol layer with smooth structures ranged from 5.5 to 15 km height as indicated by dashed lines (Fig. 9). The smoke layer is clearly visible between 6 and 11 km height. We determined the layer base and top heights by visual inspection of the mean height profile of the 1064 nm backscatter coefficient for the shown period. The base height is the altitude at which the 1064 nm backscatter coefficient started to increase again (above the top of the haze layer). The layer top was set to the height where the total-to-Rayleigh backscatter ratio at 1064 nm dropped below a value of 1.1.

In Fig. 10, we show HYSPLIT backward trajectories for the arrival height of 10 km (center of the smoke layer). They clearly indicate stable air flow conditions and suggest that the polluted UTLS air mass was trapped in the Arctic circulation system. The same air mass moved twice over the location of RV Polarstern within 10 days in these late October days.

Figure 11 presents mean height profiles of the basic lidar products for the cloud-free period in Fig. 9 from 9:30–14:00 UTC. A similar case, measured on 7 November 2019, is shown in Fig. 12. Here, the Polarstern observations at 85.9°N are compared with the AWI multiwavelength Raman lidar observations at Spitsbergen (700 km south of the icebreaker), conducted three days before on 4 November 2019. As can be seen in both figures, the maximum backscatter and extinction values were found around 9-10 km height, and thus well within the lower stratosphere. The maximum extinction coefficients in the center of the smoke layer were in the range of 15 to 25 Mm$^{-1}$ and the AOT of the UTLS polar smoke layer of the order of 0.1 at 532 nm on 25 October, and on 4 and 7 November 2019. The wavelength dependence of the extinction coefficient $\sigma_\lambda$, expressed in terms of the Ångström exponent for the 355–532 nm spectral range, was low with values around 0.7. The high particle extinction coefficients of up to 15 Mm$^{-1}$ and 532 nm AOT values of 0.1 are in contradiction with the expected extinction values of 1-2 Mm$^{-1}$ and AOTs of 0.01-0.015 in case of a volcanically disturbed stratospheric aerosol layer. Furthermore, volcanic aerosol particles typically cause Ångström exponents clearly >1.0 (Mattis et al., 2010). Such a low Ångström exponent around 0.7 is typical for wildfire smoke (Haarig et al., 2018; Ohneiser et al., 2020).

The 532 nm extinction profiles in Fig. 11c were used to estimate mass and surface-area concentration profiles by applying the smoke conversion factors of Ansmann et al. (2021a). Surface area values of 0.2-0.4 cm$^2$ m$^{-3}$ in the center of the smoke layer in Fig. 16 correspond to 20-40 $\mu$m$^2$ cm$^{-3}$. This latter unit is typically used in PSC studies (Jumelet et al., 2008, 2009). These surface area values are in the same range as found for PSC particle layers.

The particle and volume linear depolarization ratio in Fig. 11b and 12b were very low at both laser wavelengths of 355 and 532 nm. Differences between the KARL and Polly observations in Fig. 12b are insignificant. Both, the residual volcanic as well as the dominating smoke particles were spherical according to the lidar observations.

The most striking feature and an unambiguous optical fingerprint of the dominance of aged smoke in the observed polar stratospheric aerosol layer is the unusual wavelength dependence of the lidar ratio with a significantly lower extinction-to-backscatter ratio at 355 nm than at 532 nm, as shown in Fig. 11d and 12d. These unique fingerprints are also highlighted in Table 2. Disregarding the source of the smoke material (Australian, Canadian, or Siberian fires), the lidar ratio at 355 nm is typically >20 sr lower than the one at 532 nm. Furthermore, the 532 nm lidar ratio for aged smoke is frequently enhanced and in the range from 65 to 100 sr. The MOSAiC half year mean smoke lidar ratio is 85 sr at 532 nm. In the case of volcanic sulfate particles (Sarychev eruption in June 2009) and volcanic ash (Eyjafjallajökull eruption in April 2010), the lidar ratios at 355 and 532 nm were similar, a clear wavelength dependence was not found. The same holds for mineral dust. The dust values of Hofer et al. (2020) and Groß et al. (2015) indicate almost the full the range of measurable dust lidar ratios. Usually, a low spectral dependence is found. Sometimes, the backscatter coefficient is larger at 532 than at 355 nm (when very large particles dominate, preferably in near-soure regions) whereas the extinction coefficients are the same so that the dust lidar ratio at 355 nm is then clearly larger than at 532 nm (Veselovskii et al., 2016). The aged Arctic haze was observed over Leipzig in April 2002 (Müller et al., 2004). Note, that Arctic haze can also contain aged biomass burning particles and is thus also able to produce an inverse spectral behavior of the lidar ratio (Ritter et al., 2016; Engelmann et al., 2020).

The lidar ratios in Fig. 11d and 12d are only shown up to 12 km . The retrieval of extinction-to-backscatter ratios was no longer trustworthy for heights above 12-13 km because of the decreasing particle concentration and increasing signal noise.

Thus, it is possible that the residual aerosol traces in the upper part of the MOSAiC aerosol layers with top at 17-19 km height consist of volcanic sulfate aerosol particles. If we assume a dominance of smoke particles up to 12 km (532 nm AOT of 0.09, mean extinction coefficient of 15 $Mm^{-1}$ and mean lidar ratio of 85 sr) and a dominance of Raikoke aerosol particles from 12-16 km height (AOT of 0.012, mean extinction coefficient of 3 $Mm^{-1}$, mean lidar ratio of 45 sr) in Figure 11, we end up

with a Raikoke-related AOT contribution of 10–15% to the overall 532 nm AOT of the UTLS aerosol layer. This value for the Raikoke AOT contribution of 0.012 end of October 2019 is in line with the estimation starting from a maximum Raikoke sulfate AOT of 0.025 (in August) and assuming a decay time of 80±10 days as discussed in Sect. 3.

We made an alternative attempt to estimate the Raikoke volcanic aerosol contribution to the overall aerosol optical properties. We calculated the contributions of smoke and sulfate particles to the overall backscatter and extinction coefficients for different

sulfate aerosol fractions. In these calculations, we considered a typical sulfate lidar ratio of 40 sr at both wavelengths (see Table 2), a lidar ratio of 60sr and 100 sr at 355 and 532 nm, respectively, for the smoke particles, and typical extinction-related Ångström exponents of 1.5 for sulfate aerosol and 0.75 for smoke aerosol (in the 355-532 nm wavelength range). For these realistic conditions, we had to assume a Raikoke aerosol fraction of 15% (in terms of 532 nm extinction coefficient) and a smoke fraction of 85% in order to reproduce the measured mean MOSAiC aerosol lidar ratios of 55 sr and 85 sr at 355 and

532 nm, respectively. However, smoke lidar ratios of 60 sr (355 nm) and 100 sr (532 nm) represent already extremely different values. If we assume a smoke lidar ratio pair of 60 sr and 90 sr, we can reproduce the measured aerosol lidar ratio values of about 55 sr and 85 sr only with a Raikoke aerosol fraction of <10% and a respective smoke fraction of >90%.

## 4.2    The smoke layer from August 2019 to May 2020

In Fig. 13 to 15, we present an overview of our MOSAiC smoke layer observations aboard the drifting RV Polarstern in terms

of geometrical and optical properties. One set of lidar products per day is considered. Gaps in the data time series are caused by fog and low-cloud events, partly lasting over many days. We included KARL observations at Spitsbergen in Fig. 13b and c to prolong the AOT and extinction time series and to better link the strong Siberian fires (July-August 2019) discussed in Sect. 3 with the UTLS aerosol observed during the MOSAiC expedition. According to the KARL observations, the stratospheric AOT reached values close to 0.15 at 532 nm in the beginning of August 2019 and decreased to values of 0.07–0.08 in September

2019.

Figure 13a shows the temporal evolution of the geometrical properties of the High Arctic aerosol layer during the winter half year. The UTLS aerosol layer extended, on average, from 7-8 km to 17-18 km height. The layer base was frequently found below the tropopause. The vertical bars in Fig. 13a are colored to distinguish different levels of the aerosol loading expressed in terms of the particle extinction coefficient. The backscatter coefficients at 532 nm were multiplied with a lidar ratio of

85 sr (mean value of the entire MOSAiC period) to obtain the extinction coefficients. Thus, we ignore the minor impact of the Raikoke sulfate aerosol at heights >12-13 km on the AOT retrieval. For this height range, a sulfate lidar ratio of 45 sr would be more appropriate to convert backscatter into extinction coefficients. But the use of 85 sr instead of 45 sr has only a minor impact on the results shown in Fig. 13. As can be seen in Fig. 13a, the maximum light-extinction values were typically found just above the tropopause. They slowly decreased with time from values >10 $Mm^{-1}$ in October and November to <5 $Mm^{-1}$

in April 2020. Figure 13a also contains information about the occurrence of PSCs. Most of the PSCs over Polarstern were detected in January 2020. We observed a much lower number of PSCs over the North Pole region (86° to 88.6°N) during the winter and spring seasons of 2020 than the CALIPSO lidar within the latitudinal range from 60° to 81.8°N.

Figure 13b provides an overview of the development of the aerosol optical thickness (AOT) at 355 and 532 nm from August 2019 to May 2020. We computed the AOT from the particle backscatter height profiles in order to reduce the noise in the lidar AOT observations significantly and thus to better see trends in the evolution of the polar smoke layer. The directly determined extinction profiles were too noisy, especially in 2020. Therefore, the 355 nm and 532 nm backscatter coefficients were multiplied with MOSAiC mean lidar ratios of 55 sr at 355 nm and 85 sr at 532 nm. Subsequently, we integrated the extinction values between the aerosol layer base and top heights as given in Fig. 13a to obtain the AOT. We corrected our stratospheric smoke observations in Fig. 13 for clearly identified PSC effects. Nevertheless, weak PSC effects remained in the optical data for January and February as was mentioned in Sect. 2. The remaining PSC impact on the 532 nm AOT values was estimated to be of the order of 5% or less.

The combined KARL and Polly observations show a coherent downward trend in the AOT time series until the beginning of December 2019. The 532 nm AOT observed over the Polarstern decreased from 0.05–0.12 in October and November to values of 0.03-0.06 from December to mid of March, and then dropped to 0.01-0.02 in April 2020. Almost constant AOT conditions were observed from 10 December to 10 March. Based on the KARL and Polly observations, we can conclude that the UTLS perturbation decreased from about 0.15 (532 nm AOT) in the beginning of August 2019 to 0.02 at the end of April 2020 (within 9 months), thus, the e-folding decay time was about 5 months. A potential 10-15% Raikoke volcanic impact (mainly from heights >12 km) is then reflected in a sulfate-related AOT of about 0.005 at 532 nm during the December to February winter months.

The layer mean 532 nm smoke extinction coefficients in Fig. 13c (obtained from AOT divided by the respective layer geometrical depth in Fig. 13a) were of the order of 10 Mm$^{-1}$ until mid of November 2019, and around 4-5 Mm$^{-1}$ during the main winter months until mid of March 2020 and mostly ≤3 Mm$^{-1}$ at the end of the life time of the smoke layer. According to long term observations at mid-latitude lidar sites, the minimum 532 nm AOT value for a clean stratosphere is around 0.001-0.002 (Sakai et al., 2016; Baars et al., 2019) and the minimum extinction coefficients are of the order of 0.1-0.2 Mm$^{-1}$. From the measured layer mean extinction coefficients, mass concentrations of the smoke particles were derived and ranged from 0.4-2 $\mu$g m$^{-3}$ during the autumn and winter months. Minimum stratospheric mass concentrations (at mid latitudes) are close to 0.01-0.02 $\mu$g m$^{-3}$ (Baars et al., 2019). The shown surface area concentrations of 0.05-0.2 cm$^2$ m$^{-3}$ or 5-20 $\mu$m$^2$ cm$^{-3}$ are in same range as the ones for typical PSCs (Jumelet et al., 2008, 2009) as mentioned.

We finally estimated the smoke aerosol load over the Arctic by multiplying the area (considering latitudes >66.7°N) by a mean smoke layer depth of 8 km, a mean smoke extinction coefficient of about 5 Mm$^{-1}$, a volume-to-extinction conversion factor of 0.124 × 10$^{-12}$ Mm for wildfire smoke (Ansmann et al., 2021a), and the smoke particle density of 1.15 g m$^{-3}$, and yield 0.2 Mt of smoke as a guess for the mean value of the smoke aerosol load over the Arctic during the winter half year 2019-2020. The overall Siberian smoke particle mass injected into the UTLS height range of the northern hemisphere may have been a factor of 2 higher. For comparison, the particle mass injected into heights of 11-13 km during the record-breaking

Canadian pyroCb smoke event in August 2017 was about 0.3 Mt (Yu et al., 2019) and the rather strong Australian bushfires in December 2019 and January 2020 caused a stratospheric smoke particle mass of the order of 0.5-1 Mt (Peterson et al., 2021).

In Fig. 14, we compare our seasonal mean extinction profiles calculated from the MOSAiC profile data sets with CALIPSO long-term observations of Arctic aerosol profiles (Yang et al., 2021) as well as retrievals of the particle extinction coefficient from satellite-based OMPS-LP (Ozone Mapping Profiler Suite Limb Profiler) measurements at 675 nm and Stratospheric Aerosol and Gas Experiment (SAGE) II and III observations (Treffeisen et al., 2006; Taha et al., 2021). In this way, a consistent, vertically resolved view on the aerosol condition in the Central Arctic during the winter half year up to 30 km is provided, to our knowledge, for the first time.

Yang et al. (2021) analyzed Arctic CALIPSO observations for the latitudinal belt from 65°-81.8°N for the time period from June 2006 to December 2019 and made use of the latest data analysis version 4 with an improved aerosol/cloud discrimination scheme. The shown seasonal mean height profiles are in good agreement with our measurement. Since these observations include observations of the UTLS smoke from the major Canadian fires in 2017 (Baars et al., 2019) and the huge fires in Siberia in the summer of 2019, they are close to our 2019-2020 profile observations. The 13-year mean autumn profile, shown in Fig. 14, suggests a strong and regular contributions from wildfires to the polar aerosol load in the middle troposphere in late summer and autumn.

The CALIPSO lidar observations of Yang et al. (2021) agree well with the results of Di Biagio et al. (2018). These authors combined ground-based lidar observations between 80°-83°N and 7°-27°E (north of Svalbard) from October 2014 to June 2015 with CALIPSO observations from 80°-81.8°N and 5°-25°E to obtain an improved knowledge of the wintertime aerosol conditions in the high Central Arctic. The ground-based lidars were operated at 800 nm and mounted on autonomous drifting buoys (IAOOS: Ice-Atmosphere-Ocean Observing System platforms) and made, for the first time, aerosol observations at latitudes up to 83°N during the winter season.

Aerosol profiles derived from CALIPSO lidar observations are not retrieved at heights larger than 12 km. At greater heights the signal-to-noise ratio is too low to permit a proper aerosol retrieval for a trustworthy multiyear statistics. A discussion on the sensitivity of the CALIPSO lidar in the case of Arctic aerosol is given by Di Biagio et al. (2018).

Treffeisen et al. (2006) analyzed SAGE II and III data for 525 nm and were able to present, for the first time, an annual cycle of aerosol vertical layering from 4-12 km for 60°-80°N. Data collected from 2001-2006 were used in Fig. 14. The SAGE II and III measurements are obviously representative for typical background aerosol conditions. Taha et al. (2021) analyzed OMPS-LP data and SAGEIII/ISS (International Space Station) observations and provide aerosol information for the height levels of 18.5, 20.5, and 25.5 km height (2017-2019) for 60°N and 70°N for the wavelength of 745 nm. The 60° and 70°N values were used to estimate the values at 80°N, shown in Fig. 14, via extrapolation. Taha et al. (2021) used an Ångström exponent of 1.9 to convert the satellite aerosol observations at 745 nm to the ones at 532 nm. If we define the aerosol profile from 4-12 km of Treffeisen et al. (2006) in combination with the curve of Taha et al. (2021) and the interpolated dashed line in Fig. 14 as the smoke-free and volcanic-aerosol-free background aerosol level, then the particle extinction coefficient was enhanced in the UTLS regime from 8-15 km height by an order of magnitude in the winter half year of 2019-2020.

Figure 15 presents the time series of the layer mean intensive particle properties. The information in Fig. 15 is dominated by the lidar observation from 6-12 km height and thus by the impact of smoke around 10 km height. In contrast to the extensive properties in Fig. 13b and c, the depolarization ratio, lidar ratio, and backscatter-related Ångström exponents show no dependence on time. The aerosol properties remained almost unchanged over the entire winter half year. Particle coagulation or

5 significant removal processes, influencing the size distribution and thus the Ångström exponent, are not visible. The increasing variability in the depolarization ratios starting in January 2020 and a weak trend in the Ångström exponent for the 532-1064 nm wavelength range towards lower values are related to a non-perfect removal of weak PSC effects on the lidar signal profiles. In addition, the uncertainty in the lidar products increased because of the decreasing stratospheric perturbation. Table 3 contains the respective mean values of the intensive aerosol properties. Again, a clear smoke signature is visible in Fig. 15 and

10 Table 3, expressed by the inverse wavelength dependence of the lidar ratio and the rather different Ångström exponents for backscattering and extinction.

Figure 16 and Table 4 provide information on the underlying microphysical properties of the Arctic smoke and summarize the main optical and microphysical particle characteristics discussed above for the two days (25 October and 7 November 2019) and for another observation taken on 13 October 2019. The volume size distributions shown in Fig. 16 and the results

in Table 4 were obtained from the Polly observation between 8 and 12 km height by applying the lidar inversion method to the vertical mean of three backscatter and two extinction coefficients as described in Sect. 2 (Veselovskii et al., 2002). For comparison, we included smoke size distribution for aged Australian smoke, observed after long-range transport over Punta Arenas in southern Chile, and for Canadian smoke, observed after long-range transport from western Canada to central Europe (Leipzig, Germany). All size distributions are computed with the same data analysis software of Veselovskii et al. (2002), and

afterwards normalized so that the integral over each shown size distribution is one.

As typical for smoke layers, a well-defined accumulation mode was found. A distinct coarse mode was absent. The findings agree well with in-situ observations of long-transported aged smoke (Fiebig et al., 2003; Petzold et al., 2007; Dahlkötter et al., 2014). The differences between the size distribution observed over the North Pole region, Leipzig, and Punta Arenas are most probably related to the different atmospheric residence times and, correspondingly, to available time periods for aging,

coagulation and removal processes. There are many other reasons such as fire type and burning material that have an impact on the emitted smoke size distribution.

Table 4 summarizes the optical and microphysical smoke particle properties for the three MOSAiC cases shown in Fig. 16. As already visible in the figure, the small mean and effective radii indicate quite small smoke particles over the North Pole region. The effective diameter can be regarded as a typical, characteristic size of the smoke particles. The Australian and

30 Candian smoke particles were much larger and showed effective radii of 0.32 $\mu$m and 0.27 $\mu$m.

Mass and surface-area concentrations indicate a moderately polluted stratosphere, 3-4 months after the injection. In the case of the Canadian smoke we observed stratospheric layer mean extinction values of the order of 1 Mm$^{-1}$ at 532 nm and mass concentrations around 0.1 $\mu$g m$^{-3}$ three months after injection in August 2017. Surface area concentrations around 15 $\mu$m$^2$ cm$^{-3}$ in Table 4 are comparable with surface areas provided by PSCs.

The values for the refractive index (real part $n_{real}$, imaginary part $n_{imag}$) and the single scattering albedo SSA in Table 4 of the polar smoke are in good agreement with respective findings of Dubovik et al. (2002) based on extended sunphotometer observations of North American wildfire smoke ($n_{real}$=1.5±0.4, $n_{imag}$=0.0094±0.003, and SSA=0.94±0.2). However, our refractive index and SSA values are highly uncertain and must therefore be interpreted with care. Wandinger et al. (2002) showed cases of Canadian wildfire smoke, measured in August 1998, for which the SSA was 0.8 and $n_{imag}$ around 0.05. An SSA of 0.8 was also observed in the case of the Canadian smoke over Leipzig in August 2017. The articles of Müller et al. (2005) (Canadian and Siberian wildfires) and Tesche et al. (2011) (agricultural fires in central western Africa) provide an overview of the large spread of possible values for SSA and the imaginary part of the refractive index of smoke. The values can range from 0.63 to 0.98 (SSA) and from 0.001 to 0.07 ($n_{imag}$) according to these two papers. Correspondingly large differences can be found in terms of the lidar ratio (about 30 to 110 sr for 532 nm). These large ranges of values reflect that smoke can show rather different properties in terms of chemical composition, black carbon fraction, and particle size depending on fire type (flaming vs smoldering), burning material, and environmental conditions during the aging process shortly after emission and during long-range transport in the troposphere or in the stratosphere. When discussing polar smoke properties one needs to keep in mind that these observations are hard to compare with smoke properties at other places around the globe. The smoke (observed from October to May) circulated around the North Pole in total darkness at very low temperatures for months, and it is simply unknown in which way the smoke chemical and physcial properties change with time and how they influence the optical properties of the aged smoke.

The final figure of this section, Fig. 17, shows a PSC observation performed on 15 January 2020. As mentioned, most of the PSCs over Polarstern were detected in January 2020. According to the PSC classification scheme of Achtert and Tesche (2014), we observed a type Ib PSC in Fig. 17. This type is made up of supercooled liquid ternary solutions that consist of $H_2SO_4$, $HNO_3$, and $H_2O$. As shown in Fig. 17, PSCs were most frequently found in the upper part of or above the smoke layer. As mentioned, we corrected our stratospheric smoke observations in Fig. 13 for clearly identified PSC effects. But weak PSC effects remained in the optical data for January and February as was mentioned in Sect. 2 and is visible in Fig. 15a and c by slightly enhanced depolarization ratios and decreased Ångström exponents as discussed below. The remaining PSC impact on the AOT values was estimated to be of the order of 5% or less..

## 5 Smoke, sulfate aerosol, PSCs, and the record-breaking ozone hole: Any impact of the UTLS aerosol on ozone depletion?

Two facts motivated us to briefly discuss a potential impact of the **UTLS aerosol** on ozone depletion. Firstly, a record-breaking ozone depletion was observed in the spring of 2020 (DeLand et al., 2020; Manney et al., 2020; Wohltmann et al., 2020; Inness et al., 2020; Wilka et al., 2021), and secondly, at the same time a strong perturbation of the stratospheric aerosol conditions occurred. That a potential relationship between the **the high UTLS pollution levels** and strong ozone reduction is not considered in any of the studies on ozone depletion mentioned above can be regarded as the third motivating aspect.

It is well known that strong ozone reduction is linked to the development of a strong and long-lasting polar vortex, which favors increased PSC formation. In these clouds, active chlorine components are produced via heterogeneous chemical processes on the surface of the PSC particles. Finally, the chlorine species destroy ozone molecules in the spring season. It is also known that volcanic sulfate aerosol particles can serve as sites for heterogeneous chemical reaction and can lead to an increased

release of active chlorine components and thus contribute to an enhanced ozone reduction (Solomon, 1999; Zhu et al., 2015, 2018). We observed the impact of the Pinatubo aerosol on ozone depletion at midlatitudes (Leipzig and Lindenberg, Germany) in the winters of 1991-1992 and 1992-1993 and quantified the reduction as a function of the surface area concentration of the volcanic sulfate particles (Ansmann et al., 1996). **We found an ozone loss of up to 30% at sulfate particle surface area concentrations of 25-35 $\mu$m$^2$ cm$^{-3}$ in the central part of the volcanic sulfate layer during the first winter after the major**

**volcanic eruption.**

Thus, it seems to be reasonable to assume that also the observed smoke and volcanic particles over the High Arctic had an influence on the observed strong ozone reduction in the late winter and early spring months of 2019-2020. The difference to sulfate particles is that smoke particles in the stratosphere are most likely glassy, show a core-shell morphology, and are largely composed of organic material (organic carbon, OC, in the shell) and, to a minor part, of black carbon (BC, concentrated in

the core part). It is also possible that collisions and coagulation of smoke and volcanic aerosol particles partly led to internally mixed particles. All this complicates studies on the impact of the smoke-dominated Arctic aerosol in the stratosphere on ozone depletion occurring during the MOSAiC expedition.

As was pointed out by Zhu et al. (2015, 2018), there are two pathways to influence ozone depletion by aerosol pollution. The particles can influence the evolution of PSCs and specifically their microphysical properties (number concentration, size

distribution) (Hoyle et al., 2013; Zhu et al., 2015, 2018), and, on the other hand, the particles can be directly involved in heterogeneous chemical processes by increasing the particle surface area available to convert nonreactive chlorine components into reactive forms. **A third (indirect) impact of smoke, when well distributed over large parts of the northern or southern hemisphere, is via the influence on large-scale atmospheric dynamics (Hirsch and Koren, 2021).**

The goal of this section is just to compile all information we have regarding PSC occurrence, **smoke and sulfate condi-**

**tions**, and ozone depletion during the MOSAiC campaign and to provide an extended view on the Arctic ozone conditions in the winter half year of 2019-2020 based on CALIPSO PSC observations, our MOSAiC aerosol and PSC observations, and MOSAiC ozone profiles measured with sondes launched from Polarstern. These data may serve as a stimulating guide for modeling teams to clarify the role of smoke in the complex ozone depletion processes.

Figure 8a showed the polar vortex characteristics for the winter season 2019-2020. The strong, cold, and persistent polar vor-

tex controlled the atmospheric conditions above 15 km height from January to April-May 2020. The MOSAiC and CALIPSO measurements are shown in Fig. 18. 40 ozone sondes were launched during the seven-month period from October 2019 to May 2020 (von der Gathen and Maturilli, 2020; Wohltmann et al., 2020). 13 out of the 40 sondes were launched from the beginning of March to mid of April 2020 and thus during the main period with lowest ozone concentration.

We analyzed the CALIPSO observations in the latitudinal belt from 60°N to 80°N on a daily basis. The pink lines in Fig. 18a

indicate the height range in which PSCs were detected. The top of the PSC height range was always easy to identify in the

CALIPSO observations. The base height must be exercised with care because the lowermost PSCs may have produced too weak backscatter and were then not clearly detectable in the noisy CALIPSO data. According to the MOSAiC radiosondes launched four times a day aboard Polarstern, the lowest temperatures occurred between 15 km and 27 km height in the central winter (December 2019 to March 2020). The PSC relevant temperatures of $< -78°C$ were found between 18 km and 27 km

height in December 2019 and continuously propagated downward to about 15-23 km height in early March 2020. This height range of low temperatures coincides well with the PSC height range in Fig. 18a, and also with the respective PSC retrievals presented by DeLand et al. (2020).

In addition to the PSC height range, the time series of the **UTLS aerosol** layer base and top heights as well as of the tropopause height are shown in Fig. 18a. As can be seen in the composite figure, a layer with very low ozone partial pressure

between 15 and 20 km height was observed above the Polarstern in March and April 2020 (until the Polar vortex began to collapse around 20 April). This layer of low ozone concentration coincides with the PSC height range in which chlorine activation occurred in the months before. The UTLS aerosol layer, extending roughly from the tropopause to 15–18 km height did not overlap with the region with very low ozone concentration in the spring of 2020, and also not with the PSC height range until mid of January 2020.

To obtain a more detailed picture on ozone depletion during the winter and spring season 2019-2020, Fig. 18b presents ozone deviations from the long-term mean values as discussed by Inness et al. (2020) together with PSC and smoke layer information. Inness et al. (2020) used a reanalysis dataset produced by the Copernicus Atmosphere Monitoring service (CAMS, reanalysis, 2003–2019) to describe the evolution of the 2020 Arctic ozone season and to compare it with years back to 2003. There is a clear signature of chemical ozone depletion leading to the extremely low ozone values over the North Pole in March and April

2020. In March 2020, ozone values in the ozone layer over the North Pole were partly reduced to more than 10 mPa below the climatological values. We notice a clear link between PSC occurrence and anomalously large ozone reduction. But we see also a large vertical overlap between the UTLS aerosol layer and the height range with strong negative ozone deviations **at heights as low as 9-10 km.** This was **also confirmed** by Manney et al. (2020). The height range with significant ozone anomalies reached down to unusually low heights and thus to heights where smoke and, to a minor part, volcanic sulfate particles, were

permanently present. Surface area concentrations of the smoke particles were of the order of 5-15 $\mu m^2$ cm$^{-3}$ at 10-12 km height from January to April 2020 and thus in the same range of values for Arctic PSCs (as shown in Fig. 13). In April 2020 (not shown), PSC were no longer observed, however, the UTLS aerosol layer was still present. All in all, Fig. 18 may motivate model-based studies on the interplay between smoke, sulfate particles, PSCs, and ozone depletion. A first fruitful approach was presented by Zhu et al. (2018) with focus on the impact of volcanic sulfate aerosol.

**To provide some numbers regarding the potential aerosol impact on the observed ozone depletion via heterogeneous chemical processes on and in the particles, we compare our Pinatubo observations in central Europe in the winter of 1991-1992 (Ansmann et al., 1996) with the MOSAiC ozone and aerosol measurements in March 2020 and our strato­spheric smoke observations over Punta Arenas, southern Chile (Ohneiser et al., 2020) at 53°S in September 2020 when the record-breaking ozone hole over Antarctica developed. In the first winter after the major Pinatubo volcanic erup­**

**tions, the surface area concentration of the sulfate particles was as high as 20-35 $\mu m^2$ cm$^{-3}$ in the height range from**

**15-20 km height and the ozone reduction (compared to the climatological mean) was of the order of 15-30%. In March 2020, we measured smoke particle surface-area concentrations of 5-10 $\mu m^2$ cm$^{-3}$ in 10-12 km height over the North Pole region. The ozone loss was 20-25% in this height range according to Fig. 18b, taken from Inness et al. (2020). Finally, after the strong Australian bushfires in December 2019 and January 2020 (Peterson et al., 2021; Yu et al., 2021), we measured smoke surface area concentrations of the order of 1-5 $\mu m^2$ cm$^{-3}$ over Punta Arenas in the height range from 14-22 km with large ozone depletion in September 2020 (9 months after injection), and according to preliminary simulations (assuming that sulfate and smoke particles show similar chlorine-activation efficiencies) the aerosol-induced ozone loss was about 5% (Yu et al., 2021). These numbers indicate that additional aerosol in the usually clean stratosphere is able to lead to an additional ozone loss of the order of 5-10% (and more) so that events of strong ozone reduction may become record-breaking events.**

## 6 Summary and outlook

We presented a detailed optical and microphysical characterization of an unexpected UTLS smoke layer over the North Pole region in the winter half year of 2019-2020. To the best of our knowledge, such a strong perturbation of the stratospheric aerosol conditions in the High Arctic was never reported before. We hypothesized that the detected smoke originated from strong long-lasting wildfires in Siberia in July and August 2019. Importantly, a month earlier, the Raikoke volcano erupted and the resulting stratospheric sulfuric-acid aerosol layers also covered large parts of the northern hemisphere. However, using lidar measurements during our field campaign and modeling efforts presented in the literature, we showed that the volcanic aerosol could not explain the observed strong perturbation of the stratospheric aerosol layer in the High Arctic with AOTs of the order of 0.1. The Raikoke-related AOT fraction at 532 nm was estimated to be always lower than 15%. In particular, our analyses suggest that self-lifting effects (in the absence of pyroCb convection) caused the spread of smoke up to tropopause heights above Siberia.

We indirectly emphasized (without stating that explicitly) the need for a multiwavelength polarization Raman lidar, such as the Polarstern Polly operated during the MOSAiC expedition, to unambiguously identify the prevailing aerosol type based on the spectral dependence of the lidar ratio. The observed extinction-to-backscatter ratios (lidar ratios) were, on average, 55 sr at the wavelength of 355 nm and of 85 sr at 532 nm as typical for light-absorbing smoke. The extinction-related 355-532 nm Ångström exponent of around 0.65 also clearly indicated that smoke particles dominated. We were able to develop a coherent picture on aerosol structures and layering features for the autumn and winter seasons up to 30 km height. In the next step, we will analyze the MOSAiC lidar observations of the summer half year to fully cover the annual cycle of Arctic aerosol conditions as a function of height.

In this article, we also discussed the potential impact of the wildfire smoke **and sulfate aerosol on the record-breaking ozone depletion over the Arctic in the spring of 2020 based on vertically resolved information on PSC and smoke and sulfate aerosol occurrence and strength of ozone depletion. The preliminary discussions may stimulate in-depth modeling studies to clarify the role of the UTLS aerosol in stratospheric PSC formation and ozone reduction processes.**

**In the case of the strong Australian bushfires, we observed a very clear coincidence of the Australian smoke layer and the layer with record-breaking ozone destruction over the southern parts of South America.** If follow-on studies will indicate a link between huge fires (caused by unusually hot temperatures and droughts as a result of climate change), corresponding smoke occurrence in the lower stratosphere, and severe ozone depletion in the Arctic and Antarctica, the climate change debate will be added by a new, and until now, not considered important aspect.

As an outlook, we will explore the potential of wildfire smoke to influence cirrus formation during the winter half year. A first case study was discussed in Engelmann et al. (2020). Furthermore, we will contrast these results with ones of similar studies of aerosol-cirrus interaction during the summer half year when long-range transport of anthropogenic haze mixed with mineral dust from Asia, Europe, and North America as well as episodic wildfire smoke events prevailed.

# 7 Data availability

Polly lidar observations (level 0 data, measured signals) are in the PollyNET data base (PollyNet, 2021) with quicklooks at http://polly.tropos.de. All the analysis products are available at TROPOS upon request (polly@tropos.de) and at https://doi.pangaea.de/10.1594/PANGAEA.935539 (Ohneiser et al., 2021). KARL lidar results can be provided by AWI upon request. CALIPSO observations were downloaded from the CALIPSO data base (CALIOP, 2021), Fire and MODIS data are available at the NASA data base (FIRMS, 2021; MODIS, 2021). The ozonesonde data can be found by using the link in von der Gathen and Maturilli (2020). The radiosonde data are available at https://doi.org/10.1594/PANGAEA.928656 (Maturilli et al., 2021).

# 8 Author contributions

The paper was designed and written by KO, AA, and RE. The data analysis was performed by KO, RE, CR, AC, IV, HB, and HG. The co-authors RE, HGr, MR, JH, and DA took care of the lidar observations aboard Polarstern during the MOSAiC year. SD and MM were responsible for radiosonde and ozonesonde measurements.

# 9 Competing interests

The authors declare that they have no conflict of interest.

# 10 Financial support

The data was produced as part of the international Multidisciplinary drifting Observatory for the Study of the Arctic Climate (MOSAiC) with the tag MOSAiC20192020 and Project ID AWI_PS122_00. This project has also received funding from the European Union's Horizon 2020 research and innovation program ACTRIS-2 Integrating Activities (H2020-INFRAIA-2014 - 2015, grant agreement no. 654109). We gratefully acknowledge the funding by the Deutsche Forschungsgemeinschaft (DFG, German Research Foundation) – project no. 268020496 - TRR 172, within the Transregional Collaborative Research

Center "ArctiC Amplification: Climate Relevant Atmospheric and SurfaCe Processes, and Feedback Mechanisms (AC)3". The development of the lidar inversion algorithm used to analyze Polly data was supported by the Russian Science Foundation (project no. 16-17-10241).

*Acknowledgements.* We are grateful to the MOSAiC teams and the RV Polarstern crew for their perfect logistical support. We further thank
5    the entire radiosonde and ozonesonde team, especially AWI, DWD, ARM, Jürgen (Egon) Graeser, and all volunteers for their enormous efforts of producing the exemplary and uninterrupted MOSAiC dataset. We thank the editor and the three reviewers for a very fruitful discussion.

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

**Table 1.** Overview of main lidar products (smoke optical and microphyscial properties), typical vertical window lengths, $\Delta z$ (effective vertical resolution), used in signal profile smoothing (backscatter, depolarization ratio) and regression analysis (extinction) and typical time periods, $\Delta t$, of signal averaging (effective temporal resolution). Typical uncertainties in the optical properties at 355 and 532 nm and the derived microphyscial properties are given in the right column. Uncertainties in the 1064 nm backscatter coefficient and respective Ångström exponents (using 1064 nm backscatter) are around 20% and 25%, respectively. Microphysical properties can be obtained by means of the conversion method (from backscatter profiles, $\Delta z$=450 m) or by applying the inversion-with-regularization technique to the set of three backscatter and two extinction profiles ($\Delta z$=2000 m).

| Parameter | $\Delta z$ | $\Delta t$ | Uncertainty |
|---|---|---|---|
| Backscatter coefficient | 450 m | $\geq$30 min | $\leq$10% |
| Extinction coefficient | 2000 m | $\geq$2 h | 20% |
| Lidar ratio | 2000 m | $\geq$2 h | 25% |
| Ångström exponent | 450 m | $\geq$30 min | 15% |
| Depolarization ratio | 450 m | $\geq$30 min | $\leq$10% |
| Number concentration | 450 m, 2000 m | $\geq$30 min, $\geq$2 h | 30% |
| Volume concentration | 450 m, 2000 m | $\geq$30 min, $\geq$2 h | 25% |
| Mass concentration | 450 m, 2000 m | $\geq$30 min, $\geq$2 h | 30% |
| Surface-area concentration | 450 m, 2000 m | $\geq$30 min, $\geq$2 h | 25% |

**Table 2.** Typical lidar ratio pairs ($L_{355}$, $L_{532}$ in sr, measured at 355 and 532 nm) in aged tropospheric (T) and stratospheric (S) smoke layers. Lidar ratio pairs for further aerosol types are given for comparison. All measurements are performed with multiwavelength polarization Raman lidars. The particle depolarization ratio $\delta_{532}$ at 532 nm is given in addition and is usually <0.05 for spherical particles.

| Aerosol type | $L_{355}$ | $L_{532}$ | $\delta_{532}$ | Reference |
|---|---|---|---|---|
| Siberian smoke (T) | 40 | 65 | 0.06 | Murayama et al. (2004) |
| Canadian smoke (T) | 45 | 68 | ≤0.05 | Haarig et al. (2018) |
| Canadian smoke (T) | 34 | 50 | ≤0.05 | Müller et al. (2005) |
| Canadian smoke (S) | 40 | 72 | 0.18 | Haarig et al. (2018) |
| Australian smoke (S) | 71 | 97 | 0.18 | Ohneiser et al. (2020) |
| Wildfire smoke (S) | 55 | 85 | ≤0.05 | Polly, MOSAiC, High Arctic |
| Wildfire smoke (S) | 75 | 105 | ≤0.05 | Polly, Leipzig, August 2019 |
| Volcanic sulfate aerosol (S) | 40 | 40 | ≤0.05 | Mattis et al. (2010) |
| Volcanic ash (T) | 60 | 60 | ≤0.35 | Ansmann et al. (2010) |
| North American haze | 47 | 37 | ≤0.05 | Müller et al. (2005) |
| European haze | 58 | 53 | ≤0.05 | Mattis et al. (2004) |
| Arctic haze (T) | 60 | 60 | ≤0.05 | Müller et al. (2004) |
| Middle East desert dust (T) | 43 | 39 | >0.3 | Hofer et al. (2020) |
| Western Saharan dust (T) | 53 | 56 | >0.27 | Groß et al. (2015) |

**Table 3.** Mean values and standard deviations of smoke optical properties computed from the time series in Fig. 15. Up to 151 daily observations from the beginning of October 2019 to mid of March 2020 are considered. $\delta_{\mathrm{p}}$ denotes the particle linear depolarization ratio PLDR. In the case of the lidar ratio $L$ at 355 and 532 nm, we considered only high-quality observations (showing low noise impact, 46 days for $L_{355}$, 36 days for $L_{532}$). The Ångström exponents for the lidar ratio ($_{\mathrm{L}}$) and the extinction coefficient ($_{\sigma}$) are calculated from the values of $L_{355}$ and $L_{532}$ in Fig. 15 and the values of $\sigma_{355}$ and $\sigma_{532}$ in Fig. 13c.

| Parameter | Mean ± SD |
|---|---|
| $\delta_{\mathrm{p},355}$ | 0.02±0.009 |
| $\delta_{\mathrm{p},532}$ | 0.015±0.005 |
| $L_{355}$ [sr] | 54.5±5.5 sr |
| $L_{532}$ [sr] | 85.3±10.4 sr |
| $A_{\beta,355,532}$ | 1.70±0.70 |
| $A_{\beta,532,1064}$ | 1.58±0.36 |
| $A_{\beta,355,1064}$ | 1.62±0.32 |
| $A_{\sigma,355,532}$ | 0.63 |
| $A_{\mathrm{L},355,532}$ | -1.07 |

**Table 4.** Optical and microphysical properties of the polar wildfire smoke layer in the autumn of 2019. Layer mean values of the particle backscatter coefficient $\beta$, extinction coefficient $\sigma$, lidar ratio $L$, and backscatter, extinction and lidar-ratio-related Ångström exponents $A_\beta$, $A_\sigma$, and $A_L$, respectively, are given in the upper part. Indices indicate wavelength in nm and wavelength spectrum. The lower block contains the retrieved particle number concentration (particles with radius >50 nm), mean and effective particle radius $r_{mean}$ and $r_{eff}$, volume ($V$), mass ($m$), and surface area ($s$) concentration, real ($n_{real}$) and imaginary part ($n_{imag}$) of the refractive index, and single scattering albedo SSA. Simlar SSA values were obtained for all three wavelengths. Uncertainties are given in Table 1 and are $\pm 0.1$ for $n_{real}$, within an order of magnitude for $n_{imag}$, and of the order of $\pm 0.05$ for SSA.

| Parameter | 13 Oct 2019 | 25 Oct 2019 | 7 Nov 2019 |
|---|---|---|---|
| Height, base to top [km] | 4.5–15 | 5.5–16 | 5.5–14 |
| $\beta_{355}$ [Mm$^{-1}$ sr$^{-1}$] | 0.250 | 0.233 | 0.250 |
| $\beta_{532}$ [Mm$^{-1}$ sr$^{-1}$] | 0.116 | 0.117 | 0.124 |
| $\beta_{1064}$ [Mm$^{-1}$ sr$^{-1}$] | 0.037 | 0.039 | 0.039 |
| $\sigma_{355}$ [Mm$^{-1}$] | 13.8 | 11.2 | 11.3 |
| $\sigma_{532}$ [Mm$^{-1}$] | 10.4 | 8.2 | 8.7 |
| $L_{355}$ [sr] | 55 | 48 | 45 |
| $L_{532}$ [sr] | 90 | 70 | 70 |
| $A_{\beta,355,532}$ | 1.88 | 1.71 | 1.72 |
| $A_{\beta,532,1064}$ | 1.66 | 1.59 | 1.68 |
| $A_{\beta,355,1064}$ | 1.74 | 1.63 | 1.69 |
| $A_{\sigma,355,532}$ | 0.68 | 0.78 | 0.63 |
| $A_{L,355,532}$ | -1.22 | -0.93 | -1.09 |
| $N$ ($r$ >50 nm) [cm$^{-3}$] | 42 | 55 | 42 |
| $r_{mean}$ [$\mu$m] | 0.18 | 0.15 | 0.17 |
| $r_{eff}$ [$\mu$m] | 0.22 | 0.20 | 0.22 |
| $V$ [$\mu$m$^3$ cm$^{-3}$] | 1.2 | 0.98 | 1.0 |
| $m$ [$\mu$g m$^{-3}$] | 1.38 | 1.13 | 1.15 |
| $s$ [$\mu$m$^2$ cm$^{-3}$] | 16.0 | 14.8 | 14.0 |
| $n_{real}$ | 1.49 | 1.50 | 1.50 |
| $n_{imag}$ | 0.010 | 0.007 | 0.007 |
| SSA, 532 nm | 0.956 | 0.967 | 0.969 |

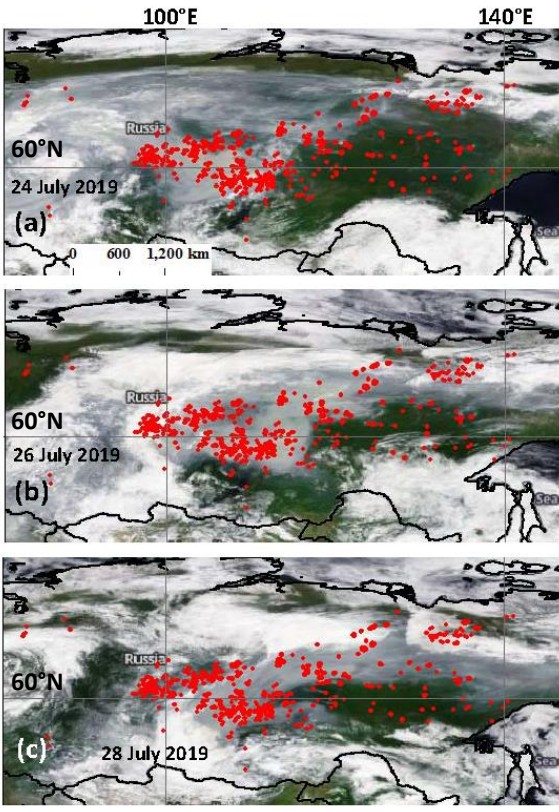

**Figure 1.** Extended fields of heavy wildfires and large areas covered by smoke (grey areas) north of Lake Baikal (53.5°N, 108°E), Russia, in central and eastern Siberia (MODIS, 2021), (a) MODIS overpass on 24 July 2019, (b) on 26 July 2019, and (c) on 28 July 2019. White areas indicate cloud layers. Fires detected at the three days are given as red dots (FIRMS, 2021).

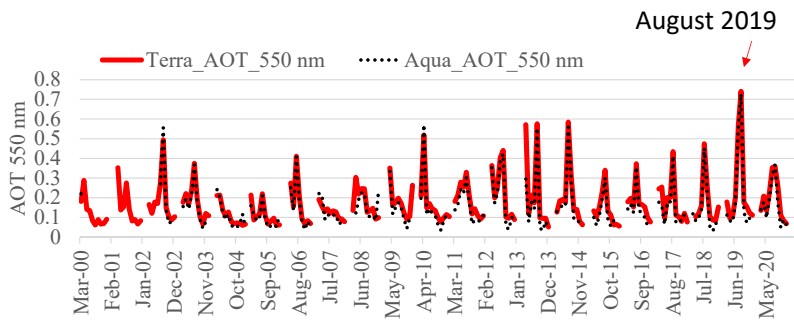

**Figure 2.** Monthly and regionally mean 550 nm AOT measured with MODIS aboard Aqua and Terra (MODIS, 2021) in central and eastern Siberia from March 2000 to December 2020. The considered region is defined by the latitudes from 55°–72°N and longitudes from 100°–138°, north of Lake Baikal (53.5°N, 108°E).

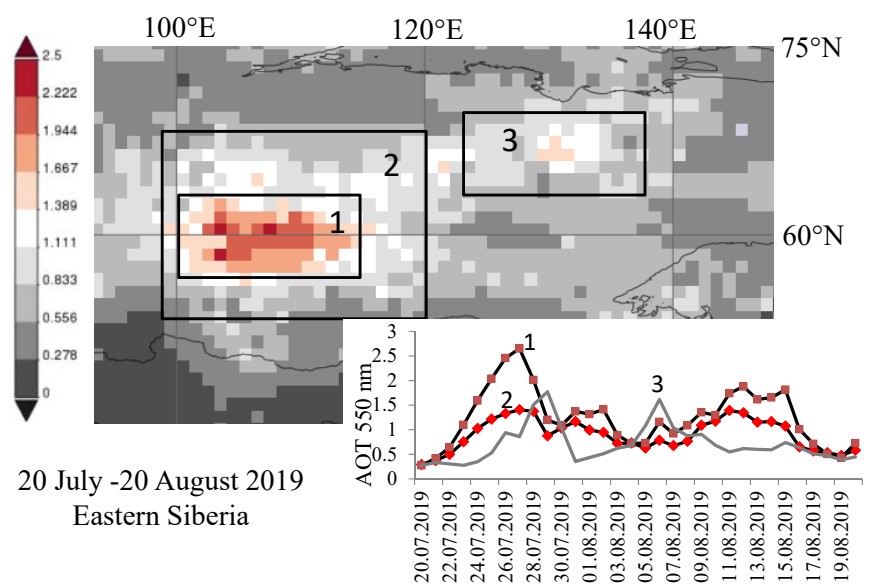

**Figure 3.** Monthly mean AOT (550 nm, 20 July - 20 August 2019) north (boxes 1 and 2) and northeast (box 3) of Lake Baikal (53.5°N, 108°E), Russia, and time series of daily mean AOT (mean AOT of boxes 1, 2, and 3) (MODIS, 2021).

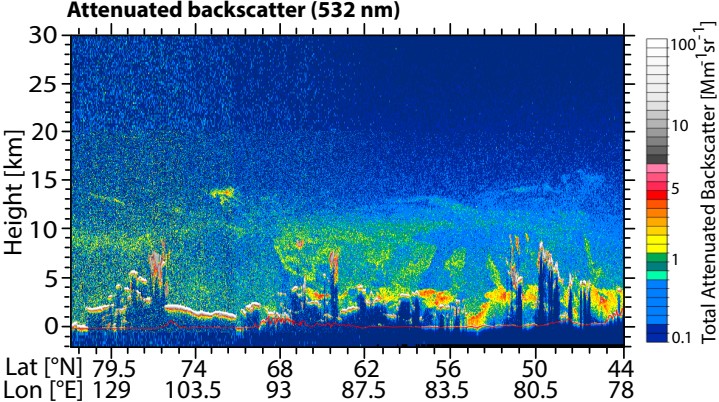

**Figure 4.** CALIPSO lidar measurement (height–latitude/longitude display of 532 nm attenuated aerosol backscatter) of wildfire smoke over central and eastern Siberia on 26 July 2019, 21 UTC, during an overpass west of box 2 defined in Fig. 3, and downwind of large fires according to Fig. 5. Thick smoke plumes (red, green and yellow) reaching 10 km height (tropopause level) were observed west of box 2 (55°–68°N). Bluish layers above 10 km height are very likely sulfate particle layers originating from the Raikoke volcanic eruption.

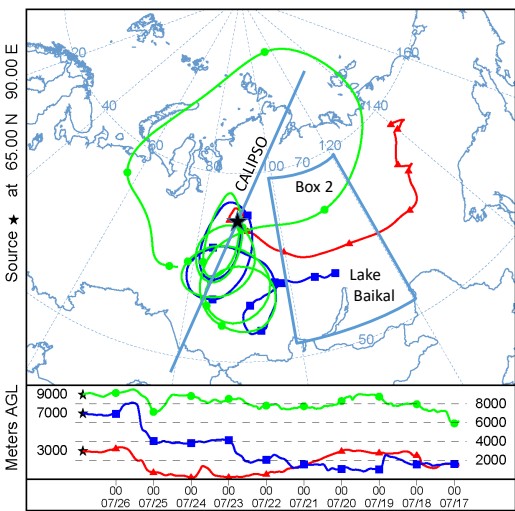

**Figure 5.** HYSPLIT 10-day backward trajectories (HYSPLIT, 2021; Stein et al., 2015; Rolph et al., 2017) for 26 July 2019, 21:00 UTC, arriving at 3000 m (red), 7000 m (blue) and 9000 m height (green) above the location indicated by a star (65°N, 90°E). The CALIPSO flight track of the measurements shown in Fig. 4 is indicated by a straight line, west (downwind) of a large fire area (box 2 in Fig 3).

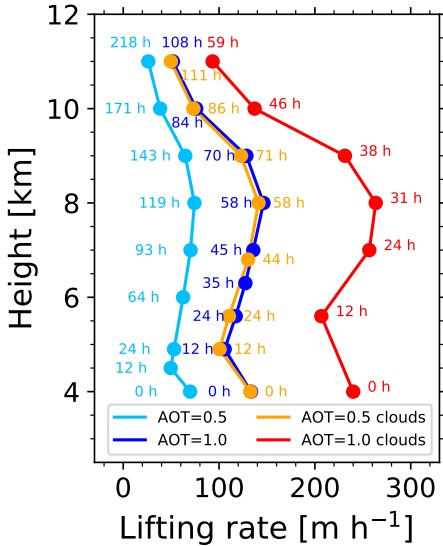

**Figure 6.** Simulation of self-lifting of Siberian wildfire smoke (injected to 3-5 km height). The 532 nm AOT of the 2-km thick smoke layer, initially centered at 4 km height, was assumed to be 0.5 and 1.0. Cloudfree (blue curves) and cloudy conditions (orange and red curves, with a cloud deck below the smoke layer, totally reflecting incoming solar radiation) are simulated. Smoke is able to reach heights close the tropopause at 10 km within about 2-6 days (indicated by the hours after ascent start) according to these simulations.

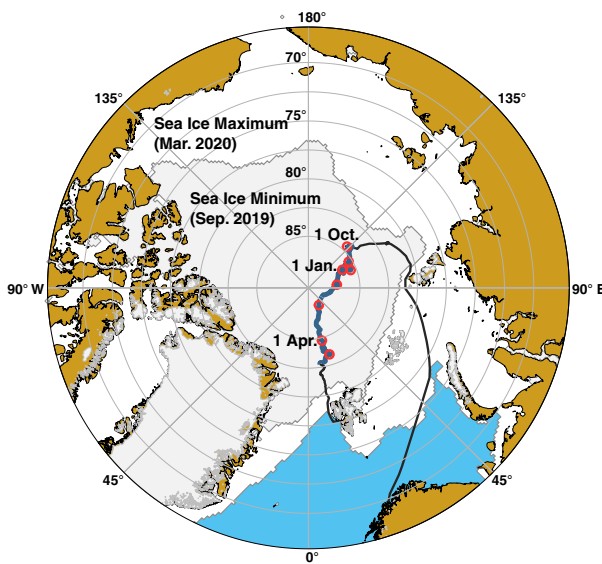

**Figure 7.** Travel (black) and drifting (blue) route of RV Polarstern from 1 October 2019 to 1 May 2020. The beginning of the next month is indicated by a red circle. The map was produced with 'ggOceanMaps' (Vihtakari, 2021) by using Sea Ice Index Version 3 data (Fetterer et al., 2020).

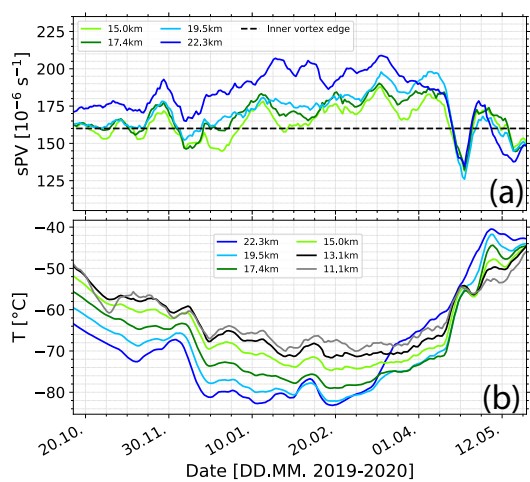

**Figure 8.** (a) Scaled potential vorticity $sPV$ and (b) temperature $T$ along the Polarstern drift route (Fig. 7) for up to six height levels indicated by different colors. $sPV$ values exceeding $160 \times 10^{-6}$ s$^{-1}$ (vortex edge, thick horizontal line in a) for heights $>17$ km indicate that Polarstern was below the strong vortex from mid December 2019 to mid April 2020. The polar vortex began to collapse around 20 April (deep $sPV$ minimum in a). A downward trend in temperature is visible in (b) until the end of February 2020. Favorable conditions for PSC formation were given at heights $>17$ km from 20 December 2019 to 10 March 2020.

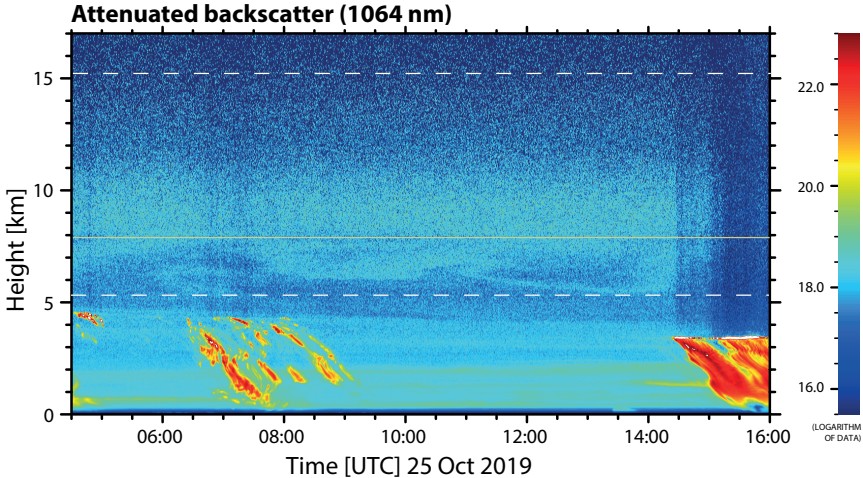

**Figure 9.** Wildfire smoke over RV Polarstern at 85.4°N, 128.0°E on 25 October 2019, 4:30–16:30 UTC. The tropopause height according to the Polarstern radiosonde (launched at 12:00 UTC) is given as an orange solid line and the base and top height of the smoke layer are indicated by white dashed lines. Further haze layers and embedded cirrus clouds (virga in yellow to red) are visible at heights below 5 km. The uncalibrated attenuated backscatter coefficient at 1064 nm (in arbitrary units) is shown.

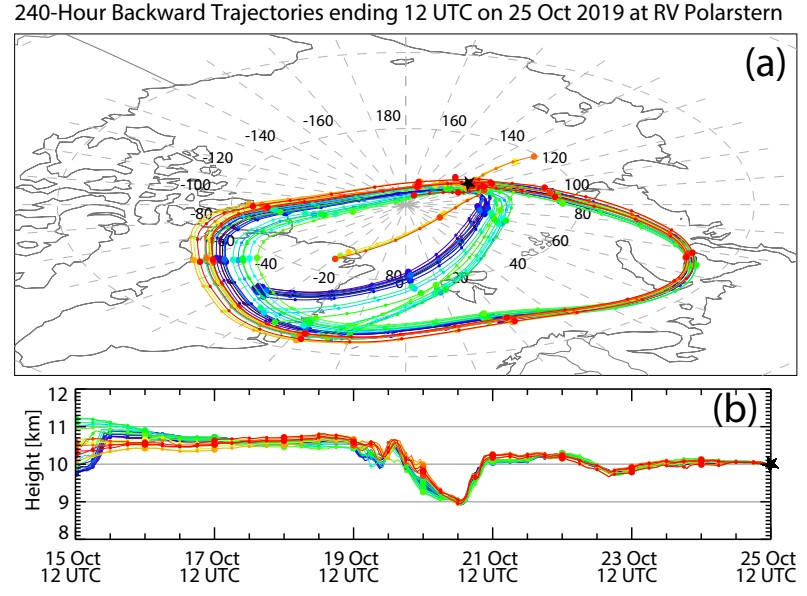

**Figure 10.** HYSPLIT 10-d ensemble backward trajectories arriving at 10 km height above RV Polarstern (black star) on 25 October 2019, 1200 UTC (HYSPLIT, 2021). Thin and thick symbols indicate 6-hour and 24-hour time steps, respectively. Colors are used to better identify different subgroups of trajectories.

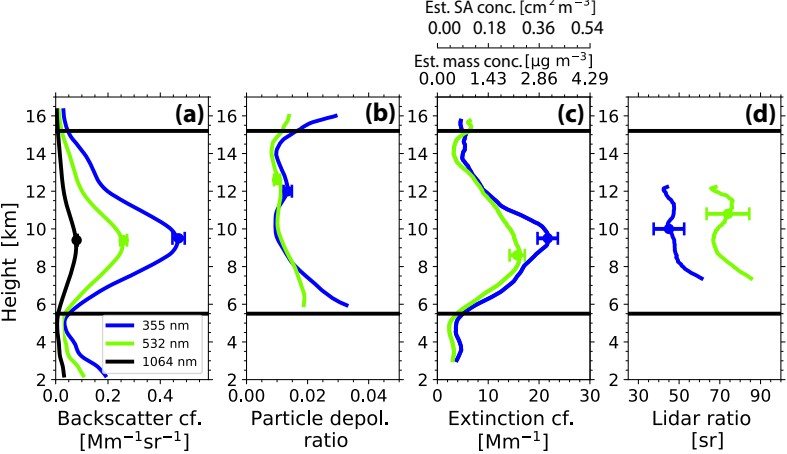

**Figure 11.** Profiles of optical properties (4.5 h mean values) of the wildfire smoke layer on 25 October 2019, 09:30–14:00 UTC (cirrus-free period, Fig. 9). Base and top heights of the smoke layer are indicated by black horizontal lines. (a) Particle backscatter coefficient at three wavelengths, (b) particle linear depolarization ratio at 355 and 532 nm, (c) smoke extinction coefficient at 355 and 532 nm, and d) respective smoke extinction-to-backscatter ratio (lidar ratio) are shown. Vertical resolution is about 450 m (backscatter, depolarization ratio), 2000 m (extinction coefficient), and 2400 m (lidar ratio). Estimated mass concentration and surface area concentration (obtained from conversion of the 532 nm extinction coefficients) are given in addition (upper x-axis in c). Error bars (one standard deviation) indicate the estimated uncertainties in the observations around layer center. Depolarization ratios and extinction coefficients close to and around the layer boundaries have to be interpreted with caution.

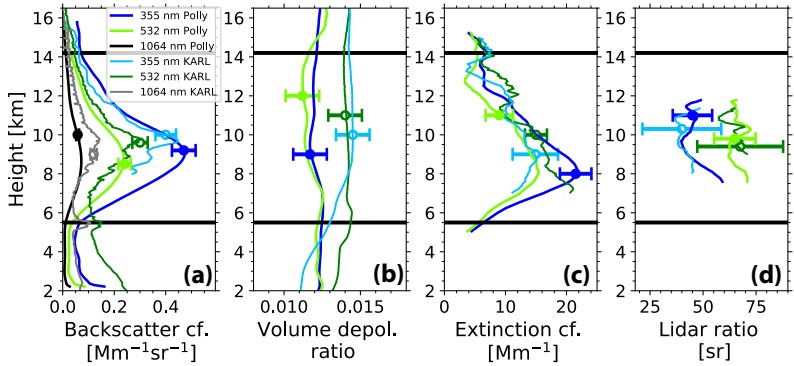

**Figure 12.** Same as in Fig. 11, except for 7 November 2019 (20:25-23:35 UTC, 85.9°N, 116.8°E). KARL (AWI lidar at Ny-Ålesund, Svalbard, Norway, 78.9°N, 11.9°E) observations from 4 Nov 2019 are shown for comparison (open symbols, thin lines). 2400 m vertical signal smoothing is applied in the case of the KARL measurements. Note that (b) shows the volume depolarization ratio instead of the particle linear depolarization ratio as in Fig. 11. Good agreement between the different observation was found. Tropopause over Polarstern was at 7.6 km height.

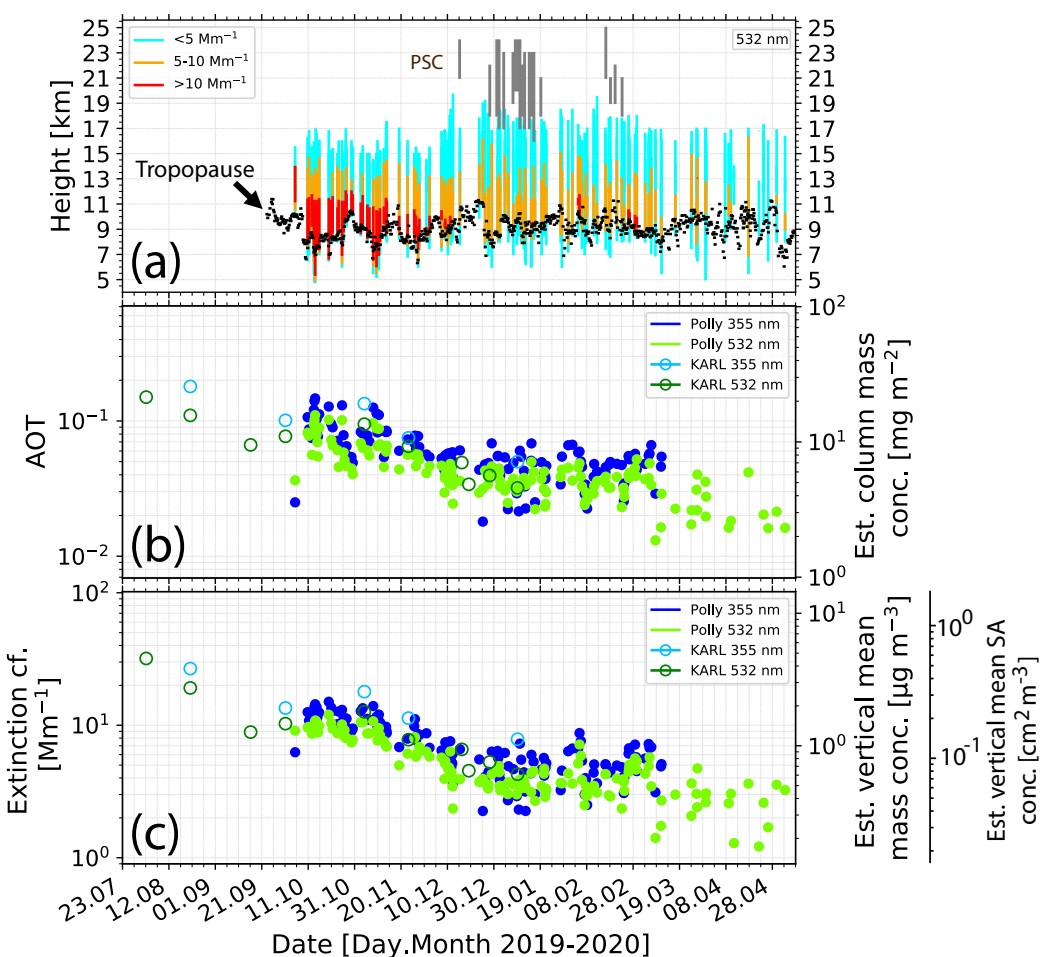

**Figure 13.** (a) Overview of Polly observations of UTLS smoke layers (colored bars from bottom to top, one bar per day) from 23 July 2019 to 8 May 2020. Observational gaps between bars are caused by opaque low level clouds and fog. The colors in each bar indicate segments with different extinction coefficient levels ($< 5\,\mathrm{Mm^{-1}}$, $5-10\,\mathrm{Mm^{-1}}$, and $> 10\,\mathrm{Mm^{-1}}$, see legend in the panel). Furthermore, the tropopause is indicated as small black bars, and PSCs layers are shown as gray vertical lines. (b) Smoke layer AOT (KARL, open symbols, Polly, closed symbols) at 355 nm and 532 nm, calculated from the profiles of the backscatter coefficients multiplied by a lidar ratio of 55 sr and 85 sr, respectively. Column mass concentrations are indicated as well (right y-axis). (c) Layer mean 355 and 532 nm particle extinction coefficient (i.e., AOT in panel b divided by layer depth in panel a), and respective estimated mass and surface area concentrations (right y-axis). For comparison, background AOT and extinction levels (532 nm) are of the order of 0.001–0.002 and 0.1–0.2 $\mathrm{Mm^{-1}}$, respectively (Baars et al., 2019).

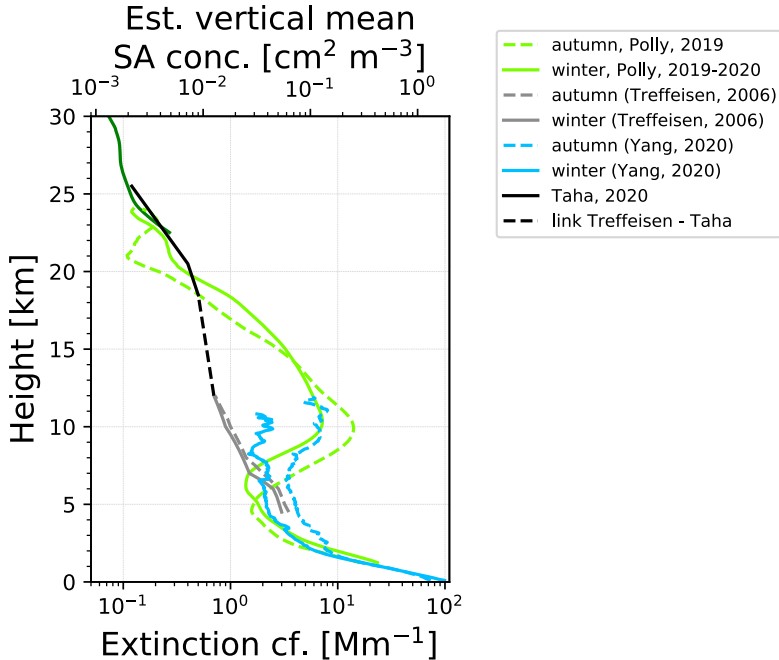

**Figure 14.** Seasonal mean particle extinction coefficient profiles (532 nm, autumn: October and November, winter: December to February), measured with the Polarstern lidar up to 30 km height. In addition, extinction profile observations (500-550 nm range) found in the literature (see list to the right of the figure) are given for comparison and to indicate aerosol background conditions during undisturbed times (black and gray profile segments). The dark green extinction curve from 22.5 to 30 km (MOSAiC autumn and winter mean value) is computed from the 532 nm signal profile by applying the Klett method in order to indicate the background aerosol level above 25 km height. The surface area concentration (upper y-axis) is related to the MOSAiC smoke extinction profiles. More details are given in the text.

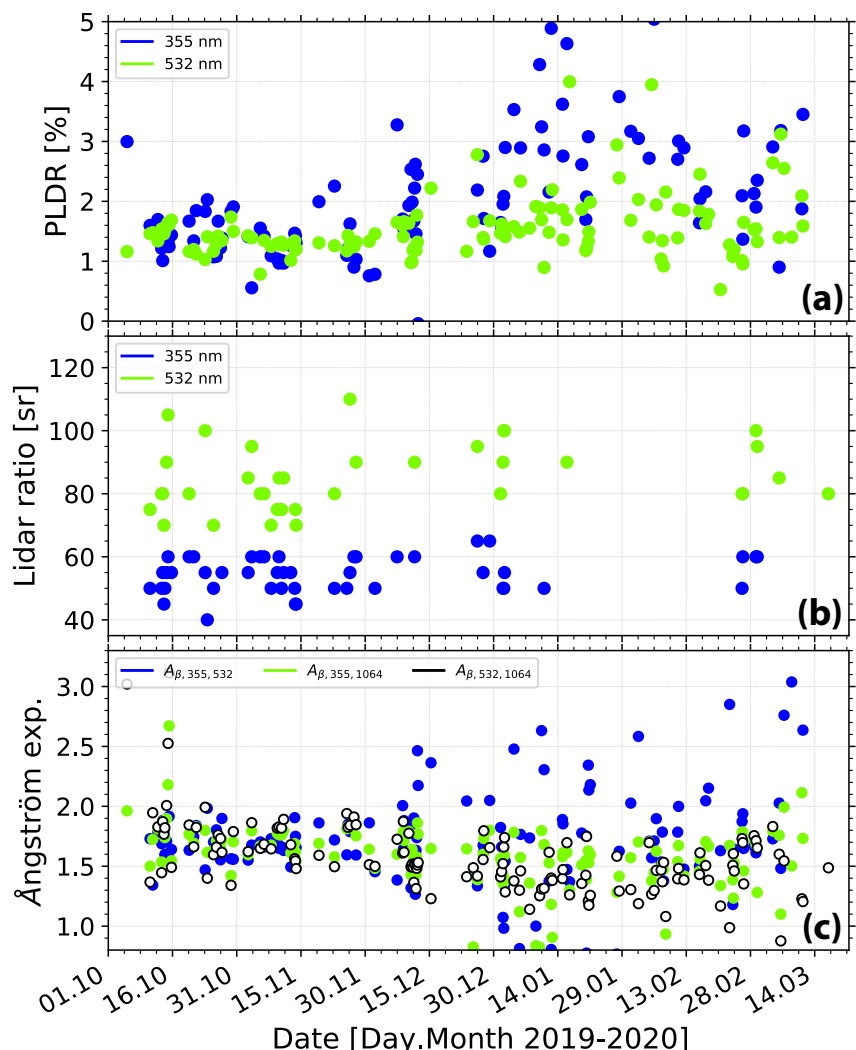

**Figure 15.** (a) Overview of smoke optical properties in terms of (a) particle linear depolarization ratio (PLDR), (b) extinction-to-backscatter ratio (lidar ratio), and (c) backscatter-related Ångström exponent. All results are derived from Polly observations within the smoke layer. The considered time period spans from 1 October 2019 to 22 March 2020. The scatter in the data is caused by signal noise and atmospheric variability, and also weakly by a residual effect after PSC impact correction (depolarization ratio, Ångström exponent, January to March 2020).

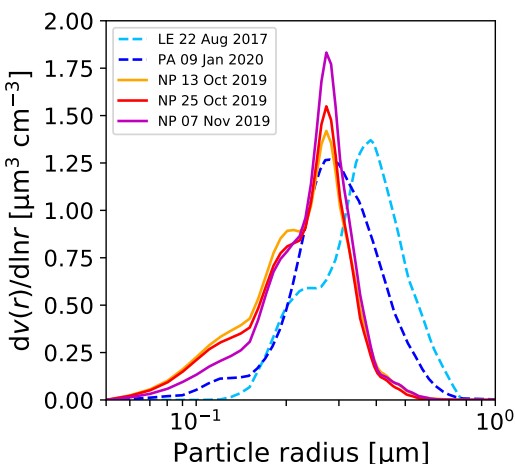

**Figure 16.** Volume size distributions of the stratospheric smoke particles retrieved from the multiwavelength lidar observations on 13 and 25 October and 7 November 2019. A narrow accumulation mode with particle sizes (diameters) from 400 to 1000 nm and a weak Aitken mode to the left is typical for aged wildfire smoke particles. For comparison, the size distribution of Australian wildfire smoke particles measured over Punta Arenas, Chile (PA, dark blue), and of Canadian smoke measured over Leipzig, Germany (LE, light blue), are presented as well. All size distributions are normalized so that the integral over each distribution is one.

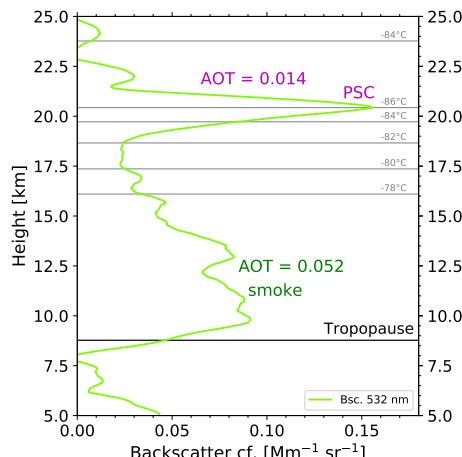

**Figure 17.** Polar stratospheric cloud (PSC) from 18-22.5 km height on top of the smoke layer on 15 January 2020, 22:30-23:30 UTC. The 532 nm particle backscatter coefficient is shown and the AOT values for the smoke (computed from the backscatter values multiplied by a lidar ratio of 85 sr) and of the PSC layer (computed from the backscatter values multiplied by a lidar ratio of 50 sr) are given as numbers. Horizontal gray lines show different temperature levels. Tropopause was at 8.8 km height.

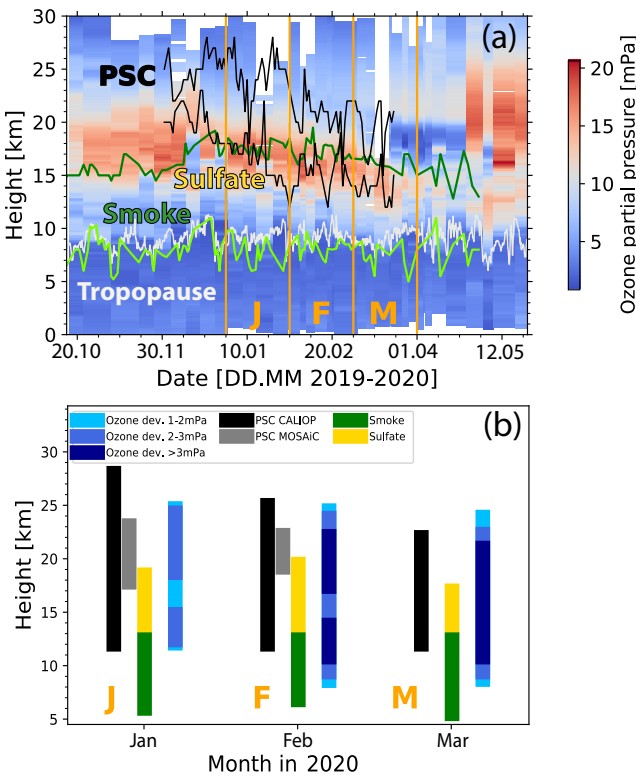

**Figure 18.** (a) Height-time display of the ozone partial pressure observed with ozonesondes launched at Polarstern. The PSC height range (according to daily CALIPSO lidar observations between 60° and 80°N) is indicated by black lines. The UTLS smoke and sulfate aerosol layer from base (green) to top (dark green) as observed with the MOSAiC lidar over Polarstern (at >85°N) is indicated as well. The white line shows the tropopause. (b) Height range with negative ozone deviation of 1-2 mPa (light blue), 2-3 mPa (blue) and >3 mPa (dark blue) from the long-term climatological monthly mean (2003–2019) at 90°N (as given in Fig. 6c1-c4 in Inness et al. (2020)). The ozone deviations are shown as mean values for January (J), February (F) and March (M) also indicated in (a). The black and gray vertical bars indicate the height ranges in which PSCs were detected with the CALIPSO lidar (as in a) and with the Polarstern lidar (at >85°N), respectively. **The combined yellow-green bars indicate the UTLS aerosol layer with the main layer containing wildfire smoke and a volcanic sulfate layer on top.**