# Peer review of "The unexpected smoke layer in the High-Arctic winter stratosphere during MOSAiC 2019-2020"

_Atmospheric Chemistry and Physics, 2021_

## Community Comment (CC1)

Ohneiser et al. (2021), hereafter abbreviated as "O21," is a provocative study of upper troposphere, lower stratosphere (UTLS) aerosols at high northern latitude between fall 2019 and spring 2020.  It is provocative in that O21 observe UTLS aerosol nearly daily in that temporal span and conclude that the composition is wildfire smoke, the source is fires in a sector of Siberia in July 2019, and the transport pathway to the UTLS is diabatic heating/lofting. The manuscript is also provocative in that Raikoke volcano (Kuril Islands) erupted in late June 2019 and polluted the UTLS with a mass of $SO_2$ (https://twitter.com/simoncarn/status/

1142713198480482304) on par with or exceeding other eruptions that generated stratospheric clouds persisting for greater than a half year (E.g. Kasatochi, Sarychev Peak, Nabro, Kelut, Calbuco. See Solomon et al. (2016) for a tabulation.).

If O21's conclusions are borne out, it will be a new insight into the polar UTLS and smoke transport to the UTLS.  However, there is overwhelming observational evidence that Arctic UTLS in the second half of 2019 and early months of 2020 was blanketed by Raikoke sulfates. Secondly, there is abundant evidence that the UTLS aerosol picture O21 describe over the July Siberia source sector is also dominated by Raikoke $SO_2$ and sulfates. Thirdly, the lidar data presented by O21 are more closely aligned with spherical sulfate droplets than smoke particles. These three points are elaborated on below.

Point 1

Kloss et al. (2021) show that the Raikoke volcanic cloud dominated the high-latitude northern hemisphere from eruption through the early months of 2020. I consulted Chris Boone, ACE-FTS Project Scientist and co-Principal Investigator, to help qualify the 2019/20 UTLS plume further. ACE not only delivers aerosol extinction profiles but profiles of $SO_2$ as well. These were combined by Cameron et al. (2021) in an examination of several UTLS volcanic events including Raikoke; the results further qualify those of Kloss et al. and show the strong presence of Raikoke $SO_2$ and sulfates at high northern latitude in summer and fall 2019. In addition, ACE IR spectra have been used to identify smoke aerosol in connection with ACE Imager extinction profiles (Boone et al. (2020)).  The same technique was applied to high-latitude northern hemisphere 2019/20 ACE data while invoking published sulfate IR spectra for comparison. The findings are summarized as follows. In July 2019, the lower stratosphere in the latitude region near 60 degrees north is stuffed with sulfate aerosols from the Raikoke eruption.  Identification of the composition is accomplished by looking at the infrared spectrum of the aerosols and noting the coincident enhancement of $SO_2$ and ACE Imager aerosol extinction layers. In September/October 2019, there is a blanket of aerosols in the lower stratosphere in the latitude region near 80 degrees north.  The blanket appears to be composed of Raikoke sulfate aerosols. In February/March 2020, the aerosol blanket in the lower stratosphere near 80°N is still present. $SO_2$ has decayed.  However, the spectra associated with the Imager aerosol layers

are consistent with sulfate. At no point did we find any evidence of biomass burning smoke playing a role in these stratospheric aerosols (Boone et al., 2020). Detailed support for these findings is available upon request.

Point 2

O21 hypothesize that their MOSAiC Arctic 2019/20 lidar signals are dominated by an impressive build-up of UTLS smoke that began between ~24-28 July 2019 in a zone within Siberia centered roughly at 60$^{o}$N, 110$^{o}$E (Figure 2). They present nadir satellite imagery combining fire hot-spot data, true-color imagery, and aerosol optical depth retrievals to show an intensification of burning and smoke concentration. They also display a single CALIOP curtain on 26 July to characterize the vertical structure and ascent of smoke layers (Figure 3). There are several concerns regarding Figures 2, 3 and their interpretation, listed below.

It is simply not possible to know from a single CALIOP curtain if ascent is taking place; there is insufficient information from such a quasi-instantaneous vertical snapshot. Additional information must be brought to bear.

There are technical issues with Figure 3 and its caption. The date of the CALIOP image is given as "2016." This is obviously an inconsequential typographical error. But more substantially, the latitude, longitude coordinates labeled along the bottom are wrong. The curtain displayed is actually situated west of the Figure 2 boxed focal point. It is west of the labeled coordinates by approximately 25$^{o}$ longitude. Compare Figure 3 with the actual orbital track and lidar data here:

https://www-calipso.larc.nasa.gov/products/lidar/browse_images/show_v4_detail.php?s=production&v=V4-10&browse_date=2019-07-26&orbit_time=21-04-20&page=1&granule_name=CAL_LID_L1-Standard-V4-10.2019-07-26T21-04-20ZN.hdf

https://www-calipso.larc.nasa.gov/products/lidar/browse_images/show_v4_detail.php?s=production&v=V4-10&browse_date=2019-07-26&orbit_time=20-12-10&page=4&granule_name=CAL_LID_L1-Standard-V4-10.2019-07-26T20-12-10ZD.hdf

This is scientifically relevant for two reasons. One is that the vertical aerosol profile over the Siberia focus zone is unknown to the reader. Secondly, the aerosol profile on display is nominally upwind of the Siberia box, meaning that the history of those aerosols may be disconnected with processes occurring in the Siberia box.

Indeed, it appears from the CALIOP curtains linked above that the "Ascending smoke plumes (mostly in green)
are visible up to the tropopause at 10-11 km height as well as in the lower stratosphere…"
[Quoted from Figure 3 caption] are mostly assigned as sulfate by the CALIOP version 4 aerosol subtype algorithm.

One can see from CALIOP curtains just upwind of the Figure 2 Siberia box, on days leading up to the O21 smoke AOD buildup, an assignment of sulfate subtype to "mostly in green" backscatter in the UTLS. Here is an example from 20 July:
https://www-calipso.larc.nasa.gov/products/lidar/browse_images/show_v4_detail.php?s=production&v=V4-10&browse_date=2019-07-20&orbit_time=20-33-51&page=1&granule_name=CAL_LID_L1-Standard-V4-10.2019-07-20T20-33-51ZN.hdf
https://www-calipso.larc.nasa.gov/products/lidar/browse_images/show_v4_detail.php?s=production&v=V4-10&browse_date=2019-07-20&orbit_time=19-41-41&page=4&granule_name=CAL_LID_L1-Standard-V4-10.2019-07-20T19-41-41ZD.hdf

Similar scenes are found each day thereafter leading up to the 24-28 July O21 period of focus (https://www-calipso.larc.nasa.gov/products/lidar/browse_images/std_v4_index.php?d=2019)
This multi-day, broad, high-latitude swath of aerosols primarily defined as "sulfate/other" conforms to the findings of Cameron et al. and our deeper investigation into ACE July 2019 profiles near 60°N.

It stands to reason then, that if Raikoke sulfates are blanketing high northern latitudes at that time (Kloss et al., Cameron et al., and our investigation), they would also be evident over the Figure 2 Siberia box before, during, and after the hypothesized smoke uplift. The CALIOP example below shows aerosol subtype findings consistent with the prior examples on 20 July over the Figure 2 box:

https://www-calipso.larc.nasa.gov/products/lidar/browse_images/show_v4_detail.php?s=production&v=V4-10&browse_date=2019-07-20&orbit_time=18-55-21&page=1&granule_name=CAL_LID_L1-Standard-V4-10.2019-07-20T18-55-21ZN.hdf
https://www-calipso.larc.nasa.gov/products/lidar/browse_images/show_v4_detail.php?s=production&v=V4-10&browse_date=2019-07-20&orbit_time=18-03-11&page=4&granule_name=CAL_LID_L1-Standard-V4-10.2019-07-20T18-03-11ZD.hdf

Hence, if the July 2019 Siberia-zone smoke is the initial condition for the O21 lidar observations in fall and winter, the presentation surrounding Figure 2 and 3 is insufficient to make that case.

Point 3

O21 acknowledge that the stratospheric aerosols detected by MOSAiC lidars were spherical according to aerosol depolarization measurements (Page 7, line 21). In principle, there seems to be less uncertainty as to the shape, composition, and depolarization of volcanic liquid sulfates in contrast to biomass burning smoke. Lidar measurements of tropospheric and stratospheric smoke converge on the idea that smoke is depolarizing (e.g. Burton et al., 2015; Fromm et al., 2008), suggesting some amount of asphericity. The first reported lidar observations of stratospheric smoke emphasized the unmistakable signal of depolarization (Siebert et al., 2000; Fromm et al., 2000). More recent stratospheric smoke papers, cited within O21, consistently report depolarization by smoke particles. Hence O21 would be establishing a new finding--stratospheric smoke with essentially no aerosol depolarization. Arguing for this peculiarity, O21 mention an aging process and a collapse of black carbon core. Neither process is described, and no publications are cited. Given the weight of evidence for the overarching presence of Raikoke UTLS liquid sulfates during the O21 reporting period and the MOSAiC depolarization results provided therein, the arguments for particle aging and black carbon core-shell collapse must be made more substantively.

References

Boone et al. (2020), https://doi.org/10.1029/2020GL088442

Burton et al. (2015), https://doi.org/10.5194/acp-15-13453-2015

Cameron et al. (2021), https://doi.org/10.1016/j.jqsrt.2020.107341

Fromm et al. (2000), https://doi.org/10.1029/1999GL011200

Fromm et al. (2008), doi:10.1029/2007JD009147

Kloss et al. (2021), https://doi.org/10.5194/acp-21-535-2021

Siebert at al. (2000), https://doi.org/10.1007/s00585-000-0505-0

Solomon et al. (2016), https://doi.org/10.1126/science.aae006

---

## Author Comment (AC1)

Dear Reviewer!

First of all thank you for careful reading and all good and constructive suggestions which improved the paper significantly (we hope). Before we answer all comments and questions, step by step and item by item, let us provide a brief overview of the essential changes:

We went deeply into the literature with special focus on volcanic sulfate aerosol (originating from the recent Raikoke and from the Sarychev volcanic eruptions), into our own Leipzig lidar data analysis, into the CALIPSO observations, we performed an extended HYSLPIT trajectory analysis and conducted further simulation studies to bring together much more solid information and substantial argumentation to provide a plausible link between the Siberian fires (in July-August 2019) and the polar smoke layer we observed later on in the North Pole region (since late September 2019). The focus is on the potential self-lifting process, on smoke aging and even on the consequences for the morphological and optical properties of smoke (such as the depolarization ratio).

We did all this very carefully and even if we present a large number of reasonable and plausible arguments for our chain of hypotheses we emphasize that our entire set of arguments still remains at a hypothetical level.

Nevertheless, there is no doubt! What we observed during the MOSAiC is clearly and unambiguously wildfire smoke. There is no way around. We observed this inverse spectral dependence of the lidar ratio (LR355 < LR532), and that is a unique fingerprint of wildfire smoke, and we observe this since 1998, in Canadian, Siberian, Australian smoke, … again and again.

And one of the convincing and motivating arguments to search for a much more massive aerosol source than the Raikoke volcano was: The Raikoke-related AOD at high northern latitudes was about 0.01 at 500 nm in autumn 2019 (according to simulations of the 2019 Raikoke as well as of the 2009 Sarychev volcanic aerosol impact and corroborated also by Raikoke aerosol observations at lower latitudes, discussed in Kloss et al., 2021). But, we observed AODs around 0.1 in autumn 2019 in the High Arctic, and thus an order of magnitude higher values. We were forced to find a massive aerosol source.

Some significant changes before we start the step by step reply:
- We show a new figure (Fig. 2, time series, MODIS AOD from 2000-2020), highlighting the extraordinarily strong 2019 fire season in central and eastern Siberia.
- We introduce a new table (Table 2), highlighting the unique signature (fingerprint) of aged wildfire smoke, namely the inverse spectral slope of the lidar ratio (LR355nm < LR532nm) together with high LR532. We compare these smoke features in the new table with the ones for mineral dust, volcanic sulfate and ash, Arctic, European, American haze, etc.
- We show a new HYSPLIT plot (Fig. 5) to better explain the link between the CALIPSO overflight over Siberia and the strong fires… downwind (west!) of the flight track.
- We show a new figure with the Polarstern track (Fig. 6, same as in Engelmann et al., 2021).
- We created a new figure (Fig. 7) to better show the influence of the polar vortex on all the observations.
- We introduced a new section (Section 3) on Siberian fires, self-lifting hypothesis, consequences for aging (leading to spherical smoke particles), and also on the potential miss-classification of smoke layers as sulfate layers when using the CALIPSO aerosol typing scheme (Sect. 3.1).
- We shortened the ozone-depletion/smoke/PSC section as requested, and combined the two figures of this section to one figure (Sect. 5, Fig. 17).

- We skipped several figures (layer-depth histogram, monthly mean extinction profiles) to keep the paper as short as possible.

We think, more cannot be done. Many parts of the paper are substantially improved, caused by the recommendations of the reviewers. But this kind of research must also tolerate hypothetical explanations and argumentation. We feel this approach, as we present it now, is justified.

**Step-by-step reply: our answers in BLUE**

Major comments

In the introduction, the authors mention that the burning season in 2019 was largest on record, they also employ superlatives like "tremendous environmental disaster" without a quantification of how the wildfires compare to previous years.

**This is now improved, see new Figure 2**

Little information is provided in the article on the Polarstern trajectory with respect to the stratospheric polar vortex. A figure showing potential vorticity and temperature evolution along the cruise is lacking. Such a figure would be very useful for interpretation of the measurements and especially for section 4 that addresses PSC, smoke aerosols and ozone depletion.

**Thank you! We followed this idea, see new Figure 7 (presenting the scaled potential vorticity and temperature time series for different height levels). Before, we present the track of the Polarstern in Figure 6.**

The fact that the observed aerosol layer originates from the Siberian wildfires is not demonstrated in the manuscript. As mentioned in the paper, several other sources of aerosols can be identified during that winter: volcanic aerosols from the Raikoke eruption, aerosols from pyroCb injection linked to fires in Northern America and PSC. The distinction between smoke and volcanic aerosols is made primarily from the spectral difference of the lidar ratio without consideration of meteorological processes that could further document the origin of the observed aerosol layer. Only one ensemble of trajectories is displayed for one day in the campaign in the manuscript. Such considerations on the evolution of meteorological evolution during the Arctic winter is needed especially in order to better differentiate smoke aerosol from PSCs.

**We provide now a large number of plausible arguments and explanations that the Siberian fires were most probably the source for the smoke over the North Pole. We introduced a new section for an extended discussion (Section 3). But it remains impossible to 'demonstrate' this. It remains a hypothesis. All the fires in 2019 at high latitudes were of minor importance (according to the Kloss et al., 2021, paper). Thus, we have Raikoko aerosol and the Siberian smoke aerosol to consider.**

**We think any meteorologically based argumentation (including simulations of long-term air mass transport) would not help, is too uncertain. The temporal distance between the fires in July and August 2019 and the Polarstern observation in October 2019 is too large. Therefore we show Spitsbergen and Polarstern lidar measurements together in Fig. 12b. Spitsbergen lidar observations of the stratospheric aerosol are available from August 2019 (and thus for the fresh fire smoke from the Siberian fires) to January 2020, and thus we have even a long overlap between Spitsbergen and Polarstern lidar observations from October to January.**

**However, we know from many observations: If smoke enters the stratosphere it will be distributed quickly over large parts of the northern hemisphere, and this within a few weeks. This is already shown several times, e.g., for example recently after the strong Canadian fires (Baars et al, 2019)**

and much earlier by Fromm et al (2008). Fact is also that the decay of a smoke-related stratospheric perturbation takes more than a half year (Baars et al., 2019).

We believe, the key point is to show (and demonstrate): Do we have convincing argument that the Siberian smoke was able to reach the lower stratosphere (in the absence of pyroCb activity)? This aspect is discussed in large detail in Sect. 3. We develop a consistent picture that this was the case. The self-lifting process (solar absorption by smoke at high APT levels, warming of the air, and ascending of the warmed air within 3-5 days up to the tropopause) was able to lift smoke up to stratospheric heights so that the smoke was later on observed over Leipzig and Ny Alesund since August 2019 and over Polarstern since the late days of September 2019.

We discuss already in the Introduction section that the Raikoke volcanic eruption caused a stratospheric perturbation in terms of AOT of 0.015-0.02 at 500 nm during its maximum impact (around 10 August 2019). This is based on observations and Raikoke simulations (Kloss et al, 2021) as well as simulations of the similar Sarychev volcanic aerosol (Haywood et al., 2010, the eruption occurred 10 years before the Raikoke eruption, i.e., in the summer of 2009) and the Raikoke AOD decreased to 0.01 in October 2019 at high northern latitudes according to the simulations. To that time we observed AOTs of 0.1 (an order of magnitude higher AOT). So, we argue we clearly need another, a much stronger source for this massive stratospheric perturbation we observed during the MOSAiC campaign. The Raikoke sulfate aerosol cannot explain the strength of the aerosol load we observed. And all the other fire-related events in 2019 were even smaller than the Raikoke impact (as Kloss et al. , 2021, pointed out). All this is now given in the introduction, with the goal to clearly state: We need a strong source to explain the observed massive stratospheric perturbation. And after checking our lidar data at Leipzig and Ny Alesund and the strong increase in stratospheric AOD (towards 0.1-0.15 over Spitsbergen in the beginning of August 2019) we had to search for a strong fire area in July-August 2019. All this is described in the Introduction. All this is quite reasonable and consistent. But sure, it remains a hypothesis as we state that finally in Sect. 3.

Regarding the lidar ratio spectral dependence (LR355 < LR532) together with the high LR532: This is clearly our number-one argument to conclude: We observed aged smoke! There is no way around it. However, to corroborate this and to better convince the reader, we present a Table 2 with lidar ratio pairs (355, 532 nm) for aged Siberian, Australian, Canadian smoke, mineral dust, volcanic ash, volcanic sulfate, urban haze, Arctic haze, etc…. And the message is then clear. This was smoke what we observed during MOSAiC.

The methodology used for the analysis and retrieval of aerosols parameters is not described in sufficient quantitative details. The description relies mainly on references, some of which e.g. Ansmann et al., 2020 is still under review. For instance, section 2 does not summarize the method used for deriving main aerosols parameters such as the backscatter coefficient or the lidar ratio and their respective error. A table could summarize main characteristics of these retrieved parameters in terms of error and vertical resolution. A summary of the POLIPHON method used to derive the aerosol mass concentration is also needed. The GMAO method is used to retrieve the tropopause height but no explanation is given on why such method is better than the classical WMO one. Since tropopause height is generally more difficult to determine at high latitude than at lower ones, such explanation is needed. Also, a description of the method used to derive the bottom and top of the aerosol layer is lacking.

We considered all the remarks and improved Section 2 significantly along the suggestions.

The hypothesis of mixing between several particle types is not fully explored. For instance, there is no clear explanation of the quantification of less than 20% for the fraction of volcanic aerosol

observed in Fall 2019 (page 7). In Figure 9, PSC are only identified over the smoke aerosol layer. Is there a possibility that PSC are also formed within the aerosol layer? Without knowledge of temperature history, it is difficult to conclude.

**We skipped this part of the discussion on the mixing of volcanic and smoke aerosol and conclusion on the smoke and sulfate aerosol fractions.**

**Regarding the PSC influence: Yes, there were several cases with PSC development in the centers of the aerosol layer. We decided not to do any correction. It is better to live with the bias than to produce a new one by correcting the effect and introduce new uncertainties in this way. We had precise temperature information, because there were 4 radiosonde launches per day. So, we had always precise PSC formation temperatures.**

What is the objective of section 3.3 (comparison with foregoing Aerosol studies)? This section cites a number of other studies analysing aerosol vertical distribution in the Arctic but not clear conclusion is driven from this section.

**The goal was to show all these different measurements from 2000 to 2019 together with our MOSAiC observations to corroborate that our measurements fit very well into the High Arctic aerosol climatology, except the smoke layer with an order of magnitude higher extinction efficients. We leave the figure in (now Figure 13), but removed Section 3.3.**

Specific comments

P2 L16 – 19: What parameters were considered from FIRMS and CAMS databases?

**We skipped the respective sentences. However, one would find maps with fire spots (FIRMS) and information about the extraordinarily strong fire season of 2019 (CAMS).**

P2 L34 – 35: Provide details on the mentioned simulations.

**We do so now in Section 3, we used the radiative transfer model ecRAD of Hogan and Bozzo 2018, and we give precise information on all input parameters needed to simulate heating rates, and how we computed ascent rates as a function of heating rates by using the approach of Boers et al., (2010).**

P3 L7 – 9: How do we know that the aerosol was trapped in the strong polar vortex?

**We removed this statement. We do not know what is going on (in detail) below the vortex!**

P5 L8 – 9: How were identified the PSC: by visual inspection? There is no explanation.

**Yes, by visual inspection. We explain what we did to remove the impact, and how large the remaining uncertainty is in terms of AOT (about 5% or less). We did not make any further attempt because we believe we would introduce a (new) bias. It is better to see the remaining effect in the depolarization ratio and Angstroem time series and to get, in this way, an idea about the impact.**

P6 L14: How are the bottom and top of the aerosol layer determined?

**Again by visual inspection and Rayleigh signal fit to the measured signal profiles and then by using a threshold values of 1.1 for the 1064 nm total-to-Rayleigh backscatter ratio. We provide the details now in Section 4.1.**

P6 L18 – 20: The HYSPLIT trajectories do not demonstrate that the smoke aerosol layer could have been trapped within the polar vortex. Figure 5 is not very clear: provide explanation for the colours of trajectories.

**We improved the text. But we still mention that the aerosol was obviously trapped.**

**Colors are just used to distinguish different sub groups of trajectories. This is now explained.**

P6 L23 – 24: How are determined error bars in Figure 6? Are they shown as one or 2 sigma?

**This is one standard deviation, as usual.**

P8 L28 – 31: How are the refractive index and SSA shown in Table 1 determined?

**We state now in Section2: The single scattering albedo SSA, defined as the ratio of scattering-to-extinction coefficient, is finally calculated from the retrieved particle size distribution and complex refractive index characteristics with an uncertainty of $\pm$0.05.**

P9 L22: Figure 10 is not well explained. Significance of the layers mentioned in the legend is not clear.

**We skipped this figure.**

P9 L23 – 24: If the aerosol layer was trapped in the strong polar vortex, how could it be influenced by smoke aerosol from lower latitudes? What about subsidence within the vortex? The situation of the lidar measurements with respect to the polar vortex is not clear and needs better description.

**We introduced the new figure (Fig. 7) with the scaled potential vorticity (sPV) in (a) and temperature for different height levels in (b), along the Polarstern route. This was requested by the second reviewer as well. However, we leave out to discuss the impact on horizontal or vertical transport. We just mention that the vortex has a strong impact on the weather and air flow conditions below the vortex and widely suppresses meridional exchange.**

---

## Author Comment (AC2)

Dear Reviewer!

First of all thank you for careful reading and all good and constructive suggestions which improved the paper significantly (we hope). It is usually often of great and complementary help when a reviewer is not an expert in the same field as the authors. You realized certainly that we are not working every day in the field of stratospheric ozone depletion and are also not just experts for polar meteorology.

Before we answer all comments and questions, step by step and item by item, let us provide a brief overview of the essential changes:

Motivated and mostly forced by the other two reviews, we went deeply into the literature with special focus on volcanic sulfate aerosol (originating from the recent Raikoke and from the Sarychev volcanic eruptions), into our own Leipzig lidar data analysis, into CALIPSO observations and HYSLPIT trajectory analysis and conducted further simulation studies to bring together much more solid information and substantial argumentation to provide a plausible link between the Siberian fires (in July-August 2019) and the polar smoke layer we observed later on in the North Pole region (since late September 2019).

Some significant changes before we start the step by step reply:

- We show a new figure (Fig. 2, time series, MODIS AOD from 2000-2020), highlighting the extraordinarily strong 2019 fire season in central and eastern Siberia.
- We introduce a new table (Table 2), highlighting the unique signature (fingerprint) of aged wildfire smoke, namely the inverse spectral slope of the lidar ratio (LR355nm < LR532nm) together with high LR532. We compare these smoke features in the new table with the ones for mineral dust, volcanic sulfate and ash, Arctic, European, American haze, etc.
- We show a new HYSPLIT plot (Fig. 5) to better explain the link between the CALIPSO overflight over Siberia and the strong fires... downwind (west!) of the flight track.
- We show a new figure with the Polarstern track (Fig. 6, same as in Engelmann et al., 2021).
- We created a new figure (Fig. 7) to better show the influence of the polar vortex on all the observations.
- We introduced a new section (Section 3) on Siberian fires, self-lifting hypothesis, consequences for aging (leading to spherical smoke particles), and also on the potential miss-classification of smoke layers as sulfate layers when using the CALIPSO aerosol typing scheme (Sect. 3.1).
- We shortened the ozone-depletion/smoke/PSC section as requested, and combined the two figures of this section to one figure (Sect. 5, Fig. 17).
- We skipped several figures (layer-depth histogram, monthly mean extinction profiles) to keep the paper as short as possible.

We think, more cannot be done. Many parts of the paper are substantially improved, caused by the recommendations of the reviewers. But this kind of research must also tolerate hypothetical explanations and argumentation. We feel this approach, as we present it now, is justified.

**Step-by-step reply: our answers in BLUE**

Ohneiser et al. present a summary of a persistent wildfire smoke aerosol layer observed during the MOSAiC field campaign in the high Arctic. The authors primarily focus on observations obtained from the "Polly" Raman lidar operated onboard the Polarstern, but also compare with other lidar platforms (e.g., CALIPSO and KARL). With the unique observations made possible by Polly and the

MOSAiC campaign, the authors are able to provide quantitative descriptions of the optical and microphysical properties of observed aerosols for a region where such measurements are sparse. These measurements are made even more important by the unique atmospheric and aerosol conditions related to the 2019 Siberian wildfires and the unique polar vortex and ozone conditions of late-2019 and early 2020.

I am not a lidar measurement expert; in fact the editor has asked me to provide my views for portions of the manuscript related to polar vortex conditions and ozone depletion. However, I of course read through the entirety of the paper (multiple times), and so I feel comfortable saying that the paper is generally well-written. While I can not provide an expert assessment of the observational aspects and descriptions, as a "knowledgeable layperson", I'd say The authors mostly gave thorough and convincing descriptions of the observational data and their characterization. The paper is clearly cutting-edge and would be a perfect fit for ACP. However, I do have a few concerns, some more major than others; most of these are related to the aspects surrounding the polar vortex and ozone depletion in Section 4. Since Section 4 is a relatively small portion of the overall paper, I would say the revisions required to address my comments would be relatively minor to implement, but I do feel strongly that they are necessary changes.

General comments

(1) In reading through the paper, reaching Section 4 felt like a definite shift. Up to that point, everything felt (as a reader) mostly authoritative and backed up by quantitative results. In contrast, Section 4 felt very vague and overall speculative. One paragraph particularly stuck in my mind -- p14, L10-15 -- as it included many instances of language such as "were probably ..." and "were likely". Three "precise" research questions are posed early on in the section, and yet the remaining results and discussion do nothing to answer them and instead involve results that only hint at possibilities. Section 4 essentially says "the smoke aerosol layer could have been important for ozone depletion" simply because the PSC, smoke, and ozone depleted layers overlapped in the vertical (Figures 15 and 16).

**We got the point and worked on Section 4 to meet the concerns. Section 4 is now Section 5 in the revised version. We re-arranged the text. See further explanations below.**

**Let us briefly start with this: There are many hypothetical arguments and aspects in the paper. This must be allowed as long we state that clearly. And we do so. For example, there is no clear answer to the question: What was the most likely source for the smoke we observed over the High Arctic? We try to answer this question in the new Section 3. There is also no clear answer to the question: Is it possible that the smoke contributed to the strong ozone depletion?**

**We think that nobody has an idea at the moment how efficient glassy organic particles are in heterogeneous chemical processes in the stratosphere and that nobody knows how these glassy particles can influence PSC formation! It will take years to investigate this properly! You need sophisticated laboratory studies for that. Modelling is insufficient.**

**Our job is simply to raise our hand to tell the ozone science community that there was a persistent smoke layer over the Arctic in the spring of 2020 when a record-breaking ozone depletion occurred. Please keep that into consideration in your future studies … .**

**By the way, we operate another lidar at Punta Arenas, Chile, and detected the Australian smoke from the tropopause up to 26-30 km height, from January 2020 to April 2021. We were sure, already in March-June 2020, that there will be a strong ozone depletion in the southern**

**hemispheric spring season (September to November) because of the smoke…. sure, without any (precise) idea about the pathways…**

There are many details that would have to be worked out in much more detail to answer the questions posed in Section 4. For instance, if the MOSAiC measurements were to be of use in answering these questions, then there would need to be details about the time-varying geometry of the vortex. While it's a fairly safe assumption that the vast majority of measurements (lidar, ozonesondes, etc) sampled air within the polar vortex, on weekly timescales the vortex and its edge are quite mobile (and similarly, so is the region of air cold enough to form PSCs). This means that measurements could sometimes sample air in different parts of the vortex, meaning some measurements would be less relevant than others in establishing where things were "in the right place at the right time".

**First of all, this triggered us to present the new Figure 7 with the scaled potential vorticity and temperatures for different height levels along the Polarstern route from October 2020 to May 2020.**

**As mentioned, we re-phrased and rearranged the text to give the impression in Section 5 that we just want to point to a new aspect: smoke and ozone depletion. …. not more…**

**If we do not show that, others will do it. Furthermore, we see the same in the southern hemisphere:  a strong  ozone depletion height range,  but this time right in the center of the Australian smoke layer.**

As written, I do not feel Section 4 should be kept.  I would suggest that it should be shortened and perhaps folded into the current Section 5, with content from Section 4 forming the basis of some discussion (i.e., providing motivating context for future work). At the very least, Section 4 should be shortened and rearranged so that the authors present the "possibility" before posing the big picture questions. The authors should also be clear (if they keep any of Figures 15 or 16) that conclusions cannot presently be drawn from their analysis.

**This is done! We agree! We shortened Section 4 and rearranged the remaining text parts.**

In my specific comments below I provide more detailed questions/comments related to the content of Section 4.

(2) Again, this is not my area of expertise so perhaps these are naive questions: The title of the paper and the introduction generally outline the assumption that the smoke aerosol layer measurements were tied to the Siberian wildfires, but are there not other potential confounding sources?

**We changed the title! Removed 'Siberian'!**

**We agree, we should be  more careful and therefore rephrased the introduction. But as you will see, there was the volcanic sulfate layer (with Raikoke as the source) and the smoke layer (from the huge Siberian fires as we hypothesize) and nothing else. There were many fires within the Arctic circle in 2019. But very strong and long-lasting fires were needed and probably also stagnant weather conditions to initiate self-lifting in order to end up with smoke in the stratosphere. Only if the smoke reaches stratospheric heights it will be distributed of the entire northern part of the Northern Hemisphere and will survive for months.**

Can the authors provide more evidence to say that the Siberian wildfires were likely the dominant/overwhelming contribution to the observations discussed throughout the paper?

**This motivated the new Section 3 on the Siberian fires, on self-lifting aspect, on smoke aging aspects, and even on the failure of the CALIPSO aerosol typing scheme to identify smoke, but instead to miss-classify that as sulfate aerosol.**

The paper lacks some descriptive context as well: how severe were the 2019 Siberian wildfires in comparison to prior years? It is perhaps fortuitous that the MOSAiC campaign was able to sample aerosol conditions largely influenced by Siberian wildfires, but is there a quantitative measure of how unusual 2019 was that would better emphasize the importance of the MOSAiC measurements? These kinds of details may be present in the articles that the authors cite, but I think it's worthwhile for the authors to make them explicit where possible.

**We include the new Figure 2 with MODIS AOD observations from 2000 to 2020. The maximum AODwas observed in August 2019 during the last 20 years.**

**Yes, it was fortuitous! And we (as aerosol lidar enthusiasts) were happy to observe this exotic aerosol layer throughout the winter half year of 2019-2020.**

(3) There were a couple of times in the paper where a Figure is introduced and a portion of it is discussed, but then the discussion moves onto one or more different figures before later coming back to the initial one. An example of this is Figure 9: On page 9, line 17 Figure 9a is introduced, but then within the same paragraph the authors move on to Figure 10. The authors go back to Figure 9a, and eventually introduce Figure 11 before eventually mentioning Figure 9b. I recognize that this may just be personal preference, but I think that cases like these generally suggest that text or figures should be re-organized to better maintain the serial nature of the text.

I understand that in the final manuscript the authors will generally not have much of a say where figures will end up in relation to the text, but it is still a bit awkward for a reader to have to flip between multiple figures for a given piece of text (whether on paper or digitally). I wouldn't classify this as a major issue in this paper, but I still urge the authors to consider whether their figures or text could be restructured to make things flow more naturally.

**We agree, kept these suggestions in mind, and did our best to improve the structure. We partly rearranged figures to avoid such a confusing back and forth in the discussion.**

Specific Comments

P6, L17 and Figure 5: The information about the arrival height of 10km should be included in the figure. Also, what about the vertical information? How did these trajectories evolve in terms of altitude/pressure over this time period? Is this not important information for discussing the airmass origins?

**We added the plot with the height information (lower panel of Figure 9, this was Figure 5 in the old version), and mention the arrival height of 10 km in the caption.**

P10, L22: I do not believe the Lawrence et al ref here shows that the polar vortex collapsed on 20 April. In fact, their Figure 10 shows the vortex was unusually long-lived, lasting into May. Maps of the vortex on April 20 show that the vortex was undergoing a split, so it's probably fair to say the vortex began decaying around this time. However, if you click through further dates …., I think it's clear that a distinct polar vortex is present well into May. This comment is also relevant to other spots in the text where the authors mention the vortex collapsed in late April (e.g., P2, L11).

**We improved the text accordingly.**

P13, L14-15: This requires more nuance; it wasn't just weak planetary wave forcing, but also the very strong dynamical influence of downward wave reflection events during the 2019/2020 winter season (this is discussed in detail in the Lawrence et al ref).

**We agree! But at the end we removed the long paragraph with all these meteorological statements to shorten the section. We did not see a need anymore to explain why the vortex was so strong.**

P14, L3-9: While questions 2-3 are interesting and are worth further exploration, I'm not sure question 1 has much scientific merit in relation to the polar vortex. There are too many issues with timing and location and other confounding factors. Early in the season, the polar vortex was generally not extremely strong or weak. Furthermore, the very strong vortex conditions in Jan-Mar coincide closely with the dynamical downward wave reflection events mentioned above.

**We removed question 1!**

P14, L18-22: While true, I'm not convinced this is an appropriate apples-to-apples comparison. For instance, the 2019 Australian wildfires were severe enough to be comparable to a moderate volcanic eruption in terms of the impacts to solar radiation (as noted in the Khaykin 2020 paper already cited). I do not think that the Siberian wildfires have been established as being anywhere near as severe. There are other differences, but the point is, this kind of comparison runs the risk of equating inherently different events, which could be much more coincidental than this statement suggests.

**We removed this part in Section 5. Only in the conclusion section 6, we left the remark concerning the ozone hole over Antarctica and the occurrence of the smoke at the same time.**

P14, L23: How much of this introductory material is actually necessary for introducing the two figures of the section, which don't actually answer the "big questions" posed beforehand? The beginning of this section sets up big expectations that aren't met.

**We re-arranged the text as already mentioned…. The 'big questions' are just briefly mentioned in the discussion on the potential pathways of the smoke impact on ozone depletion…**

Figures 15 & 16: These figures appear to me to be mostly redundant. Figure 16 is mostly a coarse-grained version of Figure 15 with additional information about the ozone anomalies. It seems like this information could easily be combined into a single figure.

**Yes, we prepared one figure but did not find a solution to 'efficiently' combine the information in both figures. But having one figure now, it is easier to see the effects. We indicate better the different months J: January, F: February, M: March, in both figures, (a) and (b) to facilitate the study of both figures.**

P16, L16-22: This paragraph and Section 4 (which has major issues) seems like a missed opprtunity to provide expert guidance on how MOSAiC measurements could assist future studies that may attempt to answer such questions.

**We think, Section 5 (the former Section 4) is sufficient to initiate, trigger, or stimulate further research. We are not experts for atmospheric chemistry, we are not members of the stratospheric ozone community, we are even not members of a well-organized stratospheric aerosol community. So we prefer not to stand up as a teacher to tell the pupil what to do next.**

---

## Author Comment (AC3)

Dear Mike!

First of all thank you for your careful and detailed study of the manuscript and for the many, very 'direct' and constructive statements. We had your comments always in mind when we revised the manuscript. Before we answer your comments and questions step by step, we would like to briefly summarize what we did:

We went deeply into the literature with special focus on volcanic sulfate aerosol (originating from the recent Raikoke and from the Sarychev volcanic eruptions), and studied again many papers including the papers you suggested, we also re-checked our own Leipzig lidar data analysis, we again checked many CALIPSO observations and performed an extended HYSLPIT trajectory analysis and conducted further simulation studies (on self-lifting aspects) to bring together much more solid information and substantial argumentation to provide a plausible link between the Siberian fires (in July-August 2019) and the polar smoke layer we observed later on in the North Pole region (since late September 2019). The focus is on the potential self-lifting process, on smoke aging and on the consequences of slow self-lifting and thus efficient aging of the smoke particles for the morphological and optical properties of smoke. They become spherical before they reach the stratosphere and cause low depolarization ratios.

We did all this very carefully and even if we present a large number of reasonable and plausible arguments for our chain of hypotheses we emphasize that our entire set of arguments still remains at a hypothetical level.

Nevertheless, there is no doubt! What we observed during the MOSAiC expedition is clearly and unambiguously wildfire smoke. There is no way around. We observed this inverse spectral dependence of the lidar ratio (LR355 < LR532) together with the high LR532 around 85 sr, and that is a unique fingerprint of wildfire smoke! Note, that we observe such smoke fingerprints again and again, since 1998 (Wandinger et al., JGR, 2002). …. in Canadian, Siberian, Australian smoke, …

And one of the convincing and motivating arguments to search for a much more massive aerosol source than the Raikoke volcano was: The Raikoke-related AOD at high northern latitudes was about 0.01 at 500 nm in autumn 2019 (according to simulations of the 2019 Raikoke as well as of the 2009 Sarychev volcanic aerosol impact and corroborated also by Raikoke aerosol observations at lower latitudes, discussed in Kloss et al., 2021). But, we observed AODs around 0.1 in autumn 2019 in the High Arctic, and thus an order of magnitude higher values. We were forced to find a massive aerosol source.

Some significant changes before we start the step by step reply:
- We show a new figure (Fig. 2, time series, MODIS AOD from 2000-2020), highlighting the extraordinarily strong 2019 fire season in central and eastern Siberia.
- We introduce a new table (Table 2), highlighting the unique signature (fingerprint) of aged wildfire smoke, namely the inverse spectral slope of the lidar ratio (LR355nm < LR532nm) together with high LR532. We compare these smoke features in the new table with the ones for mineral dust, volcanic sulfate and ash, Arctic, European, American haze, etc.
- We show a new HYSPLIT plot (Fig. 5) to better explain the link between the CALIPSO overflight over Siberia and the strong fires… downwind (west!) of the flight track.
- We show a new figure with the Polarstern track (Fig. 6, same as in Engelmann et al., 2021).
- We created a new figure (Fig. 7) to better show the influence of the polar vortex on all the observations.

- We introduced a new section (Section 3) on Siberian fires, self-lifting hypothesis, consequences for aging (leading to spherical smoke particles), and also on the potential miss-classification of smoke layers as sulfate layers when using the CALIPSO aerosol typing scheme (Sect. 3.1).
- We shortened the ozone-depletion/smoke/PSC section as requested, and combined the two figures of this section to one figure (Sect. 5, Fig. 17).
- We skipped several figures (layer-depth histogram, monthly mean extinction profiles) to keep the paper as short as possible.

We think, more cannot be done. Many parts of the paper are substantially improved, caused by the recommendations of the reviewers. But this kind of research must also tolerate hypothetical explanations and argumentation. We feel this approach, as we present it now, is justified.

**Step-by-step reply:** our answers in BLUE

Ohneiser et al. (2021), hereafter abbreviated as "O21," is a provocative study of upper troposphere, lower stratosphere (UTLS) aerosols at high northern latitude between fall 2019 and spring 2020. It is provocative in that O21 observe UTLS aerosol nearly daily in that temporal span and conclude that the composition is wildfire smoke, the source is fires in a sector of Siberia in July 2019, and the transport pathway to the UTLS is diabatic heating/lofting.

The manuscript is also provocative in that Raikoke volcano (Kuril Islands) erupted in late June 2019 and polluted the UTLS with a mass of $SO_2$ on par with or exceeding other eruptions that generated stratospheric clouds persisting for greater than a half year (E.g. Kasatochi, Sarychev Peak, Nabro, Kelut, Calbuco. See Solomon et al. (2016) for a tabulation.).

**We checked the potential strength of the Raikoke sulfate aerosol in terms of AOD at 500 nm again, and we used especially the paper of Haywood et al (2010) for the Sarychev volcanic eruption (causing a very similar perturbation 10 years ago, in the summer of 2009), and came to the conclusion that the Raikoke-related AOD in October 2019 in the High Arctic was 0.01 to 0.015 at 500nm, but we observed 0.1, and thus an order of magnitude more, when we started the MOSAiC measurements! There is no doubt! We need a 'better' source for the aerosol we observed during MOSAiC. This is now clearly mentioned in the introduction. We must say, all the papers on Raikoke aerosol are a bit confusing because they provide observations of $SO_2$ (yes this is related to Raikoke) but then they show extinction coefficients (and this is related to EVERYTHING, smoke, sulfate, background aerosol etc...). And we must say, in many papers the authors obviously just did not know what they measured. And the authors were probably miss-guided by CALIPSO lidar observations and CALIPSO aerosol typing. In the CALIPSO data base, you will find sulfate layers only (if we neglect the few pyroCb smoke cases). We (at TROPOS, Leipzig) compared our own Raman lidar observations (14 August 2019) with an almost direct over flight by CALIPSO (14 Augsut2019, same time): We saw the same layer features as CALIPSO, we measured the same depolarization ratios, they classified the layer as sulfate aerosol layer (based on depolarization observations), and our lidar ratio observations (LR355<LR532, and high LR532) clearly told us: WILDFIRE smoke. CALIPSO is a 'simple' backscatter lidar and we use an advanced, state-of-the-art Raman lidar, and at the end, we have to defend our unambiguous observations (facts) and nobody asks: What may go wrong with the CALIPSO aerosol classification (assumptions) . Motivated by this mismatch, we introduce an extra subsection (Sect. 3.1) on the the miss-classification of smoke layers when using this quite simple CALIPSO aerosol typing scheme. Furthermore, we will write a paper on this topic (submission in September 2021).**

If O21's conclusions are borne out, it will be a new insight into the polar UTLS and smoke transport to the UTLS. However, there is overwhelming observational evidence that Arctic UTLS in the second half of 2019 and early months of 2020 was blanketed by Raikoke sulfates.

**The impression ….. 'overwhelming observational evidence' …. sounds strange in our ears. You mean you have rather clear arguments that all the aerosol we detected and measured in the stratosphere in 2019 … was Raikoke aerosol? We have clearly to state: We disagree! Our observations are in full disagreement with the 'overwhelming observation evidence'.**

**And concerning the 'evidence': In the Kloss et al. (2021) paper, for example, they show color-scaled OMPS-LP-derived AODs with high resolution up to 0.025 (ONLY) and all higher values are just given in the same color as for 0.025, simply in red. Why (?), because of saturation effects? This is a strange way of presentation, when keeping in mind that the AOD was obviously close to 0.1 in August to October over high northern latitudes (according to the Leipzig, Spitsbergen, and Polarstern lidar observations at 532 nm)! Furthermore, a rather inhomogenuous AOT distribution is shown by Kloss et al. with increasing AOT from low to high latitudes. This is in contradiction with model results which indicate a homogeneous Raikoke sulfate particle distribution from 45 to 90°N. The same was observed and modelled after the Sarychev eruption. Why was the volcanic sulfate aerosol this time (in 2019) inhomogeneously distributed from 45 to 90N according to the observations of Kloss et al., and after the Sarychev eruption it was homogeneously distributed, and thus in full agreement with the models? However, this inhomogeneous stratospheric aerosol distribution in 2019 is very reasonable when keeping the strong Siberian fires into consideration. But Kloss et al. (2021) totally ignored a potential contribution by the Siberian fire smoke.**

Secondly, there is abundant evidence that the UTLS aerosol picture O21 describe over the July Siberia source sector is also dominated by Raikoke SO$_2$ and sulfates.

**If the Raikoke sulfate aerosol was dominating (with an absolute maximum AOD of around 0.02 at 500 nm in mid August 2019 and later on with an AOD of the order of 0.01 in October 2019), how can we then measure AODs even exceeding 0.1? We observed stratospheric AOTs of 0.1 with lidar at Leipzig, at Spitsbergen, and at Polarstern in August to October 2019!!!! So, we clearly have a conflict, if we want to continue with our message: Raikoke aerosol dominated the stratospheric aerosol layer in 2019. The simple answer could be: All prediction of all the models are wrong, i.e., the Raikoke impact was totally underestimated. But then please tell us why the Raikoke effect on stratospheric AOD was about a factor of FIVE higher than the Sarychev volcanic AOD although the emitted SO2 amount was just 20 % higher in comparison with the Sarychev SO2 amount.**

Thirdly, the lidar data
presented by O21 are more closely aligned with spherical sulfate droplets than smoke particles. These three points are elaborated on below.

**Your phrasing suggests, stratospheric smoke particles are 'by law' nonspherical and thus produce significantly enhanced depolarization ratios. Yes, this is probably true for smoke particles reaching the stratosphere by the fast pyroCB-lifting processes, then the depolarization ratio is significantly higher than zero. The emitted fires retained their nonspherical shape when reaching the stratosphere. After 3-6 months even these smoke particles would have developed a perfect spherical form by smoke aging processes and the depolarization ratio would be close to zero (as can be found in Baars et al., ACP, 2019). In contrast: if the smoke particles are lifted slowly into the stratosphere, by self-lifting**

**processes, so that aging processes could be completed within the humid tropospheric environment (all this takes 2-4 days only) then the smoke particles have the chance to reach the stratosphere as spherical particles. And this seems to be the case here. The Raman lidar observations of the lidar ratios at 355 and 532 nm clearly and unambiguously indicated SMOKE (LR355<LR532, LR532 very high), and at the same time the depolarization ratio was close to zero (clearly indicating spherical particles). But now, we have a problem with the CALIPSO aerosol typing scheme, because such a case of smoke self-lifting is not considered in the CALIPSO aerosol typing scheme. As a consequence, if particles are spherical they are assigned automatically as sulfate aerosol particles.**

**All this is now explained in detail in the new Section 3.**

Point 1

Kloss et al. (2021) show that the Raikoke volcanic cloud dominated the high-latitude northern hemisphere from eruption through the early months of 2020. I consulted Chris Boone, ACE-FTS Project Scientist and co-Principal Investigator, to help qualify the 2019/20 UTLS plume further. ACE not only delivers aerosol extinction profiles but profiles of $SO_2$ as well. These were combined by Cameron et al. (2021) in an examination of several UTLS volcanic events including Raikoke; the results further qualify those of Kloss et al. and show the strong presence of Raikoke $SO_2$ and sulfates at high northern latitude in summer and fall 2019. In addition, ACE IR spectra have been used to identify smoke aerosol in connection with ACE Imager extinction profiles (Boone et al. (2020)). The same technique was applied to high-latitude northern hemisphere 2019/20 ACE data while invoking published sulfate IR spectra for comparison. The findings are summarized as follows. In July 2019, the lower stratosphere in the latitude region near 60 degrees north is stuffed with sulfate aerosols from the Raikoke eruption. …

**We can confirm this. Our lidar observation at Leipzig (52N) on 23 July 2019 of 355 nm lidar ratios of 45 sr (typical value for freshly formed sulfate particles) and depolarization ratios at 355 and 532 nm close to zero suggest non-absorbing, spherical sulfate aerosol. Unfortunately, the volcanic AOD was too low, of the order of 0.005…, so that we could not determine 532 nm particle extinction and lidar ratio values to that time (before August 2019, when the stratospheric aerosol increased strongly).**

….. Identification of the composition is accomplished by looking at the infrared spectrum of the aerosols and noting the coincident enhancement of $SO_2$ and ACE Imager aerosol extinction layers. In September/October 2019, there is a blanket of aerosols in the lower stratosphere in the latitude region near 80 degrees north. ….

**Coincidence (SO2 and enhanced extinction) alone is not a convincing argument to us. Again, we confirm that there was a lower stratospheric aerosol layer around 80N in September and October 2019 (by our Spitsbergen and Polarstern lidars). And we can add, the stratospheric AOD was close to 0.1 at 500nm, and thus a factor of 5-10 higher than the expected Raikoke AOD.**

The blanket appears to be composed of Raikoke sulfate aerosols. …

**…according to CALIPSO aerosol typing…, but not in agreement with our dual-wavelength Raman measurements of LR355<LR532 together with high LR532.**

In February/March 2020, the aerosol blanket in the lower stratosphere near 80⁰N is still present. …

**Yes, again, we can confirm this and the layer was still 10 km in depth.**

…. SO2 has decayed.  However, the spectra associated with the Imager aerosol layers are consistent with sulfate.  At no point did we find any evidence of biomass burning smoke playing a role in these stratospheric aerosols (Boone et al., 2020). Detailed support for these findings is available upon request.

**We are not sure whether these spectra contain accurate information on the aerosol type. 'Evidence' and 'consistency' sound convincing, but is that sufficient to make solid statements on the aerosol type? Is our solid fingerprint (the measurement of an inverse spectral behavior of the lidar ratio as an unambiguous fingerprint for smoke) not much more 'convincing'?**

Point 2

O21 hypothesize that their MOSAiC Arctic 2019/20 lidar signals are dominated by an impressive build-up of UTLS smoke that began between ~24-28 July 2019 in a zone within Siberia centered roughly at 60°N, 110°E (Figure 2). They present nadir satellite imagery combining fire hot-spot data, true-color imagery, and aerosol optical depth retrievals to show an intensification of burning and smoke concentration. They also display a single CALIOP curtain on 26 July to characterize the vertical structure and ascent of smoke layers (Figure 3). There are several concerns regarding Figures 2, 3 and their interpretation, listed below.

 It is simply not possible to know from a single CALIOP curtain if ascent is taking place; there is insufficient information from such a quasi-instantaneous vertical snapshot. Additional information must be brought to bear.

 **Yes, sure!**

There are technical issues with Figure 3 and its caption. The date of the CALIOP image is given as "2016." This is obviously an inconsequential typographical error.

**We improved this**.

But more substantially, the latitude, longitude coordinates labeled along the bottom are wrong.

**We improved this as well. We apologize for all these mistakes!**

The curtain displayed is actually situated west of the Figure 2 boxed focal point. It is west of the labeled coordinates by approximately 25° longitude. Compare Figure 3 with the actual orbital track and lidar data …

**We now show in the new Figure 5  HYSPLIT backward trajectories, and indicate the CALIPSO flight track in this HYSPLIT map as a straight line and also show  box 2 (with strong fires) defined  in former Figure 2 (now Figure 3).**

This is scientifically relevant for two reasons. One is that the vertical aerosol profile over the Siberia focus zone is unknown to the reader. Secondly, the aerosol profile on display is nominally upwind of the Siberia box, meaning that the history of those aerosols may be disconnected with processes occurring in the Siberia box.

**The HYSPLIT backward trajectories in the new Figure 5 show that CALIPSO measured the smoke produced in box 2, because the track was downwind of box 2, and not upwind. Therefore, we are**

**convinced that the CALIPSO backscatter curtain plot provides an impression of the vertical distribution of smoke originating from the fires in box 2.**

**But is that essential? The main reason to include the CALIPSO figure is to show a convincing example: Yes, there was a lot of smoke and the smoke was everywhere up to the tropopause and even above the tropopause.**

Indeed, it appears from the CALIOP curtains linked above that the "Ascending smoke plumes (mostly in green) are visible up to the tropopause at 10-11 km height as well as in the lower stratosphere…" [Quoted from Figure 3 caption] are mostly assigned as sulfate by the CALIOP version 4 aerosol subtype algorithm.

**CALIPSO had no choice or chance…, as we already discussed: The depolarization ratio was close to zero, and then there is only one solution: this is sulfate aerosol!**

One can see from CALIOP curtains just upwind of the Figure 2 Siberia box, on days leading up to the O21 smoke AOD buildup, an assignment of sulfate subtype to "mostly in green" backscatter in the UTLS. Here is an example from 20 July …. Similar scenes are found each day thereafter leading up to the 24-28 July O21 period of focus . This multi-day, broad, high-latitude swath of aerosols primarily defined as "sulfate/other" conforms to the findings of Cameron et al. and our deeper investigation into ACE July 2019 profiles near 60°N.

**As stated already, we confirm this finding: Our July 2019 lidar profiles at Leipzig provide clear signatures for volcanic aerosols. This is not surprising because the extreme fires in Siberia accumulated between 19 July and 14 August (according to the very nice paper of Johnson et al, 2021), and caused a huge jump in the AOT (in the troposphere as well as in the stratosphere) NOT before the beginning of August 2019. By the way, even Johnson et al. saw a lot of structures in the stratosphere over their field site in Alberta in August (and later on), but unfortunately they restricted their discussion on the backscatter and ozone profile observations in the troposphere (this information is obtained by personal communication).**

**We were a bit disappointed when evaluating the Cameron et al. paper because the most interesting month (August 2019) is missing, probably because of missing observations because of saturation effects as a result of too high particle extinction coefficients.**

It stands to reason then, that if Raikoke sulfates are blanketing high northern latitudes at that time (Kloss et al., Cameron et al., and our investigation), they would also be evident over the Figure 2 Siberia box before, during, and after the hypothesized smoke uplift. The CALIOP example below shows aerosol subtype findings consistent with the prior examples on 20 July over the Figure 2 box….

**Again, CALIPSO aerosol typing in July is ok, the rest is not ok to our opinion.**

Hence, if the July 2019 Siberia-zone smoke is the initial condition for the O21 lidar observations in fall and winter, the presentation surrounding Figure 2 and 3 is insufficient to make that case.

**Yes, such a presentation is insufficient and remains insufficient. We agree that the text in the submitted version of the paper was misleading. We gave the impression that would be able to offer a solid explanation. To keep the answer short: The goal of the new Section 3 (with the discussion of Figures 2 and 3 in the old version) is now to offer a plausible way and a lot of reasonable arguments that the Siberian fires were most probably responsible for the smoke we observed during the MOSAiC expedition. However, we clearly state that all this is based on several hypotheses. As given now more**

precisely in the introduction, the motivation for the extended Section 3 on Siberian fires is simply: We observed a strong perturbation of the stratospheric aerosol layer by smoke and we realized that this perturbation is an order of magnitude stronger (in terms of AOD) than the Raikoke sulfate impact and we asked ourselves: What can cause such a strong impact on the stratospheric aerosol load? What was the source for this smoke? And the jump in stratospheric AOD toqrds 0.1 guided us to check the fire maps for July and August 2019.

Sure, there is no way at all to present a clear solution, that the smoke produced over Siberia was definitely (no doubt at all) the source of the smoke over the North Pole 3-6 months later on. Even, modelling and air mass transport analysis would not help. But we have the Spitsbergen observations from the beginning of August 2019 to January 2020, and thus a coherent link from the fire months (July and August) to October, and even an overlap (October to January) with the Polarstern lidar observations. All this is shown in Figure 12 in the revised manuscript.

Point 3

O21 acknowledge that the stratospheric aerosols detected by MOSAiC lidars were spherical according to aerosol depolarization measurements (Page 7, line 21). In principle, there seems to be less uncertainty as to the shape, composition, and depolarization of volcanic liquid sulfates in contrast to biomass burning smoke. Lidar measurements of tropospheric and stratospheric smoke converge on the idea that smoke is depolarizing (e.g. Burton et al., 2015; Fromm et al., 2008), suggesting some amount of asphericity. The first reported lidar observations of stratospheric smoke emphasized the unmistakable signal of depolarization (Siebert et al., 2000; Fromm et al., 2000). More recent stratospheric smoke papers, cited within O21, consistently report depolarization by smoke particles. Hence O21 would be establishing a new finding--stratospheric smoke with essentially no aerosol depolarization. Arguing for this peculiarity, O21 mention an aging process and a collapse of black carbon core. Neither process is described, and no publications are cited. Given the weight of evidence for the overarching presence of Raikoke UTLS liquid sulfates during the O21 reporting period and the MOSAiC depolarization results provided therein, the arguments for particle aging and black carbon core- shell collapse must be made more substantively.

This was another motivation to introduce the new Section 3. The first version of the text was six pages long to bring together all necessary information on particle aging, the influence of smoke aging on morphological properties and finally on the optical properties of smoke particles as given in Ansmann et al. (2021). But this would be too long in such a MOSAiC paper. So, we decided to provide a compact Section 3 only. All the published knowledge about stratospheric smoke (or better UTLS smoke) and resulting depolarization ratios measureable with lidar is linked to pyroCb-related smoke lifting. And this fast lifting of smoke by pyroCb convection prohibits particle aging and the development of a spherical shape before the smoke particles reach the stratosphere. They keep there nonspherical shape and thus produce significant depolarization of laser light. This is the present status of knowledge, and this is considered in the CALIPSO aerosol typing scheme, and this is the reason that any stratospheric aerosol layer with low depolarization ratio is classified as sulfate aerosol. In the Burton et al (2015) paper the smoke reached the dry upper troposphere, where favorable conditions (dry, no condensable gases) were given to slow down the particle aging process so that the particles remained nonspherical.

But now (in the case of the extreme Siberian fires) we had to introduce the self-lifting process. Otherwise there would be no smoke in the stratosphere, because as you told us (and we checked that also), there was no pyroCb development. And self-lifting takes some days, and this time is sufficient to finalize the aging process. And at the end of any aging of smoke particles is the spherical shape, the evolution of a perfect spherical shell indicated by the low depolarization ratios. And as we show in the

paper of Baars et al. (2019), even the stratospheric smoke is aging and reaches the spherical form, but this takes months… before the depolarization ratio goes down to zero again. By the way, we observed the same for the Australian smoke over Punta Arenas. In January and February 2020, the smoke depolarization ratio was significantly enhanced and then dropped immediately to zero in March-April 2020, at least at the lower heights of the stratosphere.

In conclusion, this is the first time that we have smoke in the stratosphere (and there is no doubt that is was smoke because of the invsere lidar-ratio spectral dependence) and that this smoke was not depolarizing. This is quite a very new aspect. And it is therefore not surprising that such a unique event is not considered in the CALIPSO aerosol typing scheme. But the other way around, it is impossible that the UTLS aerosol layer we observed consisted of sulfate particles. All the observed facts (lidar ratios, Angstroem exponent) would be in severe contradiction with this conclusion. And even the simple Mie models would be  in trouble to produce lidar ratios of 55 sr at 355 nm and 85 sr at 532 nm for typical size distributions and sulfuric-acid water droplets.

---

## Referee Report (RR1)

**Summary Statement**

O21's defense of their Siberian smoke hypothesis is admirable yet unconvincing. Hinging all their conclusions on the position that one and only one conclusion can be drawn from a peculiar wavelength dependence of lidar ratio places this paper at odds with a mountain of evidence that something other than smoke was in the stratosphere in 2019 before, during, and after O21's hypothesized smoke incursion scenario. If one considers the combination of stratospheric aerosol abundance, omnipresence, and longevity O21 show, this event is thereby in a category with ultra-strong pyroCb events and big volcanic eruptions. It is imperative then for O21 to offer more than a hypothesis if this work is to merit publication in its present form.

The MOSAiC lidar data are an invaluable resource that will illuminate multiple, exciting findings. It is of course exciting to contemplate a new UTLS pathway for smoke to pollute the stratosphere in such a big way. But it would be equally exciting to learn that stratospheric

smoke and volcanic sulfate might both give somewhat similar lidar-signal patterns. Given that there is overwhelming support for a hemispheric volcanic sulfate plume in 2019/20, the MOSAiC lidar-data analysis would be fundamentally improved if Raikoke influence was put on an equal footing (at least) with the hypothesized Siberian smoke explanation.

Major Concerns

O21 are standing by a concise claim that all of the MOSAiC UTLS lidar-detected aerosol they show is wildfire smoke. They give no quarter to other compositions (such as Raikoke sulfate). This is a huge challenge, which not only requires convincing the reader that only smoke can explain their measurements, but also to explain the fate of non-smoke particles in the UTLS that were undoubtably abundant at high latitudes in summer 2019.

The new Fig. 5 shows that all trajectory overpasses of Box 2 occurred before or at the onset the ramp-up of Siberian AOD (Fig. 3). The 7-km trajectory is centered in

Box 2 on 17 July, 4 days before the onset of the large-AOD episode declared by O21. The 9-km trajectory parcel moves rapidly over Box 2 on 20 July, when AOD is unremarkable. For the explanation of self-lofted smoke to be of consequence, conditions within Box 2 on 20 July would have to have been primed by large, low-altitude AOD some days prior. Per O21's analysis of Fig. 1 and 3, such a condition did not exist.  Hence it is unclear why that CALIOP curtain is shown in support of their premise. Moreover, in my original review I explained that CALIOP curtains looked like the 26 July one every day before that for several days.  Thus, there is a consistent picture of ubiquitous UTLS aerosol in place that cannot be attributed to Siberian smoke. Whether the ambient UTLS aerosol is smoke from previous, unrelated injections or Raikoke sulfate, it must be confronted in terms of what happened to it such that it apparently vanished and was replaced by self-lofted Siberian smoke.

O21 base their hypothesis (and an upcoming paper) on Boers et al. and de Laat et al. Boers et al. was a theoretical prelude to de Laat et al., laying the framework for the Solar Escalator paper.  On its own, the Boers et al. paper stands as a still unproven mechanism

for lofting smoke from the lower to upper troposphere. de Laat et al.'s position, that pyroCbs did not occur on Black Saturday, has been contradicted by observations given in multiple publications (BOM, 2009; Cruz et al., 2012; Dowdy et al., 2017). Pumphrey et al. (2011) proved that stratospheric enhancements of Black Saturday emissions were detected on the day after the pyroCbs. If BOM, Cruz, Dowdy were in error, and the Boers/de Laat mechanism was solely responsible for the stratospheric smoke plume documented by Pumphrey et al. and Siddaway and Petelina (2012), it would be reasonable to predict that O21's hypothesized aging, and its impact on particle depolarization, would drive the post-Black Saturday lidar landscape. CALIOP measurements of the stratospheric plume would exhibit the same contradictory signals as claimed by O21. I.e. the Black Saturday stratospheric smoke would embody nil depolarization and thus be dominantly mis-classified as sulfate. This is not the case. A perusal of CALIOP backscatter curtains of ~1.5-month-old Black Saturday smoke reveals native measurements of enhanced depolarization. In fact, it is likely that the enhanced depolarization was a factor in the CALIOP version-4 feature classification scheme. The layers are regularly

labeled as "cirrus" in lock step with classification of the layer as cloud composed of ice. This is in spite of the fact that the layers are above 20 km altitude. An example of one such scene is given here:

https://tinyurl.com/caliopsmoke

O21 are encouraged to survey additional CALIOP aged Black Saturday smoke detections from March 2009. They reveal other spurious classifications (such as volcanic sulfate) mixed with cirrus. It is evident that the best explanation for these features is smoke from Black Saturday. (See Siddaway and Petelina and Pumphrey et a;. for maps of the advected Black Saturday plume in the tropics.).  As with the boreal 2019/20 situation, the lesson is that no single remote-sensing instrument probing the UTLS is sufficient for unambiguous characterization of particulate composition. This is why total reliance on MOSAiC lidars for characterizing three seasons' worth of aerosol observations requires several complementary data items, and the context provided by publications such as Kloss et al. (2021) and Cameron et al. (2020) in addition to Johnson et at. (2021).

The argument made by O21 regarding the published sAOD (e.g. Kloss et al.) falling short of the MOSAiC 0.1 value is without much merit. There is little doubt that sAOD in Kloss et al. is probably biased low, in part for the reason given in O21—saturation. The sAOD values shown therein, peaking at about 0.025, are also an artifact of the broad aerial/temporal averaging applied. Hence they make a poor point of comparison with individual lidar profiles. That being said, it is straightforward to see in CALIOP data that the stratospheric aerosol at high latitude prior to the hypothesized Siberia incursion, far exceeds sAOD=0.025. Take for example a CALIOP curtain on 22 July, with an aerosol layer over North America with native level stratospheric backscatter exceeding .003/sr. Applying a conservative lidar ratio of 50 gives extinction exceeding 0.1. https://tinyurl.com/gtdot1

The MODIS AOD analysis in Fig. 2 is impressive but inconclusive. No accounting is given of any significant difference in peaks. Several additional peaks are also quite impressive. Might one conclude that in those years a similar, scalable impact on the stratosphere was predictable? Were any observed? It should be straightforward to do so with the available ground-based

lidar and satellite remote sensing data sets. Another caveat is that MODIS AOD is severely low-biased in the presence of high-concentration aerosol plumes (Figure 7; Fromm et al., 2008). It is akin to a saturation bias; thick aerosol is classified as cloud. The import here is that there is a huge unknown in any MODIS AOD analysis focused on extraordinary plumes. The Siberia smoke situation in July/August 2019 was indeed extreme, but the true quantifiable extreme here and in many other cases is unknowable based solely on MODIS AOD retrievals. Hence it is unclear how quantifiably unique the Siberia 2019 smoke situation was. That being said, if indeed the Siberia 2019 smoke was lofted to the UTLS, it should be elementarily possible to follow the lofted smoke plume with satellite data such as CALIOP. If O21 can show observations of day-to-day, stepwise escalation of optically dense smoke from its initial placement to the tropopause and beyond (in accord with their preliminary theoretical calculations), this could be a compelling argument to include in the present thesis.

**References**

Bureau of Meteorology (BOM) (2009), Meteorological aspects of the 7 February 2009 Victorian fires, an overview. Bureau of Meteorology report for the 2009 Victorian Bushfires Royal Commission, Bureau of Meteorology, Melbourne, Australia.

Cruz, M. G., A. L. Sullivan, J. S. Gould, N. C. Sims, A. J. Bannister, J. J. Hollis, and R. J. Hurley (2012), Anatomy of a catastrophic wildfire: The Black Saturday Kilmore East fire in Victoria, Australia, For. Ecol. Manage., 284, 269–285.

de Laat, A. T. J., D. C. Stein Zweers, R. Boers, and O. N. E. Tuinder (2012), A solar escalator: Observational evidence of the self-lifting of smoke and aerosols by absorption of solar radiation in the February 2009 Australian Black Saturday plume, J. Geophys. Res., 117, D04204, doi:10.1029/2011JD017016.

Dowdy, A. J., M. D. Fromm, and N. McCarthy (2017), Pyrocumulonimbus lightning and fire ignition on Black Saturday in southeast Australia, J. Geophys. Res. Atmos., 122, 7342–7354, doi:10.1002/2017JD026577.

Fromm, M., O. Torres, D. Diner, D. Lindsey, B. Vant Hull, R. Servranckx, E. P. Shettle, and Z. Li (2008a), Stratospheric impact of the Chisholm pyrocumulonimbus eruption: 1. Earth-viewing satellite perspective. *J. Geophys. Res.,* **113**, D08202, doi:10.1029/2007JD009153.

Pumphrey, H. C., M. L. Santee, N. J. Livesey, M. J. Schwartz, & W. G. Read (2011), Microwave limb sounder observations of biomass-burning products from the Australian bush fires of February 2009. *Atmospheric Chemistry and Physics*, *11*(13), 6285–6296. https://doi.org/10.5194/acp-11-6285-2011.

Siddaway, J. M., and S. V. Petelina (2011), Transport and evolution of the 2009 Australian Black Saturday bushfire smoke in the lower stratosphere observed by OSIRIS on Odin, J. Geophys. Res., 116, D06203, doi:10.1029/2010JD015162.

---

## Editor Decision (ED1)

There is clearly a disagreement between the authors and Mike Fromm on the validity of the inferences in this paper. To resolve these I require a number of points to be clarified.

The first point that I'd like clarified is the assertion very early in the paper (p.2, l.13) that the AOT of the Raikoke volcanic aerosol cloud should have been of order 0.005-0.01 at 500 nm in the winter half year over the Arctic. For the given references, Kloss et al showed 0.01 – 0.02 at high latitude for 675 nm, my observations at 355 nm and 52°N showed AOD above 12 km of 0.01-0.02, Cameron et al do not show AOT at 500 nm and Gorkaiyvi et al show the same measurements as Kloss. A minor point maybe but where do the authors get the 0.005-0.1 range? Is this based on previous eruptions?

More important in this context is the definition of AOT. A meaningful comparison between different AOT values requires them to be over the same height range. According to figs 11 and 12, you calculate AOT from 6 to 15 km, with the maximum at 9 km. There is nothing scientifically wrong with this of course, but much if not most of this aerosol is in the troposphere and comparison with stratospheric AOT measurements is not meaningful. Kloss for example presents sAOD, the integrated AOT above the tropopause.

Indeed it would seem to me that much of the dispute between the authors and Mike Fromm comes from the altitude at which the measurements are made. The clear evidence of smoke particles comes from the measured lidar ratios, but in figs 11 and 12 these are only shown between 7 and 12 km. Is there evidence that the layer above 12 km was actually smoke? The smoothness of the profiles could arise from the considerable smoothing used in the data processing, and hide the possibility that the aerosol layer was not homogenous.  Could the Raikoke aerosol be above the smoke layer? (That would help with the conceptual difficulty many readers will have of the smoke self-lofting to 17 km which is above the region where it was geographically confined). The authors comment on fig. 4 that the blue colours between 13 and 17 km from 45 to 60 N 'may indicate Raikoke – related volcanic aerosol'. Like Mike Fromm, I would contend that it's 'very likely', not 'may'.

Your estimate on p.13 of the Raikoke volcanic aerosol contribution does not consider the possibility that there were two or more aerosol layers (e.g. smoke below the tropopause, smoke + some Raikoke aerosol 1-2 km above the tropopause and volcanic aerosol above). The 10-15% contribution to the extinction coefficient at 532 nm from volcanic aerosol applies (as I understand it) to the entire layer from 6 km upwards, so it is not at all surprising that it is dominated by the smoke.

I don't understand how the self-lofting argument gets smoke into the stratosphere, if the response in the troposphere is to produce plumes. But there is continual exchange of air between the troposphere and the bottom couple of km of the stratosphere so over a couple of months it is entirely possible that some smoke originally in the upper troposphere would be found in the lowest layers of the stratosphere. However, such mixing would not lift the smoke to 17 km, which is well above the extratropical tropopause.

Fig 4 – please give date and time of Calipso overpass in the caption

---

## Author Response (AR2)

**Dear Editor,**

**Please find our answers to the reviews (second round, Mike Fromm, Ref.#1, Ref.#2).**

**First of all, we thank all reviewers for careful reading, their interest, and for making further critical remarks and good suggestions.**

**Our answers in BLUE:**

*Editor comment: Extraordinary claims need extraordinary evidence, and the self-lofting of smoke from the lower troposphere to the stratosphere falls into that category for me.*

Let me (Albert Ansmann) start with the following remark. We shouldn't mix the major goal of the paper which is the occurrence of a thick smoke layer over the North Pole region (from our MOSAiC campaign). The self-lifting aspect is of a second-order importance. The self-lifting aspect was introduced to explain our statement: if we detected smoke, what was the most obvious source for it? So, the reader expects an answer, even if we have only one possible explanation to our finding- yet presented it as hypothesis.

The message of the paper is simple We observed a smoke-dominated aerosol layer over the North Pole region throughout the winter half year, and that's it! And there is no doubt that we observed a smoke-dominating aerosol layer because of the unique potential of a dual wavelength Raman lidar to measure the lidar ratio (LR) at two wavelengths and thus to provide this unambiguous optical fingerprint of smoke, the inverse LR wavelength dependence. This is highlighted in Table 2 and discussed in the Sect.4.1. There is no way for an alternative interpretation. And again, there is no better way to identify smoke than with dual wavelength Raman lidar. However, without the self-lifting aspect, the explanation for the "smoke source" might be missing by a reader. So we need this inspiring aspect to further initiate a scientific discussion.

Back to the paper content, we did a rather comprehensive and careful discussion on the self-lifting. There is nothing to add. Our revision consumed several weeks of preparing convincing – from our opinion- results, generation and revision of figures, and a careful and critical discussion. We clearly left open to the reader to accept or not accept our argumentation. The paper will trigger new research regarding the source of the smoke that was measured over the North Pole region.

Regarding Mike Fromm comments, we were glad to find out that our arguments in a revised version are much more convincing comparing to the original submitted manuscript. However, there were some points for additional discussion.

The main points of Mike Fromm's review to our opinion are, (a) we ignored the volcanic sulfate contribution to the observed aerosol. This point is now improved, Sect 4.1, 4.2; (b) that MODIS AOT maps (Fig.1 and 3) are biased by clouds. Here we provide missing information in Sect.3 now, that we carefully analyzed the data to avoid a cloud contamination bias and provide references that studied the cloud impact during moderate pollution events with the conventional MODIS retrieval that we used (Chudnovsky et al. 2013a, 2013b); and (c) that presentation of observations showing the stepwise ascent of smoke, day by day would be nice. Unfortunately, the latter is not possible, and we explain this point based on simulations (see explanations in Sect 3). Any heating of an absorbing layer in the troposphere will immediately lead to incoherent structures in the aerosol layering, because of much stronger heating at the top than at the center or base of a given layer. This behavior is different for troposphere and stratosphere (in the stratosphere coherent structures may be retained). Instead of the requested observations, we present a first self-lifting simulation figure (new Fig. 6).

Let me first summarize all the improvements we did in our secondly revised version:

- Figures 8, 11, 12, 13, 14, 15, and 18 are now colorblind-safe as recommended by Ref #2. We used blue, green and black/dark-gray colors, only. Fig 10 is not improved because all curves are close together. Colors are unimportant then. See more information at the end of this letter.

- We followed all suggestions of Ref #1, regarding Raman method (<20km), smoke particle density, depolarization ratio, etc, and provide proper references. See step by step answers at the end of this letter.

- We discuss the impact of the Raikoke volcanic sulfate aerosol on the observed aerosol optical properties in more detail (Sect 4.1, 4.2). The Raikoke aerosol fraction was most probably about 10-15\%, and AOT of the order of 0.005 in Dec 2019 to Feb 2020. We mention that also in the abstract and in the conclusion section now.

- We present a new Figure (Fig. 6) with first results of our self lifting simulations in Sect 3, and provide a discussion including the consequences of vertically inhomogeneous heating (highest values at smoke layer top, lowest at layer base) which leads probably to diffuse rather than coherent aerosol structures …. which is reflected in the CALIPSO observations.

**Below please find a point by point response to Mike Fromm comments:**

Summary Statement

O21's defense of their Siberian smoke hypothesis is admirable yet unconvincing. Hinging all their conclusions on the position that one and only one conclusion can be drawn from a peculiar wavelength dependence of lidar ratio places this paper at odds with a mountain of evidence that something other than smoke was in the stratosphere in 2019 before, during, and after O21's hypothesized smoke incursion scenario. If one considers the combination of stratospheric aerosol abundance, omnipresence, and longevity O21 show, this event is thereby in a category with ultra-strong pyroCb events and big volcanic eruptions. It is imperative then for O21 to offer more than a hypothesis if this work is to merit publication in its present form.

**This argumentation was already stated during the first round of review. We are afraid that Mike Fromm does not accept lidar-based results and facts that are evident to all those who work with lidar. The basic Mike concern is: How one can be confident that he/she detected aged smoke?**

**We discussed this point comprehensively. We improved accordingly the text. We introduced a new convincing Table2. Unfortunately it seems that we start from the beginning. Below we clarified our point more comprehensively.**

**The fact is that this** *'peculiar wavelength dependence of lidar ratio'* **(lidar ratio=LR) clearly and unambiguously points to a smoke-dominated source of aerosols. There is no way around this conclusion. Simply- impossible. Obviously, lidar experts can judge and acknowledge this enormous value of multiwavelength Raman lidars. And in the case of the MOSAiC smoke layer, the difference of 30 sr (LR355= 55sr, LR532= 85 sr) is high and does not leave room for a significant (e.g., 50%) contribution of volcanic sulfate aerosol. We cannot ignore it. We state and discuss this result clearly in Sect. 4.1 and 4.2. There is no room for any alternative combination. Modeling of particle optical properties would show that volcanic aerosol (from young to aged), urban haze, dust, and marine particles are unable to produce such a 'peculiar' LR spectral behavior. For readers who are not directly working with lidar data this result was clarified again in our revised version. And we thank Mike for this comment making our revised version more multidisciplinary, better understanding each other scientific language.**

**Importantly, we observe these aged smoke layers since the Lindenberg Aerosol Characterization Experiment 1998 (since more than 20 years, see special issue in JGR on LACE98, 2002), and we summarized our smoke findings including this clear optical fingerprint for aged smoke, for the first time, in the paper of Mueller et al, (JGR 2005), and we detected so many layers since then, and never observed any other different exception cases of this nice LR spectral behavior of aged smoke. The pattern was the same for all our studied cases through many years of research.**

**Considering that- may be our explanation was lost across the revised paper – since Mike did not see it clearly, our discussion was improved and this point was clarified there again (on the ignorance of the volcanic sulfate impact and** *'a mountain of evidence that something other than smoke was in the stratosphere in 2019 before, during, and after O21's hypothesized smoke incursion scenario').*

**Regarding "a mountain of evidence" pointed out by Mike Fromm. It does not match the reality. 'Evidence' already indicates: We (as a whole scientific community) do not know exactly. To be clear again (as during the last round), I do not see this mountain! And why do I not see this mountain? There is no optical method (satellite passive remote sensing) that permits us to decide CLEARLY and UNAMBIGUOUSLY: This is sulfate aerosol! We discussed that already exhaustingly during the first round (first reply letter).**

Motivated by this view of Mike Fromm, I discussed this point in a quite long e-mail conversion with Corinna Kloss (Kloss et al., 2021). She based her work on OMPS observations and came up with her constructive Raikoko ACP paper in early 2021. OMPS allows to retrieve TOTAL extinction (AOD), which is great, but not enough to distinguish between smoke and sulfate particles. Obviously guided by the volcanic aerosol modelers (co authors), Kloss et al (2021) exclusively focused on the Raikoke eruption and nothing else.

However, we understand and aware that we shall include also the impact of the Raikoke sulfate aerosol in the discussion. We improved this now in Sect. 4.1 (last paragraph) and 4.2.

In particulate, if we assume a volcanic sulfate LR of 40 sr (for both wavelengths, see Table 2) and a smoke LR355 of 60sr and a smoke LR532 of 100 sr, and an extinction-related Angstroem exponent of 1.5 for sulfate aerosol, and 0.75 for smoke aerosol, and finally assume a 15% contribution of smoke to the total AOT at 532 nm, only then we can reproduce the found overall LR values of LR355 of 55 sr and LR532 of 85 sr.

But LR355 of 60 and LR532 of 100 sr is already an extreme LR pair, if we would take 60sr and 90 sr, we could reproduce the measured values of about 55sr and 85 sr only with a smoke fraction of 5-10%.

All this is explained now in the final paragraph in Sect. 4.1.

And the final critical comment of Mike Fromm is …… *If one considers the combination of stratospheric aerosol abundance, omnipresence, and longevity O21 show, this event is thereby in a category with ultra-strong pyroCb events and big volcanic eruptions. It is imperative then for O21 to offer more than a hypothesis if this work is to merit publication in its present form…*

This ONE hypothesis of self-lifting explains so many optical features and measurements including the rather low particle depolarization ratio. In this regard, we do not think that we shall include additional arguments to our statements. We want to be on the safe side as much as possible. Any further (speculative) argument will not help. It remains hypothetically. Better to have just ONE, if that already is in consistency with our lidar observation of lidar ratio and depolarization ratio.

Importantly, we do not present our hypothesis in a dogmatic way. We leave it open to the reader to accept our hypothesis or not. In this regard, we truly believe that our paper will be extremely interesting to atmospheric science community initiating multidisciplinary discussion and new research in this direction. We have several views on the observed phenomenon of pollution transport in Polar regions. And we need to be ready that some of the results will transform our understanding of the different chemical and physical processes in the atmosphere.

In this regard, we did not change much in the discussion. Important addition is a simulation of the smoke self-lifting process (new Fig. 6). This will help to get an idea about self lifting times scales and that full particle ageing (after 24—48 hours the aging process is usually completed at tropospheric conditions) is really possible during the slow lifting over days before reaching the UTLS region.

Again, our major argument and a fact based on our measurements is: There was a dense smoke layer over the North Pole area…. and the logical consequence then is: Please tell us: What was the source? And if you do not know exactly: Please provide your opinion (hypothesis)? Only that is expected by a reader … and exactly that was done (comprehensively) in our paper.

If there would be a clear link (between the MOSAiC smoke layer and Siberian fires in July/August 2019), we would be happy to provide it. But we did not find any other possible source and stick to Siberian fires argumentation- which seems logical. This fire event was clearly remarkable and very large on a

**spatial extent: by the end of the July 2019, the size of the fires reached 2,600,000 hectares. While Siberian wildfires are common during summer, record-breaking high temperatures and strong winds have made 2019 fires particularly devastating (https://www.bbc.com/news/world-europe-49224776). We offer a careful and detail discussion with many arguments and figures and argue that perhaps Siberian fires were the source of smoke in the Polar region. And all details provide a consistent picture. If later one, scientists find a better explanation (triggered by this paper), that will be great! We do not state, that we are right. We carefully state again and again: this is just a hypothesis, but for - the most convincing one so far.**

The MOSAiC lidar data are an invaluable resource that will illuminate multiple, exciting findings. It is of course exciting to contemplate a new UTLS pathway for smoke to pollute the stratosphere in such a big way. But it would be equally exciting to learn that stratospheric smoke and volcanic sulfate might both give somewhat similar lidar-signal patterns. Given that there is overwhelming support for a hemispheric volcanic sulfate plume in 2019/20, the MOSAiC lidar-data analysis would be fundamentally improved if Raikoke influence was put on an equal footing (at least) with the hypothesized Siberian smoke explanation.

**After** *'mountain of evidence'* **we get confronted with** 'overwhelming support for a hemispheric volcanic sulfate plume in 2019/20'. **However, although bombarded by these overwhelming arguments, we remain biased by our solid, clear, and unambiguous observation that a smoke-dominated aerosol covered the North Pole throughout the winter half year of 2019/20.**

**And regarding similar lidar-signal patterns: The relevant literature and own simulations of particle optical properties 'teach' us …….. that aged LIQUID sulfuric-acid-containing water droplets of volcanic origin with a typical and realistic accumulation-mode size distribution and simple, well-known refractive index characteristics for sulfuric-acid containing water droplets (with single scattering albedo close to 1.0) are UNABLE to produce lidar ratio of 40-50 sr at 355 nm and at the same time lidar ratios of 70-90 sr at 532 nm. Impossible! All the papers on volcanic aerosol (which partly include modelling results) show this clearly. Especially for aged volcanic sulfate layers (several month after formation), there is almost no way to produce significant differences between LR355 and LR532. We corroborated this simulation-based finding by our dual-wavelength Raman lidar measurements in 2009 (Mattis et al, JGR, 2010). These Mattis et al. volcanic LR355 and LR532 values are given in Table 2. We found LR355 and LR532 of about 40 sr. And in the case of these highly light-absorbing smoke particles with complex core-shell configuration and partly in glassy state, the story is quite RATHER different.**

**And regarding…** *'Given that there is overwhelming support for a hemispheric volcanic sulfate plume in 2019/20, the MOSAiC lidar-data analysis would be fundamentally improved if Raikoke influence was put on an equal footing (at least) with the hypothesized Siberian smoke explanation'.*

**What can we answer? After 'mountain of evidence' and 'overwhelming support…, now 'equal footing'. No comment from our side. We are exhausted!**

**As mentioned previously, we try to emphasize now more clearly that Raikoke sulfate aerosol may have contributed by 10-15% to the observed particle extinction coefficient and AOT at 532 nm, and that this is in agreement with the expected sulfate AOD of 0.005 (or max values of 0.01) in Dec 2019-Feb 2020 at high northern latitudes.**

O21 are standing by a concise claim that all of the MOSAiC UTLS lidar-detected aerosol they show is wildfire smoke. They give no quarter to other compositions (such as Raikoke sulfate). This is a huge challenge, which not only requires convincing the reader that only smoke can explain their measurements, but also to explain the fate of non-smoke particles in the UTLS that were undoubtably abundant at high latitudes in summer 2019.

**We re-checked the text to avoid the impression that we completely ignored the volcanic aerosol. We explained that already in this reply letter above.**

The new Fig. 5 shows that all trajectory overpasses of Box 2 occurred before or at the onset the ramp-up of Siberian AOD (Fig. 3). The 7-km trajectory is centered in Box 2 on 17 July, 4 days before the onset of the large- AOD episode declared by O21. The 9-km trajectory parcel moves rapidly over Box 2 on 20 July, when AOD is unremarkable.

**The analysis of the trajectories presented by Mike From is incorrect. The red trajectory (arriving at 3 km height) was clearly within BOX 2 on 22 and 23 July 2019 (and thus during the heavy fire days) and even within BOX 1 (the hot spot fire region). The fires were intense from 19 July to 14 August 2019. Then the smoke had 3-4 days to ascend before CALIPSO came along and measured the lofted smoke. In these 3-4 days, the smoke can be lifted by 5-8 km as our simulations show. The new Fig. 6 corroborates that.**

**BOX 1 and BOX 2 were arbitrarily defined to better guide the reader to the most intense fire places. But there was also smoke to the left of BOX 2 with monthly mean AOD around 0.8. So, even the 7 km backward trajectory clearly shows… that the air mass was close to the ground (below 3000m) until 22 July. So, there were four days to accumulate smoke since 19 July 2019.**

**Finally, yes, the 9 km trajectory was not in direct contact with emitted smoke plumes. We state that in the manuscript. So, we need self-lifting to get smoke into the 9 km height range. Obviously, this was the case (from 3-9 km in about 2-6 days, according to Fig.6).**

**Regarding 'ramp-up', this was not a ramp-up event… (like a pryoCb event). It was a smooth change in the layering structures that occurred in the end of July and beginning of August. Even over Leipzig (51 N), and thus not only over latitudes >70N, we saw a significant change in the aerosol geometrical (and optical) properties, between 29 July and 5 August 2019, from the occurrence of only sharp and geometrically thin layers (usually of 500 to 1000 m vertical extent) to more complex conditions with a smooth and thick layer structure (with vertical extent over several kilometers) just above the tropopause in addition to the sharp layers above 15 km height. All this will be shown in a new paper in preparation (Ansmann et al.: Misclassification of stratospheric Siberian fire smoke as Raikoke volcanic aerosol in 2019 by the CALIPSO aerosol-typing scheme). This paper will be submitted in August/September 2021.**

For the explanation of self-lofted smoke to be of consequence, conditions within Box 2 on 20 July would have to have been primed by large, low-altitude AOD some days prior.

**The smoke needs 2-6 days to ascend from 3-4 km height (typical injection height) to 9-10 km height. So, to explain CALIPSO observation on 26 July, we need smoke at low level on 21-23 July (and not before 20 July).**

**Another aspect that became clear to us based on the simulations: The heating of a given dense smoke layer is vertically inhomogeneous. For example, at cloud-free conditions, strong heating occurs at the top and much lower heating occurs at the center and even much lower heating occurs at the base of a given layer. This means, you will find immediately incoherent structures. There is no way to lift the smoke layer as a whole to large**

**tropospheric heights. And exactly these diffuse (almost random) structures are visible in the CALIPSO lidar observations.**

**This feature becomes different for stratospheric heights (for stable layering conditions, very different profile of pot. temp. compared to the profile in the troposphere). Smoke plumes reaching the stratosphere (e.g., by pyroCB activity) and forming stratospheric layers can keep their layer structures for a long time during the ascent by roughly 3-7 km during a 10000 km travel, as we showed in the Ohneiser et al. (2020) paper on Australian smoke observed over southern Chile for layers arriving in the beginning of January 2020.**

Per O21's analysis of Fig. 1 and 3, such a condition did not exist. Hence it is unclear why that CALIOP curtain is shown in support of their premise. Moreover, in my original review I explained that CALIOP curtains looked like the 26 July one every day before that for several days. Thus, there is a consistent picture of ubiquitous UTLS aerosol in place that cannot be attributed to Siberian smoke. Whether the ambient UTLS aerosol is smoke from previous, unrelated injections or Raikoke sulfate, it must be confronted in terms of what happened to it such that it apparently vanished and was replaced by self-lofted Siberian smoke.

**Yes, there was Raikoke aerosol in the stratosphere before 26 July, nobody is telling the opposite. But in the beginning of August there was quite a big change in observable aerosol properties. Please, start to accept this fact!**

**The CALIPSO lidar observation are in full agreement with the simulations that lead to diffuse layering structures in the free torposphere, and therefore we show this CALIPSO observations. These observations perfectly support our hypothesis. So we are puzzled by Mike Fromm comment stating that** 'there is a consistent picture of ubiquitous UTLS aerosol in place that cannot be attributed to Siberian smoke'. **We agree, if we discuss all the sharp layers above the tropopause (with 500 to 1000 m vertical extent). Most of them seem to be related to Raikoke aerosol. But all the diffuse layers (with 2-4 km vertical extent) in the UTLS regime seem to be of different origin. In the new paper (Ansmann et al., misclassification … when using the CALIPSO aerosol typing scheme), we will show an example (Leipzig, Polly lidar, 14 August 2019, layer from tropopause to 4 km above tropopause, layer mean LR355=70sr, LR532=105sr), so again…. clearly smoke! But depol was low as measured by Polly and by the CALIPSO lidar (near Leipzig overflight on 14 August 2019, within the Polly measurement window) so that CALIPSO announced: Sulfate aerosol!**

O21 base their hypothesis (and an upcoming paper) on Boers et al. and de Laat et al. Boers et al. was a theoretical prelude to de Laat et al., laying the framework for the Solar Escalator paper. On its own, the Boers et al. paper stands as a still unproven mechanism for lofting smoke from the lower to upper troposphere. de Laat et al.'s position, that pyroCbs did not occur on Black Saturday, has been contradicted by observations given in multiple publications (BOM, 2009; Cruz et al., 2012; Dowdy et al., 2017).

Pumphrey et al. (2011) proved that stratospheric enhancements of Black Saturday emissions were detected on the day after the pyroCbs. If BOM, Cruz, Dowdy were in error, and the Boers/de Laat mechanism was solely responsible for the stratospheric smoke plume documented by Pumphrey et al. and Siddaway and Petelina (2012), it would be reasonable to predict that O21's hypothesized aging, and its impact on particle depolarization, would drive the post-Black Saturday lidar landscape.

**The paper of Boers et al. (2010) is clearly the first paper that points to the smoke self lifting potential. I studied all papers in this field already in 2017.**

**On the other hand, there is no doubt that the Black Saturday smoke (in 2009) reached the stratosphere by pyroCb activity. And CALIPSO depolarization ratios were clearly enhanced, a clear sign for the impact of pyroCb activity.**

CALIOP measurements of the stratospheric plume would exhibit the same contradictory signals as claimed by O21. I.e. the Black Saturday stratospheric smoke would embody nil depolarization and thus be dominantly mis-classified as sulfate. This is not the case. A perusal of CALIOP backscatter curtains of ~1.5-month-old Black Saturday smoke reveals native measurements of enhanced depolarization. In fact, it is likely that the enhanced depolarization was a factor in the CALIOP version-4 feature classification scheme. The layers are regularly labeled as "cirrus" in lock step with classification of the layer as cloud composed of ice. This is in spite of the fact that the layers are above 20 km altitude. An example of one such scene is given here: https://tinyurl.com/caliopsmoke

**Agreement at all. The consequence of fast lifting by pryoCB convection is that smoke has no time for aging and developing a spherical shape. So, depol is enhanced.**

O21 are encouraged to survey additional CALIOP aged Black Saturday smoke detections from March 2009. They reveal other spurious classifications (such as volcanic sulfate) mixed with cirrus. It is evident that the best explanation for these features is smoke from Black Saturday. (See Siddaway and Petelina and Pumphrey et a;. for maps of the advected Black Saturday plume in the tropics.). As with the boreal 2019/20 situation, the lesson is that no single remote-sensing instrument probing the UTLS is sufficient for unambiguous characterization of particulate composition. This is why total reliance on MOSAiC lidars for characterizing three seasons' worth of aerosol observations requires several complementary data items, and the context provided by publications such as Kloss et al. (2021) and Cameron et al. (2020) in addition to Johnson et at. (2021).

**We do not agree with your comment. We totally agree that the combination of many techniques is always of advantage, to create a trustworthy, consistent view on the observed (puzzle-like) features and processes. However, we disagree when all these techniques only provide 'evidence' and 'consistency' but not unambiguity or clarity! ... as the dual wavelength Raman lidar is able to provide. This makes a big difference. We rely on our good measurements and (my) almost 40 years of experience in the field of lidar aerosol field observations.**

The argument made by O21 regarding the published sAOD (e.g. Kloss et al.) falling short of the MOSAiC 0.1 value is without much merit. There is little doubt that sAOD in Kloss et al. is probably biased low, in part for the reason given in O21—saturation. The sAOD values shown therein, peaking at about 0.025, are also an artifact of the broad aerial/temporal averaging applied. Hence they make a poor point of comparison with individual lidar profiles. That being said, it is straightforward to see in CALIOP data that the stratospheric aerosol at high latitude prior to the hypothesized Siberia incursion, far exceeds sAOD=0.025. Take for example a CALIOP curtain on 22 July, with an aerosol layer over North America with native level stratospheric backscatter exceeding .003/sr. Applying a conservative lidar ratio of 50 gives extinction exceeding 0.1. https://tinyurl.com/gtdot1

**The volcanic aerosol was inhomogeneously distributed over the Northern Hemisphere, but the general Raikoke impact and the general trends are clear in Kloss et al. 2021 paper, and show an increasing AOT with increasing latitude, which is in full contradiction with model results (Sarychev, 2009, Raikoke, 2019) of a homogeneous spread of the volcanic aerosol over the northern part of the Northern Hemisphere and not an increase of AOT towards the North Pole. All this was already discussed last time and is written in our revised manuscript (already last time).**

The MODIS AOD analysis in Fig. 2 is impressive but inconclusive. No accounting is given of any significant difference in peaks. Several additional peaks are also quite impressive. Might one conclude that in those years a similar, scalable impact on the stratosphere was predictable? Were any observed? It should be straightforward to do so with the available ground-based lidar and satellite remote sensing data sets. Another caveat is that MODIS AOD is severely low-biased in the presence of high-concentration aerosol plumes (Figure 7; Fromm et al., 2008). It is akin to a saturation bias; thick aerosol is classified as cloud. The

import here is that there is a huge unknown in any MODIS AOD analysis focused on extraordinary plumes. The Siberia smoke situation in July/August 2019 was indeed extreme, but the true quantifiable extreme here and in many other cases is unknowable based solely on MODIS AOD retrievals. Hence it is unclear how quantifiably unique the Siberia 2019 smoke situation was. That being said, if indeed the Siberia 2019 smoke was lofted to the UTLS, it should be elementarily possible to follow the lofted smoke plume with satellite data such as CALIOP. If O21 can show observations of day-to-day, stepwise escalation of optically dense smoke from its initial placement to the tropopause and beyond (in accord with their preliminary theoretical calculations), this could be a compelling argument to include in the present thesis.

**Now we enter another new field- MODIS AOD bias in the products.**

**As mentioned previously, we present a simulation of self-lifting (Fig 6), we cannot show observations of coherently ascending layers. Vertically inhomogeneous heating of the smoke layers lead to turbulent aerosol structures. This is written in the manuscript (Sect. 3).**

**Alexandra Chudnovsky (our co author)  wrote:  Very important critical comment. However, I do not think that AOD maps were so largely biased by cloud-contaminated pixels. Based on my experience and works, the AOD retrieval is biased during low pollution events (no AODs are generated) and for high/thick, dense dust storms (misinterpretation with clouds). For the Eastern USA- during smoke events (pollution transport following Canadian fires), the AOD data was provided and clouds were perfectly masked. This area as we know is largely covered by clouds and only 40-50% of yearly data is available for the analyses. For Siberian smoke- I looked at daily RGB and AOD maps. The latter show relatively high retrieval rate- the smoke was not so thick with a nice and spatially continues pattern. For cloud contamination I would expect spurious /spatially uneven pattern.**
**Here we deal with a moderate pollution event, of a very large spatial extent, clouds were masked and the algorithm – I would say- aggressive- meaning that also "good pixels" are excluded largely reducing the data base. I looked at all AOD and RGB images during July 20- August 20- 2019 and selected high AOD values- comparing it to cloud masked ones. For cloud adjacent pixels we have uneven spatial pattern – and still it did not bias the general view of montly averaged AOD. For some pixels- the AOD was not retrieved although one can see yellowish smoke above clouds on RGB images. I would say that we rather underestimate the strength of AOD spatial pattern during this event.**

**Some publications on the impact of clouds on the quality of AOD retrieval in several publications. For example- Chudnovsky et al. 2013a; 2013b; 2014; Kloog et al. 2014; Rogozovsky et al. 2021;**

**Final remark**

**Thank You Mike for all the questions and critical remarks, for your genuine  interest in our work, spending so much time to read the different versions of the manuscript and comments writing.  We learned a lot!**

**However, at the end I should add: I am working in the aerosol lidar field (plus passive remote sensing of aerosols from space and from ground, AERONET) since almost 40 years, and prepared about 50 papers as main writing author, and I must admit, this paper is one of the most exciting, sound, concise, and well-organized paper I was ever involved.**

**So now, we think, it is time  to give readers a chance to 'learn' more about aerosols in the North Pole range in the winter of 2019/20, and as we now know, it was some kind of a mix or Raikoke and Siberian smoke aerosols.**

**Ref #2**

To the extent possible, the authors should make an effort to ensure their figures are colorblind-safe. By my reckoning, Figures 7, 9, 11, 13, and 17 use color combinations that actually cause ambiguities and issues with distinguishing figure elements under different color deficiencies. Other figures could also be revisited. For example, in the figures with only 3-4 colors (such as Figures 10 and 14), there are better color choices that can be used than red, blue, and green. There are many web resources available (e.g., colorbrewer2, Adobe Color's accessibility tools, etc.) that can be leveraged to pick safe choices.

**We considered this aspect now. See, Figs. 8, 11, 12, 13, 14, 15, and 18 (because of new Fig.6, new numbers). We try to use just blue (for 355 nm results) and green (for 532 nm results) and now, for 1064 nm or for PSCs, we select the color of dark gray to black. We checked several web pages and think this combination is ok…. Most important is to avoid red, when using green and blue.**

**Ref #1**

Suggestions for revision or reasons for rejection (will be published if the paper is accepted for final publication)

The authors have substantially improved the article after the reviews. They addressed in detail the major points of concern raised in my review. The article can thus be published after following minor revisions have been considered:

P4 - L29: The sentence "The Raman lidar method was exclusively used to determine particle backscatter and extinction profiles" is not completely correct since it is mentioned a few lines later that "At heights above 20 km, the backscatter and extinction properties of the stratospheric background aerosol are determined from the elastic backscatter signal profiles (alone) by assuming a particle extinction-to-backscatter ratio (lidar ratio) of 50 sr".

**We improved this: Raman lidar method is exclusively used for smoke (<20 km), and higher up the Fernald method is used.**

P5 - L11: what is the reference for the fact that dust cause depolarization ratios around 0.3 at 532 nm?

**We now use Gross et al. (2015) here, this reference was already included in the paper.**

P5 - L25: The Baars et al (2016) reference is mentioned for the least-squares regression analysis used to determine particle extinction and extinction-to-backscatter ratio profiles, but I did not find an explanation of the method in the mentioned reference.

**The references are now properly given: Pappalardo et al., 2004, and Russo et al., 2006.**

P6 - L9: Some more explanation is required about the value of the smoke density used to obtain smoke mass concentrations.

**We extended the text and provide some more information from the Ansmann et al (2021) paper, which is now published.**

---

## Author Response (AR3)

**Dear Editor!**

**Thank you for your comments to further clarify several critical points. We worked on this and added the requested information in the revised version of the article.**

**Our answers in the step-by-step reply in BLUE.**

There is clearly a disagreement between the authors and Mike Fromm on the validity of the inferences in this paper. To resolve these I require a number of points to be clarified.

**We provide answers to all of the points and improved the manuscript accordingly!**

The first point that I'd like clarified is the assertion very early in the paper (p.2, l.13) that the AOT of the Raikoke volcanic aerosol cloud should have been of order 0.005-0.01 at 500 nm in the winter half year over the Arctic. For the given references, Kloss et al showed 0.01 – 0.02 at high latitude for 675 nm, my observations at 355 nm and 52°N showed AOD above 12 km of 0.01-0.02, Cameron et al do not show AOT at 500 nm and Gorkaiyvi et al show the same measurements as Kloss. A minor point maybe but where do the authors get the 0.005-0.1 range? Is this based on previous eruptions?

**We think that a compact Introduction (of a paper with focus on wildfire smoke over the North Pole region) is not the right place for the necessary and then quite detailed discussion of this aspect. So, we just say in Section 1 that a detail discussion is given in Sect. 3.**

**In Sect. 3, all the details regarding the potential Raikoke impact (max AOT, and decrease of the impact with time) to the stratospheric aerosol burden over the North Pole region are given within three paragraphs.**

**Here, we also introduce the reasonable hypothesis that the smoke dominated in the height range from 6 to 12-13 km height, and the volcanic sulfate aerosol may have dominated above 13-15 km height up to 17-19 km height.**

**Afterwards, we mention this potential smoke-sulfate layering several times, on pages 11, 14, and 15.**

More important in this context is the definition of AOT. A meaningful comparison between different AOT values requires them to be over the same height range. According to figs 11 and 12, you calculate AOT from 6 to 15 km, with the maximum at 9 km. There is nothing scientifically wrong with this of course, but much if not most of this aerosol is in the troposphere and comparison with stratospheric AOT measurements is not meaningful. Kloss for example presents sAOD, the integrated AOT above the tropopause.

**We avoid any comparison of UTLS AOT and pure stratospheric AOT. We also avoid to state that the Siberian smoke formed a stratospheric layer. We mention, throughout the paper, that the smoke contributed to an UTLS aerosol layer.**

Indeed it would seem to me that much of the dispute between the authors and Mike Fromm comes from the altitude at which the measurements are made. The clear evidence of smoke particles comes from the measured lidar ratios, but in figs 11 and 12 these are only shown between 7 and 12 km. Is there evidence that the layer above 12 km was actually smoke?

**No! By using the smoothing lengths of 2000 m, lidar values at 12 km (maximum height shown) are based on signals measured up to 13 km height. So, there is no evidence for smoke occurrence at heights above about 13 km height.**

The smoothness of the profiles could arise from the considerable smoothing used in the data processing, and hide the possibility that the aerosol layer was not homogenous. Could the Raikoke aerosol be above the smoke layer? (That would help with the conceptual difficulty many readers will have of the smoke self-lofting to 17 km which is above the region where it was geographically confined).

**Yes! We agree with this argumentation and state that now at several places (page 11, 14, and 15).**

The authors comment on fig. 4 that the blue colours between
13 and 17 km from 45 to 60 N 'may indicate Raikoke – related volcanic aerosol'. Like Mike Fromm, I would contend that it's 'very likely', not 'may'.

**We changed that accordingly, in the figure caption as well as in the text. On page 8, we now state that CALIPSO observations indicate smoke up to the tropopause. We removed any impression that the smoke was already in the stratosphere above the Siberian fire areas.**

Your estimate on p.13 of the Raikoke volcanic aerosol contribution does not consider the possibility that there were two or more aerosol layers (e.g. smoke below the tropopause, smoke + some Raikoke aerosol 1-2 km above the tropopause and volcanic aerosol above). The 10-15% contribution to the extinction coefficient at 532 nm from volcanic aerosol applies (as I understand it) to the entire layer from 6 km upwards, so it is not at all surprising that it is dominated by the smoke.

**You are right! Now we present two different approaches or ways to estimate the Raikoke sulfate contribution of 10-15%, on page 14. We think that the revised version of the manuscript meets properly all the points Mike Fromm criticized. The Raikoke sulfate aerosol is no longer ignored.**

I don't understand how the self-lofting argument gets smoke into the stratosphere, if the response in the troposphere is to produce plumes. But there is continual exchange of air between the troposphere and the bottom couple of km of the stratosphere so over a couple of months it is entirely possible that some smoke originally in the upper troposphere would be found in the lowest layers of the stratosphere. However, such mixing would not lift the smoke to 17 km, which is well above the extratropical tropopause.

**On page 9, we mention that smoke reaching the upper troposphere can easily enter the stratosphere over the High Arctic: When these lifted smoke plumes are afterwards advected northward they can easily enter the lower stratosphere over the High Arctic because the tropopause is no longer a barrier for airmass exchange between the upper troposphere and lower stratosphere.**

**Such a statement should be sufficient as an argument that we found smoke up to 12 or 13 km height with the tropopause at 8-10 km height.**

**The smoke had time in August and September 2019 to get further lifted up to 12-13 km height, slowly, but steadily. We observed smoke up to 14 km height at Leipzig on 14 August 2019. A probably similar ascent (slowly and steadily) was observed after the Canadian fires in 2017 where the smoke, injected by pyroCb to 11-13km height, was finally found up to 23 km height.**

**On page 10 we state: (1) The self-lifting hypothesis is of key importance in our attempt to relate the MOSAiC UTLS smoke observations from October 2019 to May 2020 with the strong fires in Siberia in July-August 2019.  We leave open here in which way the smoke was further lifted into the lower stratosphere. Khaykin et al. (2018) observed  smoke layers, injected by pyroCb activity into the tropopause region over Canada in August 2017, which further ascended by 2-3 km per day during the first days after injection. After a few weeks, theses layers were found up to 23~km height and distributed over large parts of the northern hemisphere (Baars et al., 2019, Torres et al., 2020).**

Fig 4 – please give date and time of Calipso overpass in the caption

**We improved this.**

**Besides all these improvements, we added the following points:**

- **On page 8, we had to mention, that pyroCb activity was absent over Siberia in July and August 2019.  Peterson et al. (2021) stated that pyorCb activity was responsible for the Siberian smoke in the UTLS height range. This is wrong. And this was the main reason to introduce the self-lifting concept.**
- **In Section 3, and especially in Section 3.1, we introduce our new paper, Ansmann et al. (2021b), on the 'misclassification of Siberian smoke as Raikoke sulfate aerosol in 2019 by the CALIPSO aerosol typing scheme'. This paper was submitted to Frontiers Earth Science end of August, and is now under review. The paper is a contribution to a wildfire-smoke-related Special Issue. In this paper, we present and discuss the lidar observations of smoke layers at Leipzig performed on 14 August 2019 in detail.**
- **Finally, on page 16, in Sect. 4.2, we compare the Siberian fire smoke amount with the ones of the Canadian fire event in 2017 and the Australian smoke event in 2019-2020. We write:….. and yield 0.2 Mt of smoke as a guess for the mean value of the smoke aerosol load over the Arctic during the winter half year 2019-2020. The overall Siberian smoke particle mass injected into the UTLS height range of the northern hemisphere may have been a factor of 2 higher. For comparison, the initial particle mass injection of the record-breaking Canadian pyroCb smoke event into heights of 11-13 km in August 2017 was about 0.3 Mt (Yu et al., 2019) and the rather strong Australian bushfires in December 2019 and January 2020 caused a stratospheric smoke particle mass of the order of 0.5-1 Mt (Peterson et al., 2021).**

---

## Author Response (AR4)

**Dear Editor!**

**Thank you for your comments to further clarify several aspects.**

**Our answers in the step-by-step reply in BLUE.**

Thank you for your response to my comments. It is much clearer now what was actually observed with the lidar, but on going through the paper I realise that the rest of the text needs to be brought in line with the interpretation that there was a thick smoke layer up to around 12 km and aerosol above that whose microphysical properties could not be determined, and is likely to contain at least some volcanic aerosol.

**Thank you very much for your priceless support by careful reading and checking the entire manuscript again as an expert!**

I have made a number of comments in the attached pdf which I think need to be addressed to make the paper self-consistent.

**We followed all instructions and most of them are highlighted in BOLD in the revised version.**

I would also ask the authors to look critically at section 5, since there is now no overlap between the height range where smoke was measured and that where ozone was depleted. At the very least that point needs to be clearly emphasized.

**We worked carefully again on Section 5.**

**We improved Figure 18a and b. Now smoke and sulfate influences are clearly indicated.**

**We carefully checked the text of Section 5 for 'smoke' and substituted 'smoke' by 'aerosol' or 'smoke and sulfate'.**

**We use the paper of Inness et al. (Figure 18b is based on this paper) in the discussion. This Innes et al paper is based on observations (CAMS high quality re-analysis data).**

**We do not agree that ozone loss was only given from about 18 to 22 km height (where almost total ozone depletion, a real hole, occurred). We think, aggressive chlorine was activated everywhere above the tropopause. And when the sunlight was back, ozone became destroyed even at 10-12 km height, i.e., in the smoke layer.**

**This is quite an open discussion (Section 5) and to a considerable extent just speculative. However, we feel we MUST present this Section 5! Otherwise, others are just waiting to take that nice topic as their own original topic. But we want to be able to give reference to our own paper(s) later on when we, e.g., will focus on Australian smoke and the record ozone loss in September 2020 over Antarctica.**

**We added a new (final) paragraph at the end of Section 5 to provide an impression (some numbers) of the potential impact of aerosol on ozone loss (just by chlorine activation, as a function of the surface area concentration).**

**We updated the conclusions (ozone hole paragraph, to be in line with the updated Section 5).**